# Persistent PKA activation redistributes Na$_V$1.5 to the cell surface of adult rat ventricular myocytes

Tytus Bernas[1], John Seo[2], Zachary T. Wilson[2], Bi-hua Tan[3], Isabelle Deschenes[3], Christiane Carter[4], Jinze Liu[4], and Gea-Ny Tseng[2]

During chronic stress, persistent activation of cAMP-dependent protein kinase (PKA) occurs, which can contribute to protective or maladaptive changes in the heart. We sought to understand the effect of persistent PKA activation on Na$_V$1.5 channel distribution and function in cardiomyocytes using adult rat ventricular myocytes as the main model. PKA activation with 8CPT-cAMP and okadaic acid (phosphatase inhibitor) caused an increase in Na$^+$ current amplitude without altering the total Na$_V$1.5 protein level, suggesting a redistribution of Na$_V$1.5 to the myocytes' surface. Biotinylation experiments in HEK293 cells showed that inhibiting protein trafficking from intracellular compartments to the plasma membrane prevented the PKA-induced increase in cell surface Na$_V$1.5. Additionally, PKA activation induced a time-dependent increase in microtubule plus-end binding protein 1 (EB1) and clustering of EB1 at myocytes' peripheral surface and intercalated discs (ICDs). This was accompanied by a decrease in stable interfibrillar microtubules but an increase in dynamic microtubules along the myocyte surface. Imaging and coimmunoprecipitation experiments revealed that Na$_V$1.5 interacted with EB1 and β-tubulin, and both interactions were enhanced by PKA activation. We propose that persistent PKA activation promotes Na$_V$1.5 trafficking to the peripheral surface of myocytes and ICDs by providing dynamic microtubule tracks and enhanced guidance by EB1. Our proposal is consistent with an increase in the correlative distribution of Na$_V$1.5, EB1, and β-tubulin at these subcellular domains in PKA-activated myocytes. Our study suggests that persistent PKA activation, at least during the initial phase, can protect impulse propagation in a chronically stressed heart by increasing Na$_V$1.5 at ICDs.

## Introduction

Persistent β-adrenergic receptor stimulation occurs in chronically stressed hearts, such as heart failure and aging (de Lucia et al., 2018). Activation of β-adrenergic downstream effector networks including cAMP-dependent protein kinase (PKA), in concert with other neurohumoral responses, can induce electrical, structural, and metabolic remodeling in the heart, leading to protective but eventually maladaptive consequences. The effects of β-adrenergic stimulation are most often discussed in the context of "fight-or-flight" syndrome and are related to acute changes in Ca$^{2+}$, K$^+$, and Na$^+$ channels (Bers, 2002; Marx et al., 2002; Matsuda et al., 1992). How persistent PKA activation affects ion channels in cardiac myocytes is less well studied.

Our aim is to study whether and how persistent PKA activation affects the function and distribution of Na$_V$1.5 channels in cardiac myocytes. Na$^+$ current through the Na$_V$1.5 channels triggers the upstroke of cardiac action potentials. Na$_V$1.5 channels localized at the intercalated disc (ICD) region provide the depolarizing current critical for impulse propagation through gap junctions between myocytes. Since modulating microtubules can affect the Na$_V$1.5 function in cardiac myocytes

(Casini et al., 2010) and microtubule plus-end binding protein 1 (EB1, also known as microtubule-associated protein RP/EB family member 1, MAPRE1) may be involved in targeting Na$_V$1.5 to the ICDs (Marchal et al., 2021), we investigate the effects of persistent PKA activation on microtubules and EB1.

Our data show that persistent PKA activation promotes Na$_V$1.5 trafficking from an intracellular reservoir to myocytes' peripheral membrane and ICDs. These are accompanied by an increase in EB1 expression, concentration of EB1 and dynamic microtubules to myocytes' peripheral membrane and ICDs, and enhanced Na$_V$1.5 interaction with both EB1 and β-tubulin. Together, these observations suggest a scenario whereby persistent PKA activation promotes Na$_V$1.5 trafficking to its destinations by providing dynamic microtubule tracks and enhanced guidance by EB1.

## Materials and methods
### Molecular constructs and adenovirus production
The GFP-Na$_V$1.5 construct was a kind gift from Dr. Hugues Abriel (University of Bern, Bern, Switzerland). It has eGFP

---

[1]Department of Anatomy and Neurobiology, Virginia Commonwealth University, Richmond, VA, USA; [2]Department of Physiology and Biophysics, Virginia Commonwealth University, Richmond, VA, USA; [3]Department of Physiology and Cell Biology, The Ohio State University, Columbus, OH, USA; [4]Massey Center Bioinformatics Shared Resource, Virginia Commonwealth University, Richmond, VA, USA.

Correspondence to Gea-Ny Tseng: gea-ny.tseng@vcuhealth.org.



directly fused to the N-terminus of $Na_V1.5$ (isoform hH1a). Human EB1 with monomeric red fluorescent protein fused to the C-terminus through a peptide (RDP) linker (EB1-mRFP) was a gift from Tim Mitchison and Jennifer Tirnauer (plasmid #39323; Addgene; Harvard Medical School, Boston, MA). Both constructs were subcloned into shuttle vectors (by GenScript) for adenovirus production (by Vector Biolabs). The plaque-forming unit (PFU) titers were Adv-GFP-Nav1.5, $1.1 \times 10^{10}$ PFU/ml, and Adv-EB1-mRFP $1.5 \times 10^{10}$ PFU/ml.

## Ventricular myocyte isolation, culture, and adenovirus transduction

All experiments on cardiac myocytes were conducted using young (3–5 mo) male Sprague–Dawley rats. The investigation conformed to the Guide for the Care and Use of Laboratory Animals published by the National Institutes of Health. The animal protocol (AM10294) was reviewed and approved annually by the Institutional Animal Care and Use Committee of Virginia Commonwealth University.

Ventricular or atrial myocytes were isolated as described previously (Zankov et al., 2019; Wilson et al., 2021). Briefly, the heart was cannulated through aorta and mounted on a Langendorff apparatus, retrogradely perfused with the following solutions (36°C, oxygenated): (a) normal Tyrode's (in mM: NaCl 146, KCl 4, $CaCl_2$ 2, $MgCl_2$ 0.5, HEPES 5, and dextrose 5.5, pH 7.3) for 10 min, (b) nominally Ca-free Tyrode's ($CaCl_2$ replaced by equimolar $MgCl_2$) with 1% fat-free bovine serum albumin (BSA) for 5 min, (c) same Ca-free, BSA-containing solution with collagenase II (0.3 mg/ml; Worthington) for 15–25 min, and (d) Kraftbruhe medium (KB, in mM: glutamic acid 120, KOH 120, $KH_2PO_4$ 10, $MgSO_4$ 1.8, EGTA 0.5, taurine 10, HEPES 10, dextrose 20, and mannitol 15, pH 7.2) for 3 min. Atria and ventricles were separated and myocytes were isolated by gentle mechanical trituration in KB. Myocytes were allowed to recover in KB (1 h, room temperature). The supernatant was exchanged for nominally Ca-free Tyrode's supplemented with (in mM): 2,3-butanedione monoxime (BDM) 5, L-carnitine 2, taurine 5, glutamate 2, fat-free BSA (0.1%), and penicillin/streptomycin. After recovery for 1 h at room temperature, the supernatant was exchanged for Ca-containing (1.8 mM) medium 199 supplemented with (in mM) L-carnitine 5, taurine 5, creatine 5, cytochalasin-D 0.0002, fat-free BSA (0.2%), fetal calf serum (5%), and penicillin/streptomycin. After recovery at 36°C for 1–2 h, myocytes were plated on mouse laminin–coated dishes or coverslips for experiments.

To activate PKA, myocytes were exposed to 8-(4-chlorophenylthio) adenosine 3′, 5′-cyclic monophosphate (8CPT-cAMP, 100 µM) and okadaic acid (100 nM) for specified durations (noted in figure legends or related text). Myocytes cultured under the control conditions for the same durations served as controls.

For adenovirus (Adv) transduction, myocytes were incubated with Adv-GFP-Nav1.5 and/or Adv-EB1-mRFP ($3–4 \times 10^6$ PFU/ml) for 12 h. The medium was exchanged for virus-free medium 199 (with supplements as described above) and the culture continued for specified durations. To examine the effects of PKA activation on GFP-$Na_V1.5$ and/or EB1-mRFP expressed in myocytes, 8CPT-cAMP (100 µM) and okadaic acid (100 nM) were added after the first 12 h of incubation with adenovirus(es).

Adenovirus-transduced myocytes cultured under the control conditions for the same durations served as controls.

## Cell-attached patch clamp recording and data analysis

Myocytes were superfused with nominally Ca-free Tyrode's (supplemented with 2 mM $MgCl_2$) at room temperature. The pipette was filled with normal Tyrode's. The tip resistance was kept approximately constant (1.3–1.5 MΩ) to minimize variations in $I_{Na}$ amplitude due to differences in tip size. The pipette tip was positioned at the cell center or close to the cell end (less than three sarcomeres from the cell end).

Patch clamp experiments were controlled by Clampex of pClamp 10 via Digidata 1440A using Axopatch 200B (Molecular Devices). The pipette voltage was stepped from a holding voltage of +120 mV to test voltages from +100 to –75 mV in –5 mV increments for 200 ms, once every 3 s. With this protocol, the patch membrane was held at 120 mV, hyperpolarized to the myocyte's resting membrane potential (RMP), to test voltages from 100 mV hyperpolarized to RMP to 75 mV depolarized to RMP. We verified that this holding voltage totally removed the inactivation of Na channels in the membrane patch (no further increase in maximal peak $I_{Na}$ when the holding voltage was shifted to +140 mV, i.e., 140 mV hyperpolarized to RMP). The range of test voltages allowed us to observe stochastic Nav channel openings at threshold depolarization and an increase in $I_{Na}$ amplitude as the patch membrane was further depolarized. $I_{Na}$ reached a maximal peak amplitude and then decreased as the patch membrane voltage approached the Na channel reversal potential. In one experiment, the myocyte was bathed in a high [K] solution (in mM: KCl 140, $CaCl_2$ 1.8, $NaH_2PO_4$ 0.33, and HEPES 5, pH 7.4 with KOH [Lin et al., 2011]) to zero the RMP. Under these conditions, the maximal peak Na current occurred at –35 mV, consistent with data in the literature (Lin et al., 2011). The voltage clamp protocol was repeated 4–12 times (more repetitions if $I_{Na}$ amplitude was small). Currents were leak and capacitive transient subtracted online, low-pass filtered at 10 kHz, and digitized for offline analysis.

To minimize the confounding factor of dissipating PKA effects on Nav channel distribution, we limited the patch clamp recording time to 1 h after myocytes were removed from the incubator and placed in a cell chamber on the microscope stage superfused with room temperature Tyrode's solution. On average, recordings from approximately five patches could be completed within this 1-h time limit. Patch clamp recording began on CON myocytes roughly 4 h after myocyte isolation, and recording alternated between CON and PKA myocytes over 10–12 h of total patch clamp time.

Patch clamp data analysis was done using Clampfit of pClamp 10. Families of current traces elicited by the same protocol were averaged to produce ensemble currents from a patch. We report the amplitude of maximal peak $I_{Na}$ and its time constant of inactivation from each patch.

## Immunostaining, high-resolution Airyscan imaging, and image analysis

EB1 and unpolymerized tubulins are largely cytosolic. To catch these cytosolic proteins for immunostaining, after fixation (4%

Table 1.  **Antibodies used in the experiments**

| Target | Source | Catalog # | Host | Knockout validated | Application/fold dilution |
|---|---|---|---|---|---|
| $Na_V1.5$ | Alomone | asc005 | Rabbit | Yes | Immunostaining/100× Immunoblot/500× |
| EB1 | Abcam | ab53358 | Rat | No | Immunostaining/100× Immunoblot/500× Immunoprecipitation/100× |
| α-tubulin | Abcam | ab6161 | Rat | No | Immunoblot/500× |
| Detyrosinated α-tubulin | Sigma-Aldrich | AB3201 | Rabbit | No | Immunostaining/100× Immunoblot/500× |
| β-tubulin | Santa Cruz Biotech | sc-5274 | Mouse | No | Immunostaining/100× Immunoblot/100× |
| GFP | Abcam | ab290 | Rabbit | No | Immunostaining/500× Immunoblot/500× Immunoprecipitation/100× |
| GFP | Santa Cruz Biotech | sc-9996 | Mouse | No | Immunostaining/100× Immunoblot/100× |
| GFP | Abcam | ab5450 | Goat | No | Immunostaining/500× Immunoblot/500× |
| nCadherin | BioTechne | AF6426 | Sheep | No | Immunostaining/100× |
| DsRed | Clontech | 632496 | Rabbit | No | Immunostaining/100× |
| mCherry | Thermo Fisher Scientific | PA5-34974 | Rabbit | No | Immunostaining/500× Immunoblot/500× Immunoprecipitation/100× |
| CREB1 | Santa Cruz Biotech | sc-240 | Mouse | No | Immunostaining/100× Immunoblot/100× |
| CREB1-S133[P] | Santa Cruz Biotech | sc-81486 | Mouse | No | Immunostaining/100× Immunoblot/100× |
| Na/K pump α-subunit | Abcam | ab7671 | Mouse | No | Immunoblot/500× |

paraformaldehyde in phosphate buffer saline [PBS], room temperature, 30 min), myocytes were incubated with primary antibodies in saponin blocking buffer (PBS with 0.1% saponin, 5% fetal calf serum, and 0.02% sodium azide) at room temperature for 24 h. During this long incubation time, saponin selectively removed cholesterol from the plasma membrane, creating "holes" to allow antibody entry. After removing primary antibodies, myocytes were incubated with fluorophore-conjugated secondary antibodies for imaging. Information about antibodies is listed in Table 1.

Fixed myocytes were imaged with Zeiss 880 in the Airyscan mode using 40× objective (1.4 numerical aperture [NA]). Relative to confocal imaging, Airyscan improved spatial resolution 1.7-fold in both lateral (X and Y) and axial (Z) directions. With 488-nm excitation, 1.4 NA objective, and 1.51 refractive index, the lateral and axial resolutions were ~100 and 300 nm, respectively. In whole-myocyte imaging mode, the X and Y pixels were set at 100 nm and the Z steps were 300 nm. For cluster analysis, the X and Y pixels were set to 50 nm and the Z steps were 50 nm. Images were analyzed by ImageJ (Schneider et al., 2012). Details of image analysis are presented in figure legends or the main text. 3-D images were created with Imaris (v. 10).

**In situ proximity ligation assay (PLA)**
Myocytes were incubated with Adv-GFP-$Na_V1.5$ and Adv-EB1-mRFP for 12 h and, and after removing viruses, cultured for another 12 or 36 h under the control conditions or in the presence of 8CPT-cAMP (100 μM) and okadaic acid (100 nM). Myocytes were fixed as described above and incubated with primary antibodies (GFP goat Ab targeting GFP-Nav1.5, and mCherry rabbit Ab targeting EB1-mRFP) in saponin-blocking buffer for 24 h at room temperature. After removing the primary antibodies, myocytes were subject to probe annealing,

ligation, and amplification using Duolink fluorescence assay kit (Sigma Millipore), following the manufacturer's instructions. During the amplification reaction, far-red fluorophore was incorporated into locations where the probes binding to primary antibodies at GFP-$Na_V1.5$ and EB1-mRFP were 30–40 nm apart (Söderberg et al., 2008). After the PLA procedures, myocytes were incubated with Alexa488 anti-goat and Alexa568 anti-rabbit, allowing imaging of GFP-$Na_V1.5$, EB1-mRFP, and PLA in the same myocytes.

Myocytes were imaged with Zeiss 880 in the Airyscan mode as described above. Images were analyzed with ImageJ. PLA signals were quantified as described previously (Jiang et al., 2019). 3-D Z stacks were collapsed into 2-D images by Z-projection of maxima. The region of interest (ROI) was determined by cellular contour. Signals within ROI were segmented to define PLA puncta, and PLA signal density was calculated as the percentage of cellular area occupied by PLA puncta.

**Fluorescence recovery after photobleaching (FRAP)**
Myocytes were plated on mouse laminin-coated 35-mm imaging dishes (no. 15 cover glass bottom; MatTek). Myocytes were transduced with Adv-GFP-$Na_V1.5$ and Adv-EB1-mRFP for 12 h and, after removing viruses, cultured for another 20–28 h under the control conditions or in the presence of 8CPT-cAMP (100 μM) and okadaic acid (100 nM) before experiments. During FRAP experiments, myocytes were bathed in normal Tyrode's solution supplemented with 5% fetal calf serum and 100 μM blebbistatin at 36°C. Images were acquired with Zeiss 880 in the confocal mode. Myocytes were imaged with 40× objective at a single Z plane. The X and Y pixels were set at 210 nm. For each myocyte, four ROIs of 100–150 μm² area were selected: ROI 1 at the cell center/perinuclear zone, ROI 2 at the cell end, ROI 3 at

the cell center remote from ROI 1, and ROI 4 in the cell-free area. GFP-Na$_V$1.5 and EB1-mRFP were imaged consecutively (GFP with 488-nm laser and band-pass filter 491–562 nm, mRFP with 561-nm laser and BP filter 571–696 nm) at low laser power (1–2%) once every 0.5 s. After 25 scans to set the baseline, 20 pulses of 488 and 561 lasers at 100% power were applied to ROIs 1 and 2, bleaching fluorescence in the two ROIs to <30%. Afterward, the low-power laser scans resumed for another 275 times to monitor FRAP. After correcting background bleach, based on fluorescence decline in ROI 3, the fluorescence intensities in ROIs 1 and 2 were normalized to between 1 (scan 25, right before photobleaching) and 0 (scan 26, the first scan after photobleaching). The FRAP time courses between scans 26 and 300 were fit with a two-exponential function using Clampfit. To avoid interference due to myocyte deterioration, each dish was imaged for 1–2 h.

## Subcellular fractionation and immunoprecipitation experiments in myocytes

The protocol of fractionating myocytes into cytosolic and polymerized microtubules/membrane fractions was modified from Scarborough et al. (2021). Myocytes were gently pelleted (500 × $g$, 3 min) and resuspended in a cytosolic buffer: glycerol 50%, DMSO 5%, Na$_3$PO$_4$ 10 mM, MgCl$_2$ 0.5 mM, pH 6.95, fresh GTP 0.5 mM, protease inhibitor cocktail, and phenylmethylsulfonyl fluoride (PMSF), 2 mM. Myocytes were lysed by four cycles of liquid nitrogen freezing/thawing. After centrifugation at 20,000 × $g$ for 45 min, the supernatant was collected as a "cytosolic" fraction. To improve the immunoblot quality of polymerized microtubules/membrane fraction, we found it necessary to disrupt membrane/actomyosin interactions using the protocol described previously (Barry et al., 1995): pellet from the above procedure was resuspended in TE buffer containing 0.6 M KI, incubated on ice for 15 min, and pelleted with ultracentrifugation at 43,000 rpm for 15 min. To remove KI, the pellet was resuspended in TE buffer and ultracentrifuged at 43,000 rpm, 15 min. This step was repeated two more times. The final pellet was extracted with RIPA buffer supplemented with 2% sodium dodecyl sulfate (SDS) with protease inhibitors and PMSF, for 1 h at room temperature. The suspension was centrifuged 20,000 × $g$ for 30 min. The supernatant was collected as an "SDS extracted" fraction.

For immunoprecipitation experiments, myocytes transduced with Adv-GFP-Na$_V$1.5 and/or Adv-EB1-mRFP were cultured for 36–48 h. Myocytes were incubated in lysis buffer (in mM: NaCl 145, MgCl$_2$ 0.1, EGTA 10, and HEPES 15, pH 7.4) supplemented with 1% Triton, protease inhibitor cocktail, and PMSF 2 mM, on ice with occasional vortex for 1 h before centrifugation 20,000 × $g$ for 30 min. The supernatant was collected as whole-cell lysates (WCLs). Protein concentrations in WCLs were measured using Micro BCA protein assay kit, and the protein concentrations were adjusted to the same across all samples. The same amount of WCL across all samples, in terms of µg of proteins, was incubated with BSA-blocked protein A/G magnetic beads and GFP rabbit Ab (targeting GFP-Na$_V$1.5) or mCherry rabbit Ab (targeting EB1-mRFP) at 4°C overnight. After collecting the supernatant, beads were washed with PBS, three times, and proteins were eluted into Tris-glycine SDS 2× sample buffer supplemented with β-mercaptoethanol (20% vol/vol) at 50°C for 30 min.

## HEK293 cell culture, transfection, biotinylation, and immunoprecipitation experiments

HEK293 cells were maintained in DMEM supplemented with 10% fetal calf serum and penicillin/streptomycin in a moist chamber at 36°C with 5% CO$_2$. The day before transfection, cells were split and plated on matrigel-coated dishes. Cells were incubated with plasmid and Lipofectamine 2000 for 5 h at 36°C. The medium was replaced with fresh, cDNA-free medium, and culture continued for another 24 h before experiments.

To test the effect of PKA activation on the GFP-Na$_V$1.5 protein level of the cell surface, cells were incubated with 8CPT-cAMP (100 µM) and okadaic acid (100 nM) for 4–6 h (noted in figure legends). In some experiments, chloroquine (100 µM) was added to the medium 2 h before 8CPT-cAMP and okadaic acid.

To label cell surface proteins with biotin, cells were gently rinsed with ice-cold Ca- and Mg-containing PBS followed by incubation with lysine-reactive, disulfide-breakable, membrane-impermeable biotin derivative (EZ-link sulfo-NHS-SS-biotin, 0.25 mg/ml) in cold for 30 min. The biotinylation reaction was quenched by adding a threefold volume of 100 mM glycine in PBS, in cold for 15 min. Cells were pelleted and lysed in lysis buffer supplemented with 1% Triton, protease inhibitor cocktail, and PMSF 2 mM on ice for 1 h. After centrifugation at 20,000 × $g$, 15 min, the supernatant was collected as WCL. Protein concentrations in WCLs were measured as described above and adjusted to the same across all samples. The same amount of WCL across all samples was used to isolate biotinylated (cell surface) proteins by incubation with neutravidin agarose beads at 4°C overnight. Beads were pelleted, washed (PBS, three times), and the bound proteins were eluted with Tris-glycine SDS 2× sample buffer supplemented with β-mercaptoethanol (20% vol/vol) at 50°C for 30 min.

To test coimmunoprecipitation among GFP-Na$_V$1.5, native EB1, and β-tubulin, HEK293 cells transfected with GFP-Na$_V$1.5 were lysed in 1% Triton lysis buffer as described above and incubated with BSA-blocked protein A/G magnetic beads and GFP rabbit Ab or EB1 rat Ab. WCL incubated with protein A/B beads without Abs served as the negative control.

## SDS-polyacrylamide gel electrophoresis (SDS-PAGE), immunoblot, and data analysis

SDS-PAGE and immunoblot experiments were used to quantify the effects of PKA activation on (a) protein levels in the cytosolic and polymerized microtubules/membrane fractions of myocytes, (b) degree of protein coimmunoprecipitation, and (c) protein level on the cell surface (biotinylated fraction). To allow quantification of immunoblot results, the amounts of proteins used during experimental procedures and in SDS-PAGE loading were carefully controlled. In (a), protein loading was checked by Coomassie blue (CB) staining of the gels after blotting and, if necessary, CB stain was used to adjust immunoblot band intensities. In (b) and (c), protein concentrations in WCLs were adjusted to the same across all samples in the same experiment, and the same amount of WCLs was used for immunoprecipitation or neutravidin pulldown (of biotinylated fraction). Sample loading for SDS-PAGE was controlled so that the ratio of WCL to "WCL equivalent used in immunoprecipitation or neutravidin pulldown" was kept at 1:20.

SDS-PAGE and immunoblots were described previously (Zankov et al., 2019; Wilson et al., 2021). Briefly, protein samples were fractionated by SDS-PAGE and transferred to polyvinylidene fluoride (PVDF) membrane. The membrane was blocked with 1× TBS-Tween 20 with 10% fat-free dry milk (room temperature, >1 h) and incubated with primary Abs diluted in 1× TBS-Tween 20 with 5% BSA for >2 h at room temperature or overnight at 4°C. After removing primary Ab, the membrane was incubated with horseradish peroxidase (HRP)-conjugated secondary Ab at 10,000× dilution for 1 h at room temperature. The membrane was washed with 1× TBS-Tween 20, three times, 10 min each. Immunoreactive bands were imaged with an enhanced chemiluminescence kit using FluorChem M (ProteinSimple). Band intensities were quantified using AlphaView SA (ProteinSimple) and numerical data were exported to Excel files for processing.

To minimize non-biological variations among experiments, immunoblot data were analyzed in the following manner. For (a), the "PKA/CON" ratio of band intensities in the same experiment was used to evaluate the effect of PKA activation on protein level in cytosolic or SDS extracted fraction. For (b), the "[+] IP/WCL" ratio of band intensities in the same experiment represented the degree of coimmunoprecipitation, and the PKA/CON ratio of the above was used to evaluate the effect of PKA activation on protein coimmunoprecipitation. For (c), the "biotinylated fraction/WCL" ratio of band intensities in the same experiment represented the degree of protein biotinylation, and the PKA/CON ratio of the above was used to evaluate the effect of PKA activation on protein expression on the cell surface.

### Statistical analysis

Statistical analysis of multiple groups was done by one-way ANOVA (v. 4.0; SigmaStat). Analysis of the two groups was done by $t$ test (Excel function). The P values are noted in the figures. Data summary is presented as a bar graph (mean with SE bar), superimposed with individual data points, or as a box plot, showing median (central line), 25% and 75% values (bottom and top of box), 10% and 90% values (lower and upper whiskers), and outliers (symbols), using SigmaPlot (v. 14).

### Online supplemental material

We used wheat germ agglutinin (WGA, a lectin that binds glycosylated protein) as the plasma membrane marker to verify differential distribution patterns of $Na_V1.5$ inside myocytes and on myocyte surface (Fig. S1). We conducted two separate experiments (RNA-seq and RT-qPCR) on the total RNA samples prepared from CON and PKA myocytes (five independent experiments) and showed that the EB1 transcript level was increased while the $Na_V1.5$ transcript level was decreased in PKA myocytes relative to CON myocytes (Fig. S2). Fig. S3 shows that the immunofluorescence signals of α-tubulin and β-tubulin (α- and β-Tub, respectively) were totally overlapped in both CON and PKA myocytes, validating β-Tub as a microtubule marker in confocal experiments. Fig. S4 presents data validating GFP-$Na_V1.5$ and EB1-mRFP as surrogates of native counterparts in rat ventricular myocytes. Fig. S5 contrasts the effects of acute PKA activation (by β-adrenergic stimulation with isoproterenol,

100 nM, for 15 min) and chronic PKA activation (by 8CPT-cMP and okadaic acid, for 12 h) on $Na_V1.5$ distribution in rat ventricular myocytes. Videos 1 and 2 are rotating 3-D views of a CON and a PKA myocyte, respectively, illustrating the distribution patterns of $Na_V1.5$, microtubules, and EB1 at, and close to, the ICDs at myocytes' ends.

## Results

Unless otherwise stated, all experiments were conducted on ventricular myocytes from young adult (3–5 mo) male Sprague–Dawley rats, and we focused on the native $Na_V1.5$ channel. To induce persistent activation of cAMP-dependent protein kinase (PKA), myocytes were incubated with 8CPT-cAMP (membrane-permeable cAMP analog, 100 μM) and okadaic acid (phosphatase inhibitor, 100 nM) for 0.5–40 h (specific incubation duration will be specified in figure legends or text) before experiments. In addition to PKA, there is another group of cAMP effectors ubiquitously expressed in eukaryotic cells: exchange protein activated by cAMP (EPAC1 and EPAC2; Laudette et al., 2018). It has been shown that 8CPT-cAMP preferentially activates PKA but not EPACs (Krishnan et al., 2012). Including okadaic acid in the incubation (inhibiting phosphatases, PP1 and PP2A, with $IC_{50}$ 150 nM and 30 pM, respectively) further enhanced PKA's effects.

PKA can phosphorylate serine 133 of cAMP response element binding protein 1 (CREB1). Phosphorylated CREB1 (CREB1-S133[P]) then enters the nuclei and regulates gene expression (Zhang et al., 2005; Iourgenko et al., 2003). To confirm PKA activation in 8CPT-cAMP/okadaic acid-treated myocytes, we checked whether CREB1-S133[P] could be detected. Immunoblots consistently detected a CREB1-S133[P]-specific band in myocytes pretreated with 8CPT-cAMP/okadaic acid but not in control myocytes, while the total CREB1 protein level (detected by a CREB1 antibody) remained the same (Fig. 1 A, left). Furthermore, CREB1-133[P] immunofluorescence appeared inside nuclei of myocytes pretreated with 8CPT-cAMP/okadaic acid but not in control myocytes (Fig. 1 A, right). In the following text, we refer to "8CPT-cAMP/okadaic acid incubation" as "PKA activation." Myocytes pretreated with 8CPT-cAMP/okadaic acid will be called "PKA myocytes" versus "CON myocytes" that were isolated from the same heart and incubated for the same duration under the control conditions.

### Persistent PKA activation increased Na⁺ current amplitude without increasing total $Na_V1.5$ protein in cardiac myocytes

We used a cell-attached patch clamp to record Na⁺ channel current ($I_{Na}$) from the top surface of CON and PKA myocytes, at the cell center and close to the cell end (within 3 sarcomeres or ≤6 μm; Fig. 1 B). This choice of $I_{Na}$ recording configuration was based on two factors. First, the cell-attached patch clamp ensures high-fidelity control of membrane voltage under investigation, a prerequisite for quantifying $I_{Na}$ amplitude and kinetics. Second and more importantly, a previous report showed subcellular variations in $I_{Na}$ amplitude and gating properties in adult rat ventricular myocytes (Lin et al., 2011). Our data showed that the maximal peak $I_{Na}$ amplitudes were not different

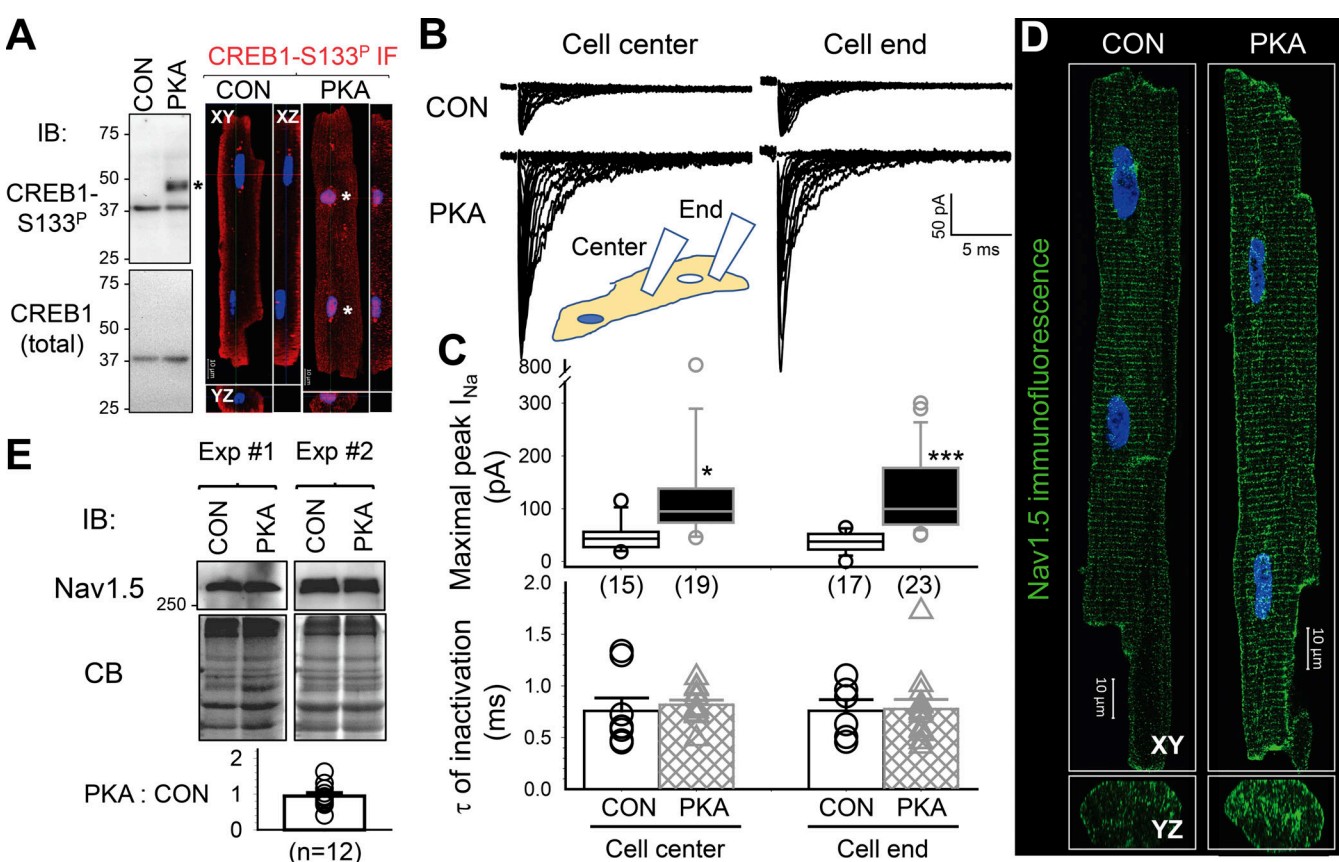

Figure 1. **Persistent PKA activation increased Na current amplitudes without increasing the total Na$_V$1.5 protein level in ventricular myocytes.** PKA myocytes had been incubated with 8CPT-cAMP (100 μM)/okadaic acid (100 nM) for 6–15 h before experiments. CON myocytes had been cultured for the same duration under the control conditions. **(A)** PKA activation is confirmed by the appearance of cAMP response element binding protein 1 with serine at position 133 phosphorylated (CREB1-S133$^P$) and its nuclear entry. Left upper: Immunoblot (IB) images of WCLs from CON and PKA myocytes probed with antibodies specific for CREB1-S133$^P$ (top) and CREB1 (bottom). Left lower: PKA to CON ratio of CREB1-S133$^P$ band intensities (5.46 ± 1.04, dotted line denotes value of 1). Right: CREB1-S133$^P$ immunofluorescence (red, nuclei stained blue) in CON and PKA myocytes. Images are presented in orthogonal view (XY, YZ, and XZ planes) to show that CREB1-S133$^P$ was within, instead of around, nuclei of the PKA myocyte. *: CREB1-S133$^P$ specific band in IB and CREB1-S133$^P$ signals within nuclei. **(B)** Top: Na current (I$_{Na}$) traces recorded using a cell-attached patch clamp with a pipette tip positioned on the top surface of the cell center or close to the cell end (within three sarcomeres) of CON and PKA myocytes. These exceptional traces were obtained on the same day (CON myocytes 5–6 pm, PKA myocytes 11 PM–midnight). During patch clamp recording, myocytes were superfused with nominally Ca-free (supplemented with 2 mM Mg) Tyrode's without 8CPT-cAMP/okadaic acid. The pipette was filled with Ca-containing (2 mM) Tyrode's solution. The tip resistance was (in MΩ): CON cell end 1.44 ± 0.06, CON cell center 1.41 ± 0.06, PKA cell end 1.33 ± 0.05, and PKA cell center 1.33 ± 0.06 (one-way ANOVA, P = 0.267). **(C)** Top: Bar graphs (mean ± SE) and individual data points of maximal peak I$_{Na}$ amplitudes. Bottom: Time constants (τ) of inactivation of maximal I$_{Na}$. Data were pooled from five independent experiments. The numbers of myocytes studied are shown in parentheses. **(D)** Airyscan images of Na$_V$1.5 immunofluorescence from CON and PKA myocytes in XY and YZ planes. **(E)** Top: Na$_V$1.5 immunoblot images of SDS extracts of CON and PKA myocytes in two independent experiments. Middle: Coomassie blue (CB) stain of the same gels to confirm even loading. Bottom: Average PKA:CON ratio of Na$_V$1.5 band intensities (0.95 ± 0.09), not different from 1 (dotted line). Information on the antibodies used in experiments shown in this and the following figures is listed in Table 1. The listed P values are from t tests against null hypothesis (A and E), or CON versus PKA myocytes (C). Source data are available for this figure: SourceData F1.

between the cell center and the cell end in CON myocytes. PKA activation for 6–15 h markedly increased I$_{Na}$ amplitudes at both locations, without changing the time constant of inactivation (Fig. 1 C). Indeed, 3-D Airyscan imaging confirmed an increase in Na$_V$1.5 immunofluorescence on the peripheral surface of PKA relative to CON myocytes (Fig. 1 D).

Previous reports have shown that increasing the cluster size of ion channels on the cell surface can enhance channels' open probability through a proposed mechanism of "positive cooperativity" in channel opening within a cluster (Dixon et al., 2022). We investigated how persistent PKA activation may affect Na$_V$1.5 clusters on myocyte surface. When imaged at

Z planes inside myocytes, Na$_V$1.5 was distributed in regular striations and on the lateral surface (Fig. 1 D). However, when imaged at the myocyte surface, Na$_V$1.5 appeared as random clusters (Fig. 2 A). This was verified by simultaneous imaging of WGA, a plasma membrane marker (Fig. S1). Fig. 2 B contrasts surface Na$_V$1.5 clusters between CON and PKA myocytes. Na$_V$1.5 cluster analysis is summarized in Fig. 2 C. Persistent PKA activation not only increased the Na$_V$1.5 cluster density but also increased Na$_V$1.5 cluster size without altering mean immunofluorescence intensity in clusters. The increase in Na$_V$1.5 cluster size is expected to contribute to the higher I$_{Na}$ amplitudes in PKA myocytes (Fig. 1, B and C).

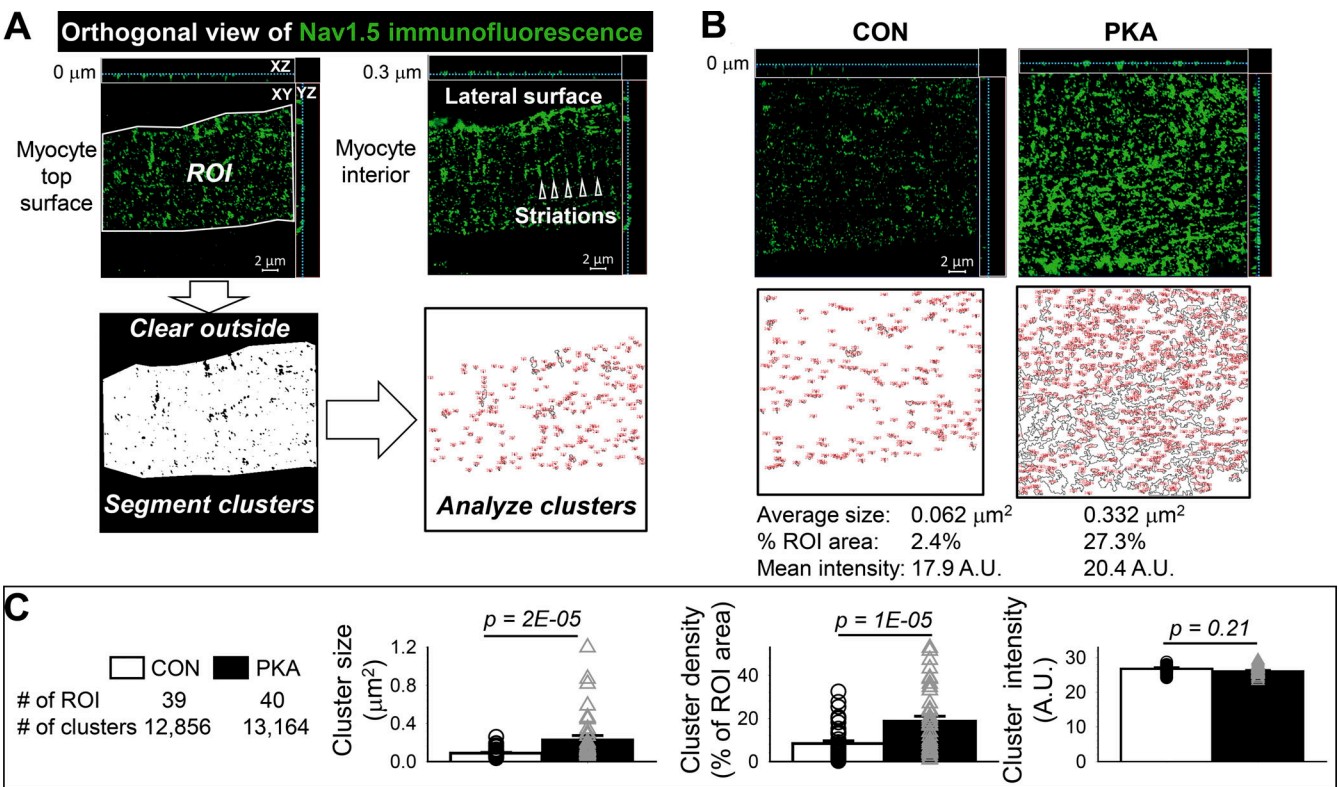

Figure 2. **Persistent PKA activation for 15 h increased the size and density of Na$_V$1.5 clusters on ventricular myocytes' surface. (A)** Procedures of detecting and analyzing Na$_V$1.5 clusters on the myocyte surface. Airyscan images of Na$_V$1.5 immunofluorescence were acquired with X and Y pixel dimensions set at 50 nm. The Z plane was advancing in 50 nm steps from intracellular to extracellular space across the myocyte surface. Top: Two orthogonal views of Na$_V$1.5 in the same area of a myocyte, illustrating how Na$_V$1.5 distribution pattern varied depending on the Z position (noted by the cyan dotted lines in XY and YZ views). The one on the right was at a Z plane 0.3 µm beneath the surface. Na$_V$1.5 was in striations (open triangles) and on lateral surfaces, typical of Na$_V$1.5 distribution in myocyte interior. The one on the left shows random Na$_V$1.5 clusters, representing its distribution on myocyte surface at a Z plane 0 µm. The XY plane image of the latter was exported to ImageJ and analyzed in the following steps: (a) demarcating ROI, (b) clearing signals outside ROI, (c) segmenting signals inside ROI to define Na$_V$1.5 clusters, and (d) calculating cluster parameters: average size (µm²), density (% of ROI area occupied by clusters), and mean immunofluorescence intensity in clusters. **(B)** Examples of Na$_V$1.5 clusters on the surface of CON and PKA myocytes (Z plane at 0 µm). Top: Orthogonal views. Middle: Na$_V$1.5 clusters outlined. Bottom: Cluster parameters. **(C)** Bar graphs (mean ± SE) and individual data points of cluster parameters from CON and PKA myocytes. Date are pooled from two independent experiments with number of ROIs analyzed and number of clusters detected listed on the left.

To check whether the increase in $I_{Na}$ amplitude was due to an upregulation of Na$_V$1.5 by persistent PKA activation, we quantified Na$_V$1.5 protein level in SDS extracted fraction of CON and PKA myocytes (the rationale for myocyte fractionation will be discussed below). Immunoblot experiments show that persistent PKA activation (12–15 h) did not increase the total Na$_V$1.5 protein level (Fig. 1 E). This suggests that the increase in $I_{Na}$ amplitude in PKA myocytes may be due to a redistribution of Na$_V$1.5 to myocyte's surface.

**Persistent PKA activation upregulated EB1 and induced microtubule reorganization in ventricular myocytes**

Na$_V$1.5 channels traffic on microtubules, targeted to and anchored at their destinations by protein–protein interactions (Chen-Izu et al., 2015; Rook et al., 2012). The microtubule end-binding protein 1 (EB1) is a major microtubule regulator in cardiac myocytes (Shaw et al., 2007; Drum et al., 2016; Marchal et al., 2021). Furthermore, it has been suggested that EB1 in conjunction with CLIP-associating protein 2 (CLASP2) is involved in targeting Na$_V$1.5 to ICDs (Marchal et al., 2021). Importantly, the promoter regions of EB1 genes in humans, mice,

and rats contain cAMP response elements (Zhang et al., 2005), suggesting an upregulation of EB1 by CREB1-S133$^P$ in PKA myocytes. Indeed, the EB1 transcript was upregulated, while Na$_V$1.5 transcript was downregulated, in PKA versus CON myocytes (Fig. S2). To test the duration of PKA activation required for EB1 protein upregulation, we incubated myocytes with 8CPT-cAMP/ okadaic acid for varying durations (0.5–40 h) and used pixel contents of EB1 immunofluorescence as a surrogate for EB1 protein expression. For comparison, the Na$_V$1.5 immunofluorescence in the same groups of myocytes was also analyzed. Fig. 3 A depicts representative images of EB1 and Na$_V$1.5 immunofluorescence in CON and PKA myocytes after different incubation times. The time courses of their pixel contents are plotted in Fig. 3 B. While the EB1 pixel contents were stable in CON myocytes between 0.5 and 40 h, there was a time-dependent increase in EB1 pixel content in PKA myocytes: it rose above the control level at 4 h, reaching a peak at 14 h and remained elevated at 40 h. On the other hand, the Na$_V$1.5 pixel contents were not increased in PKA myocytes, consistent with the immunoblot data (Fig. 1 E).

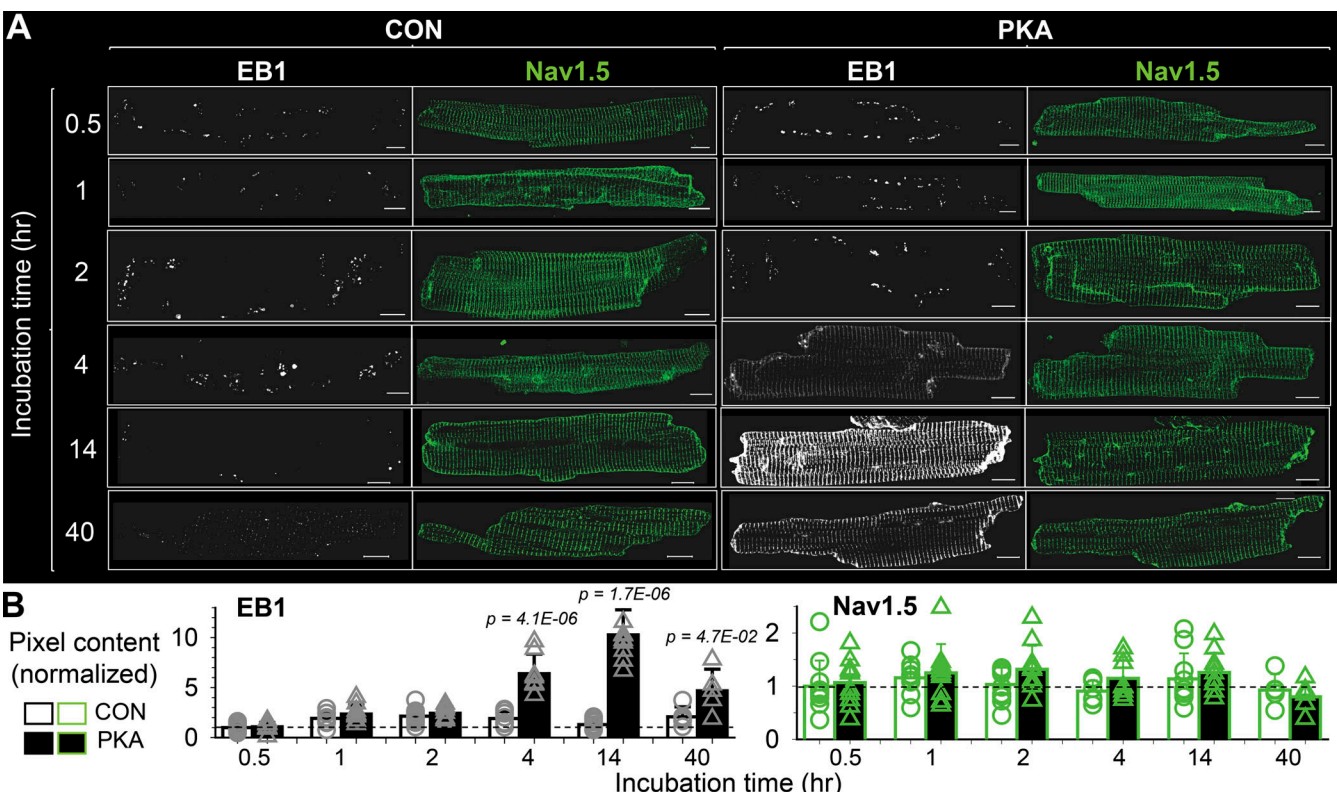

Figure 3. **Time-dependent increase in EB1 during 8CPT-cAMP/okadaic acid incubation in ventricular myocytes. (A)** EB1 and Na$_V$1.5 immunofluorescence signals in CON and PKA myocytes. The incubation times with 8CPT-cAMP/okadaic acid are listed on left. **(B)** Pixel contents of EB1 and Na$_V$1.5 immunofluorescence in CON and PKA myocytes at different incubation times listed along the abscissa. Data were normalized by the mean value of CON myocytes at the 0.5 h time point (dashed lines). Listed P values are from the $t$ test of PKA versus CON. All scale bars are 10 μm.

We then directly quantified the EB1 protein level in the cytosolic fraction of CON and PKA myocytes. Incubation with 8CPT-cAMP/okadaic acid lasted for 15 h, corresponding to the peak increase of EB1 pixel content in image data. Fig. 4 A, top left, depicts EB1 immunoblot images from two experiments. Densitometry quantification of 10 immunoblots is summarized in Fig. 4 A, top right. The EB1 band intensity in PKA samples was significantly higher than in CON samples (PKA:CON = 1.47 ± 0.18), confirming EB1 protein upregulation in myocytes by persistent PKA activation.

We next examined microtubule stability and morphology in CON and PKA myocytes. Microtubules are built by polymerization of α-/β-tubulin dimers (α-Tub and β-Tub). In stable microtubules, the last amino acid of α-Tub, tyrosine, can be enzymatically cleaved, creating detyrosinated α-Tub (deY α-Tub; Nieuwenhuis and Brummelkamp, 2019). Therefore, deY α-Tub can serve as a reporter of stable microtubules (Kerr et al., 2015). Immunoblot experiments on cytosolic fraction of CON and PKA myocytes showed that persistent PKA activation reduced the deY α-Tub fraction, without changing the total α-Tub protein level (Fig. 4 A, PKA:CON of α-Tub and deY α-Tub band intensities 1.06 ± 0.17 and 0.43 ± 0.05, respectively).

Fig. 4 B compares the distribution patterns of EB1, β-tubulin (β-Tub, reporting total microtubules), and deY α-Tub (reporting stable microtubules) in CON and PKA myocytes. Fig. S3 shows that β-Tub and α-Tub immunofluorescence signals overlapped in both CON and PKA myocytes, supporting the ability of β-Tub

to report total microtubules. In CON myocytes, EB1 formed puncta with a modest concentration at cell ends. In PKA myocytes, the puncta disappeared. Instead, EB1 was highly concentrated along the lateral surface and at cell ends.

In CON myocytes, β-Tub and deY α-Tub signals highly overlapped, representing stable interfibrillar microtubules aligned with myocytes' longitudinal axis and microtubule network surrounding nuclei (Caporizzo et al., 2019; Wang et al., 2015). In PKA myocytes, the deY α-Tub immunofluorescence was diminished, and the patterns of β-Tub and deY α-Tub became distinctly different (Fig. 4 B, images of tubulin merge). Consistent with the decrease in deY α-Tub band intensity and immunofluorescence, the stable interfibrillar microtubules were dramatically reduced (Fig. 4 C). Furthermore, in PKA, but rarely in CON myocytes, microtubules formed looped structures that were especially prominent on the lateral surface (Fig. 4 D) as well as at ICDs (see below). Microtubule looping has been reported previously: enhanced EB1 binding to the microtubule plus-ends promotes fast growth of cell cortex microtubules that push against the cell membrane and loop (Tortosa et al., 2013). This scenario is supported by the lack of deY α-Tub looping in PKA myocytes (Fig. 4 D).

**Persistent PKA activation enhanced correlative distribution among Na$_V$1.5, EB1, and β-Tub on myocyte surface and at ICDs**
The above observations suggest that persistent PKA activation may promote Na$_V$1.5 trafficking on microtubules to myocyte

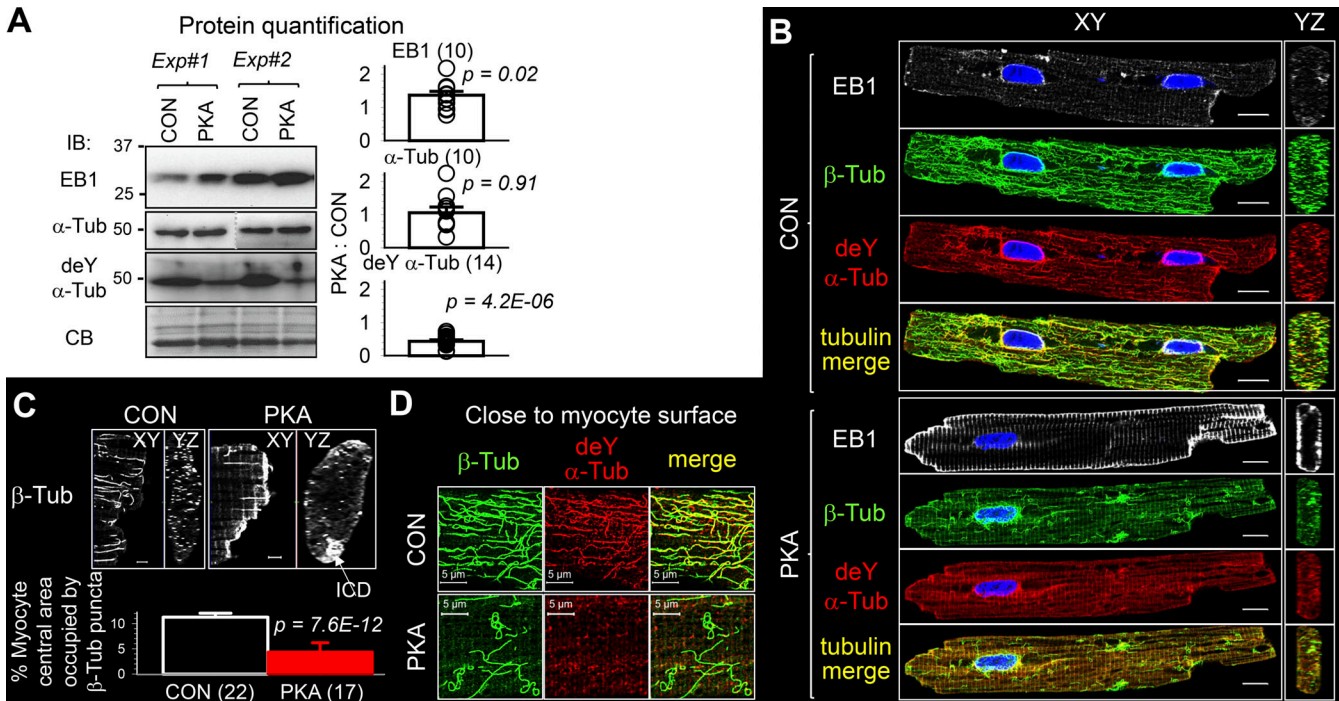

**Figure 4. Persistent PKA activation for 15 h induced microtubule reorganization in ventricular myocytes. (A)** Left: Immunoblot images of cytosolic fraction from CON and PKA myocytes in two independent experiments. Antibodies targeted EB1, total α-tubulin (α-Tub), and detyrosinated α-tubulin (deY α-Tub). CB confirms even loading. In the immunoblot image of α-Tub, and immunoblot images in the following figures, the dotted vertical line indicates lane(s) in between removed for presentation (corresponding uncropped images are shown in source data). Right: PKA:CON ratios of band intensities pooled from the number of immunoblots shown in parentheses. **(B)** XY and YZ plane images of CON and PKA myocytes immunostained for EB1, β-tubulin (β-Tub), and deY α-Tub. "Tubulin merge" is combined β-Tub and deY α-Tub signals. **(C)** Reduction of interfibrillar microtubules in PKA versus CON myocytes. Top: XY and YZ plane images of β-Tub immunofluorescence in CON and PKA myocytes. In the PKA myocyte's YZ-plane view, the β-Tub dense region was part of an intercalated disc (ICD, based on nCadherin immunostaining, not shown). Bottom: Microtubule density quantified as "% myocyte central area occupied by β-Tub puncta," where "myocyte central area" was defined as the cross-sectional area within 1 µm from cell contour. Numbers in parentheses are those of the myocytes studied. **(D)** Immunofluorescence signals of β-Tub, deY α-Tub, and their merge at a z plane close to the surface of CON and PKA myocytes. Listed P values are from t tests against null hypothesis (A), or PKA versus CON (C). Scale bars are 10 µm for B, 2 µm for C, and 5 µm for D. Source data are available for this figure: SourceData F4.

surface aided by EB1. If so, we expect to see an increase in correlation among $Na_V1.5$, β-Tub, and EB1 in PKA myocytes. To investigate whether this was the case, we quantified the distribution and correlation among the three proteins along the lateral surface and at the ICD of CON and PKA myocytes.

The procedures for quantifying immunofluorescence signals and their distribution in myocytes are summarized in Fig. 5. Fig. 6 A depicts four-color imaging of $Na_V1.5$, EB1, β-Tub, and nCadherin (nCad) in CON and PKA myocytes. nCad is the major component of adherens junction and was used to demarcate ICDs. Immunofluorescence quantification is summarized in Fig. 6 B. Consistent with immunoblot data (Fig. 1 E and Fig. 4 A), PKA activation did not increase the whole myocyte pixel content of either $Na_V1.5$ or β-Tub, but markedly increased the EB1 pixel content. $Na_V1.5$ and EB1, but not β-Tub, were more enriched along the lateral surface.

Interestingly, $Na_V1.5$, EB1, and β-Tub were all more enriched at ICDs of PKA myocytes, prompting us to investigate their distribution in ICDs. ICDs are heterogeneous 3-D structures. We used 3-D Airyscan imaging with XY pixel dimension and Z steps set at 50 nm and reconstructed YZ-plane images as en face ICD views (Fig. 7 A, top row). These en face ICD views were analyzed

by ImageJ (Fig. 7 A, middle and bottom rows). Fig. 7 B presents the summary of ICD cluster quantification. Persistent PKA activation markedly enhanced cluster density and increased cluster size of $Na_V1.5$, EB1, and β-Tub, without altering cluster signal intensity in any of them.

We used thresholded Pearson correlation coefficient (PCC, using ImageJ plug-in "Coloc2") to quantify the degrees of correlation between immunofluorescence signals of $Na_V1.5$, EB1, and β-Tub (Dunn et al., 2011). The PCC values range from +1 (total correlation) to −1 (total anti-correlation), with "0" as "no correlation." With the high spatial resolution afforded by Airyscan imaging, positive PCC values denote fluorescence signals too close to be detected as separate objects, while negative PCC values denote those that can be separated. A shift of PCC values from negative toward zero or into the positive range indicates that the distribution of two immunofluorescence signals becomes closer to each other.

Fig. 8 A depicts representative immunofluorescence images of the three proteins on myocyte surface and at ICDs of CON and PKA myocytes. The positive shift in thresholded PCC values from CON to PKA myocytes (Fig. 8 B) indicates a higher degree of correlation between $Na_V1.5$ and β-Tub, $Na_V1.5$ and EB1, and

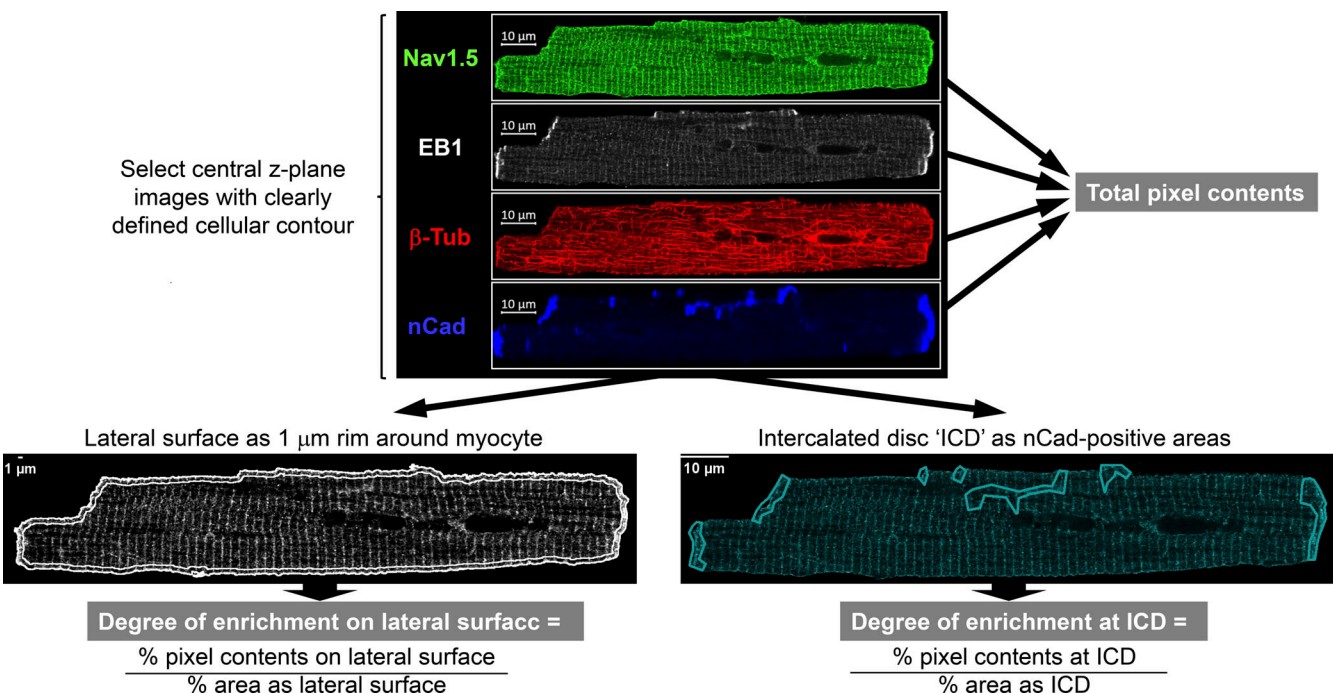

**Figure 5. Quantification of immunofluorescence (IF) intensity and distribution along the lateral surface and at ICD of ventricular myocytes immunostained for Na$_V$1.5, EB1, β-Tub, and nCadherin (nCad).** Top: Representative Airyscan images. The central z-plane image with a clearly defined cellular contour was used for quantification. Pixel contents = background-subtracted mean pixel value times cellular area. Bottom left: IF signal within an area 1 μm from cellular contour normalized by total pixel content gave "% pixel contents on the lateral surface." This value was divided by % cellular area within the 1 μm rim and defined as "degree of enrichment on lateral surface." Bottom right: IF signals within all areas demarcated by nCad IF signals were summed and normalized by total pixel content to give "% pixel contents at ICD." This value was divided by "% cellular area demarcated by nCad signals," and defined as "degree of enrichment at ICD."

β-Tub and EB1. PKA-induced changes in Na$_V$1.5, β-Tub (microtubules), and EB1 distribution and their relationships can be best appreciated by the videos of their 3-D views in the ICD region (Videos 1 and 2). In summary, persistent PKA activation not only increased all three proteins on the myocyte surface but also increased their correlation. We next seek evidence of direct interactions among Na$_V$1.5, EB1, and β-Tub and test how PKA activation influences these interactions.

**PKA activation strengthens Na$_V$1.5/EB1 and Na$_V$1.5/β-Tub interactions**
In HEK293 cells, Na$_V$1.5 (as transfected GFP-Na$_V$1.5) and the functional form of native EB1, the EB1 dimer (Chen et al., 2014), coimmunoprecipitated reciprocally, supporting direction interactions (Fig. 9 A). Furthermore, PKA activation increased EB1 dimer coimmunoprecipitation with GFP-Na$_V$1.5, suggesting a stronger interaction between the two. However, attempts to coimmunoprecipitate Na$_V$1.5 and EB1 from myocytes were not successful (Fig. 9 B). In this case, we used Adv to coexpress GFP-Na$_V$1.5 and EB1-mRFP in myocytes and immunoprecipitated them with antibodies specific for GFP or mRFP. This design was to avoid the possibility that immunoprecipitating antibodies bind to the same areas of Na$_V$1.5/EB1 interaction, hindering their coimmunoprecipitation. Fig. S4 shows that fusing GFP to the N-terminus of Na$_V$1.5 and fusing mRFP to the C-terminus of EB1 did not interfere with their distribution or function in cardiac myocytes.

Fig. 9 C provides an explanation for the different outcomes. In HEK293 cells, GFP-Na$_V$1.5 was present in a Triton-soluble fraction and could be extracted by the 1% Triton lysis buffer. This allowed coimmunoprecipitation from 1% Triton extract. In rat ventricular myocytes, native Na$_V$1.5 was largely absent in the Triton-soluble fraction but required a strong detergent (2% SDS) for extraction. This is consistent with S-palmitoylation of native Na$_V$1.5, promoting its sequestration into lipid-raft or caveolar (Triton-insoluble) domains (Pei et al., 2016; Yarbrough et al., 2002). Therefore, the 1% Triton lysis buffer used to solubilize myocyte proteins without disrupting protein–protein interactions extracted too little GFP-Na$_V$1.5, hindering its detection in EB1-mRFP immunoprecipitates.

We sought two alternative strategies to test Na$_V$1.5/EB1 interaction in cardiac myocytes. The first was to quantify the mobilities of GFP-Na$_V$1.5 and EB1-mRFP in cardiac myocytes using FRAP. Since Na$_V$1.5 is embedded in membranes while EB1 is largely freely mobile in the cytosol (Fig. 9 D), we expect to see much lower mobility of GFP-Na$_V$1.5 relative to EB1-mRFP. However, in areas where the two interact, we predict an increase in GFP-Na$_V$1.5 mobility. This was based on a previous report showing that coexpressing EB1 with GFP-Na$_V$1.5 in HEK293 cells accelerates GFP-Na$_V$1.5 FRAP in the cell periphery (Marchal et al., 2021). EB1-mRFP associated with GFP-Na$_V$1.5 may have reduced mobility. Images of FRAP experiments and data quantification are shown in Fig. 9 E, As expected, EB1-mRFP was much more mobile than GFP-Na$_V$1.5 in both the cell center

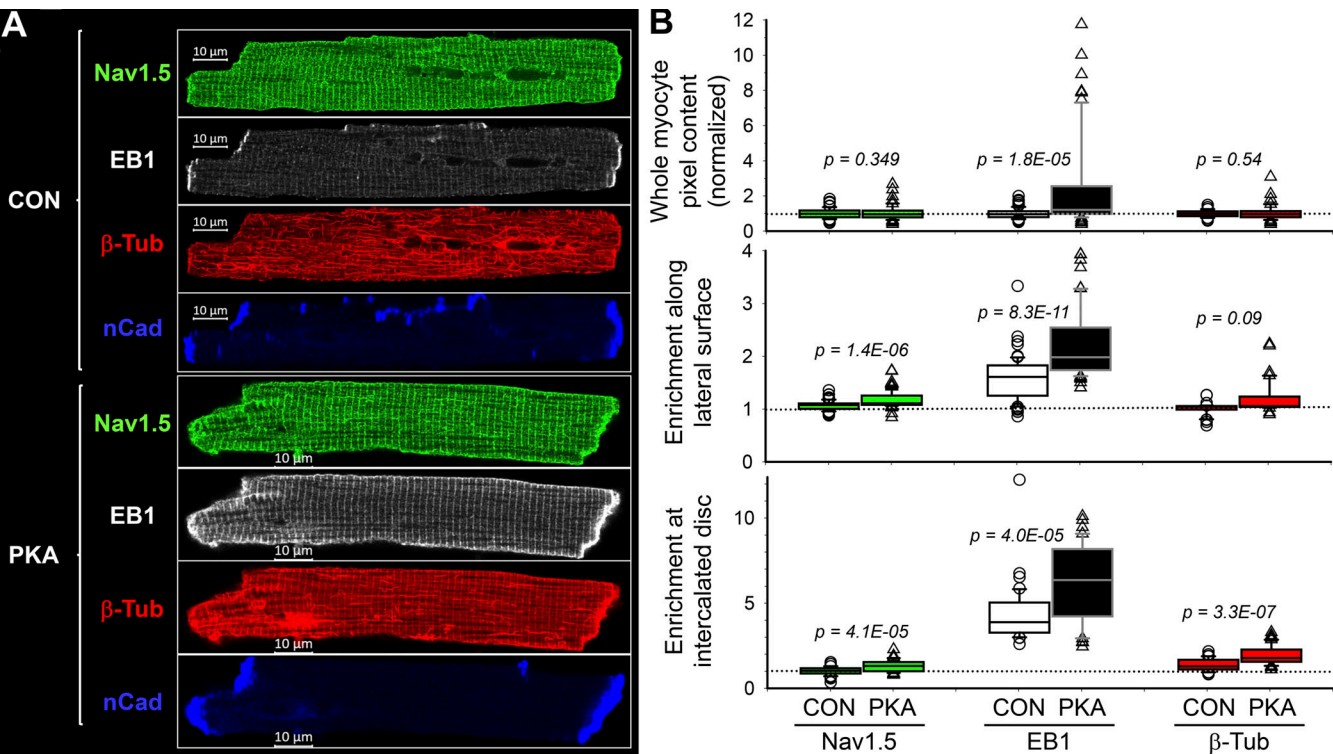

Figure 6. **Persistent PKA activation for 12 h caused an enrichment of Na$_V$1.5, EB1, and β-Tub immunofluorescence along the lateral surface and at intercalated discs of ventricular myocytes. (A)** Airyscan images of Na$_V$1.5, EB1, β-Tub, and nCad immunofluorescence in CON and PKA myocytes. **(B)** Box plots of whole myocyte pixel content (normalized to the mean values of CON myocytes), enrichment of immunofluorescence along the lateral surface, and enrichment of immunofluorescence at intercalated disc regions. Data were pooled from five independent experiments, with 40–70 myocytes per group. Dotted line indicates CON mean (top) or no enrichment (middle and bottom). Listed P values are from *t* tests of PKA versus CON myocytes.

and cell end. Importantly, GFP-Na$_V$1.5 mobility trends higher at the cell end than the cell center, while EB1-mRFP mobility was lower at the cell end than the cell center. These observations are consistent with a higher degree of interaction between GFP-Na$_V$1.5 and EB1-mRFP at cell ends where ICDs are.

The second approach was in situ PLA. In this approach, proteins are fixed in place, allowing the detection of transient interactions. We used Adv-mediated GFP-Na$_V$1.5 and EB1-mRFP expression in myocytes. By controlling the culture duration, we could test the correlation between PLA signals and protein expression as a validation of PLA specificity. Fig. 10 A shows representative images. In the myocyte cultured for 24 h, there was very little GFP-Na$_V$1.5 or EB1-mRFP expression. PLA puncta were scarce. With a longer culture time (48 h), GFP-Na$_V$1.5 and EB1-mRFP signals were much stronger and PLA puncta were much denser. PKA activation significantly increased the PLA puncta density corrected for the increase in GFP-Na$_V$1.5 and EB1-mRFP signal intensities. This indicates more frequent GFP-Na$_V$1.5/EB1-mRFP encounters (Fig. 10 B). 3-D views of PLA puncta distribution in myocytes suggest that most GFP-Na$_V$1.5/EB1-mRFP encounters occurred along the myocytes' periphery (Fig. 10 C).

β-Tub could be coimmunoprecipitated with GFP-Na$_V$1.5 in HEK293 cells (Fig. 11 A). Furthermore, the amount of β-Tub pulled down by GFP-Na$_V$1.5 trended higher after persistent PKA activation, suggesting a stronger Na$_V$1.5/β-Tub association.

On the other hand, β-Tub coimmunoprecipitation with EB1 was weakened by PKA activation, consistent with a previous report (Song et al., 2020). Putting these data together, we conclude that more frequent Na$_V$1.5/EB1 encounters in PKA myocytes (PLA experiments in Fig. 10) and an apparently stronger Na$_V$1.5/β-Tub coimmunoprecipitation in PKA-treated HEK293 cells (Fig. 11) can explain why myocyte surface Na$_V$1.5 was more correlated with EB1 and β-Tub in PKA than CON myocytes (Fig. 8). However, the higher degree of correlation between EB1 and β-Tub on the surface of PKA myocytes was not due to strong binding between the two but likely more frequent, although transient interactions.

## PKA activation promoted Na$_V$1.5 trafficking from an intracellular reservoir to the cell surface

A previous study using the *Xenopus* oocyte model showed that chloroquine prevented PKA-induced increase in cell surface Na$_V$1.5 (Zhou et al., 2000). This was attributed to an inhibition of protein recycling because chloroquine was preferentially loaded into endosomes, alkalinizing their lumen and interfering with protein sorting. To probe whether chloroquine prevented the increase in surface Na$_V$1.5 following persistent PKA activation in mammalian cells, we performed biotinylation experiments to quantify cell surface Na$_V$1.5. A pilot experiment on cardiac myocytes was not successful due to the high percentage of leaky (dead or dying) myocytes interfering with data

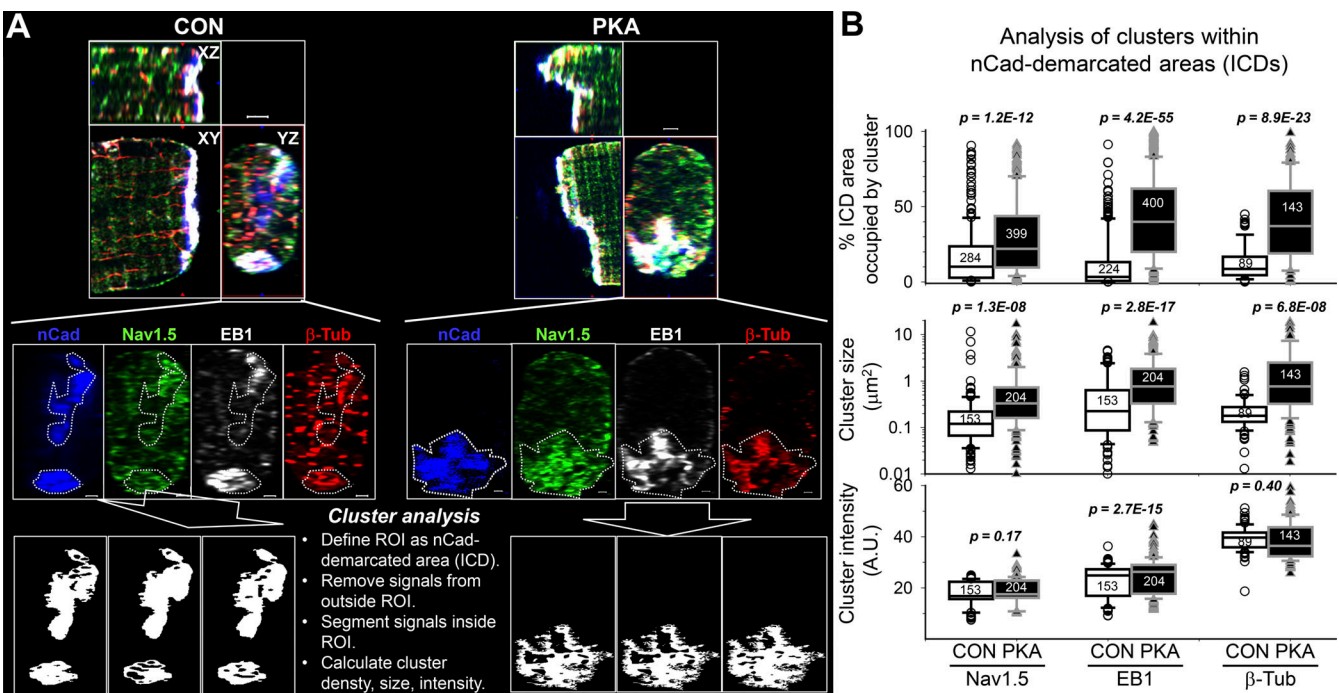

Figure 7. **Persistent PKA activation for 15 h increased the size and density of Na_V1.5, EB1, and β-Tub clusters at ICDs. (A)** Procedures of analyzing clusters at ICDs. Top: Orthogonal views of cell ends of CON and PKA myocytes, showing merged immunofluorescence signals of nCadherin (nCad, blue), Na_V1.5 (green), EB1 (white), and β-tub (red). Middle: YZ-plane views of individual immunofluorescence, with ROI demarcated by nCad (white dotted lines). Bottom: Signals outside ROIs were cleared (black background), and signals inside ROIs were segmented to define clusters (black clusters on white background). This was followed by quantification of cluster density (% ICD area occupied by clusters), average size (μm²), and mean immunofluorescence intensity in clusters. **(B)** Box plots of cluster parameters for Na_V1.5, EB1, and β-Tub at ICDs of CON and PKA myocytes. Data are pooled from four independent experiments. For each myocyte, 10–12 YZ-plane images at 0.25–0.5 μm intervals, advancing from the cell end toward the cell center (before nCad signals disappeared), were exported to ImageJ for analysis. Numbers shown in B are those of ROIs quantified. Listed P values are from t tests between CON and PKA myocytes. In A, scale bars are 2 μm for the top row and 1 μm for the middle and bottom rows.

interpretation. Therefore, we switched to HEK293 cells transfected with GFP-Na_V1.5. We compared four groups of cells: CON and PKA, each without or with chloroquine treatment. Fig. 12 A, top, depicts representative immunoblot images. WCL and biotinylated fraction (Biot') were probed for GFP-Na_V1.5, Na/K pump α subunit (NKA, loading control), and EB1 (to ensure no contamination by cytosolic proteins in Biot' lanes). Data quantification is plotted in 12A, bottom. While persistent PKA activation led to a twofold increase in GFP-Na_V1.5 on the cell surface (PKA:CON ratio of biotinylated/WCL = 3.01 ± 0.71), chloroquine prevented this increase. In fact, the PKA:CON ratio was <1 (0.46 ± 0.18), suggesting a hidden PKA effect that reduced surface GFP-Na_V1.5 unmasked by chloroquine.

## Discussion

The major findings in this study are summarized as follows: (a) PKA activation for 6–15 h caused a redistribution of Na_V1.5 to the peripheral surface and ICDs of adult rat ventricular myocytes, without altering the total Na_V1.5 protein level. In PKA myocytes, there was an increase in Na_V1.5 cluster size and density on the myocyte surface and higher Na⁺ current amplitudes recorded from cell-attached patches. (b) Persistent PKA activation caused an increase in EB1 transcript and protein levels and EB1 redistribution from puncta to peripheral surface and ICDs. (c) In PKA

myocytes, stable interfibrillar microtubules were reduced, while dynamic microtubules were prominent close to myocyte surface. (d) Persistent PKA activation increased correlative distribution among Na_V1.5, β-tubulin, and EB1 on the peripheral membrane and at ICDs. (e) Imaging and coimmunoprecipitation experiments suggested more frequent interactions between Na_V1.5 and EB1 as well as between Na_V1.5 and β-tubulin after persistent PKA activation. (f) Protein recycling inhibitor, chloroquine, prevented PKA activation-induced increase in cell surface Na_V1.5.

### Effects of persistent PKA activation are distinctly different from those of acute PKA activation on Na_V1.5 in cardiac myocytes

Earlier studies reported divergent effects of acute β-adrenergic stimulation on the $I_{Na}$ amplitudes in cardiac myocytes (increasing or decreasing, summarized in Matsuda et al. [1992]). It was then shown that the decreasing effect on $I_{Na}$ was due to a negative shift in the voltage dependence of Na⁺ channel inactivation in response to acute β-adrenergic stimulation (Matsuda et al., 1992). Recording $I_{Na}$ at hyperpolarized holding voltages, that removed Na⁺ channel inactivation, consistently showed an increase in $I_{Na}$ amplitude in response to acute β-adrenergic stimulation. Furthermore, the β-adrenergic effects can be divided into PKA-dependent and PKA-independent mechanisms (Matsuda et al., 1992). The PKA-dependent mechanism induced

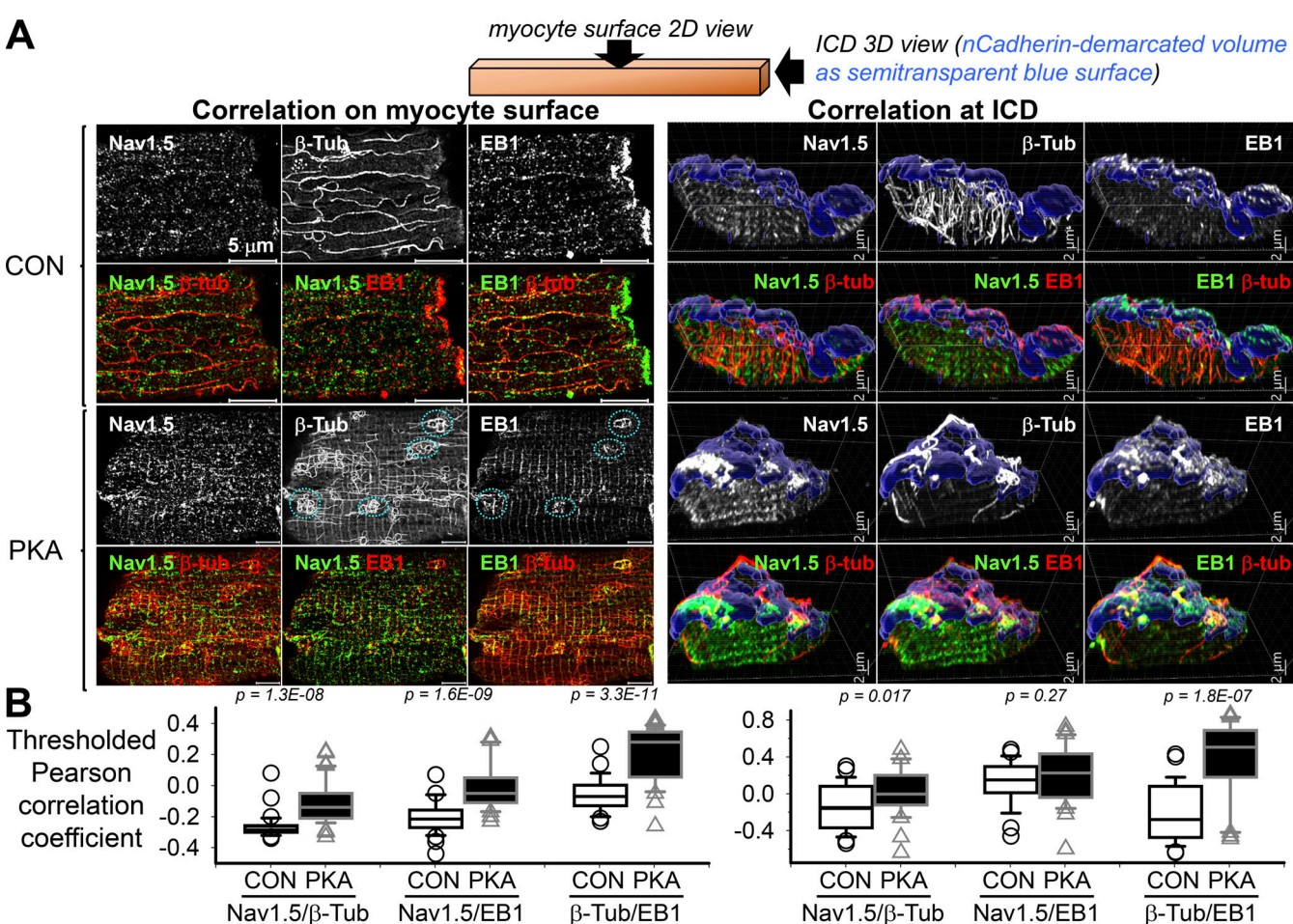

**Figure 8. Persistent PKA activation promoted the correlative distribution of Na$_V$1.5, β-Tub, and EB1 on myocytes' surface and ICDs.** Top: Cartoon depicting the planes of view. **(A)** Left: 2-D views of Na$_V$1.5, β-Tub, and EB1 immunofluorescence on myocytes' surface. Right: 3-D views of Na$_V$1.5, β-Tub, and EB1 immunofluorescence at myocytes' ICD. For each myocyte, six panels are shown: Na$_V$1.5, β-Tub, and EB1 individual immunofluorescence (upper row), and merged (lower row). In the PKA myocyte's surface images (lower left), cyan dotted circles denote looped microtubules (β-Tub panel) overlapped with EB1 puncta (EB1 panel). In the ICD images (right), semitransparent blue surfaces denote nCadherin-demarcated ICD volumes. **(B)** Box plots of thresholded Pearson correlation coefficients between the pairs of immunofluorescence signals listed along the abscissa in CON and PKA myocytes. Data were pooled from 29 to 41 myocytes per group from two independent experiments. Listed P values are from $t$ tests between CON and PKA myocytes. In A, scale bars are 5 μm for the myocyte surface views (left) and 2 μm for the myocyte ICD 3-D views (right).

an increase in I$_{Na}$ amplitude and an acceleration of I$_{Na}$ inactivation (Matsuda et al., 1992). The phosphorylation sites in the Na$_V$1.5 channel have been cataloged using the proteomics approach (Lorenzini et al., 2021). The phosphorylation sites that are relevant for the increase in I$_{Na}$ amplitude in response to acute PKA activation have been proposed to be in the cytoplasmic loop between domains I and II (Zhou et al., 2002).

Fig. S5 shows that acute β-adrenergic stimulation (by 100 nM isoproterenol, for 15 min) does not induce Na$_V$1.5 redistribution in rat ventricular myocytes. This is different from Na$_V$1.5 redistribution to the ICD region after persistent PKA activation. Furthermore, persistent PKA activation increased I$_{Na}$ amplitude without accelerating I$_{Na}$ inactivation (Fig. 1, B and C), suggesting a different mechanism than phosphorylation of the Na$_V$1.5 channel.

**Intracellular Na$_V$1.5 reservoir in cardiac myocytes**

If PKA activation promotes Na$_V$1.5 trafficking from an intracellular compartment to myocyte surface, where is this Na$_V$1.5

reservoir? Fig. 12 B provides several clues. First, adult rat atrial myocytes do not have t-tubules (no striations in WGA fluorescence signal, a t-tubule marker), but Na$_V$1.5 immunofluorescence exhibits prominent striations. Therefore, the striation pattern does not necessarily mean t-tubule localization. Second, rat ventricular myocytes transduced with Adv-GFP-Na$_V$1.5 and cultured for 36 h lost most of their t-tubules (little or no WGA striations). However, GFP-Na$_V$1.5 is in clear striations as well as in the perinuclear zone. This is consistent with newly translated GFP-Na$_V$1.5 trafficking along the nuclear envelope to SR along t-tubules or NEST pathway (He et al., 2020). The "SR along t-tubules" is junctional SR (jSR). Third, Na$_V$1.5 and RyR2 (jSR marker) overlap in their striation pattern, supporting the possibility that part of Na$_V$1.5 may be in the jSR compartment. Finally, although the localization of Na$_V$1.5 at ICD and along the lateral surface is consistent among freshly isolated rat ventricular myocytes, the striation component of Na$_V$1.5 is labile. It is missing in some myocytes, suggesting that Na$_V$1.5 may exit this

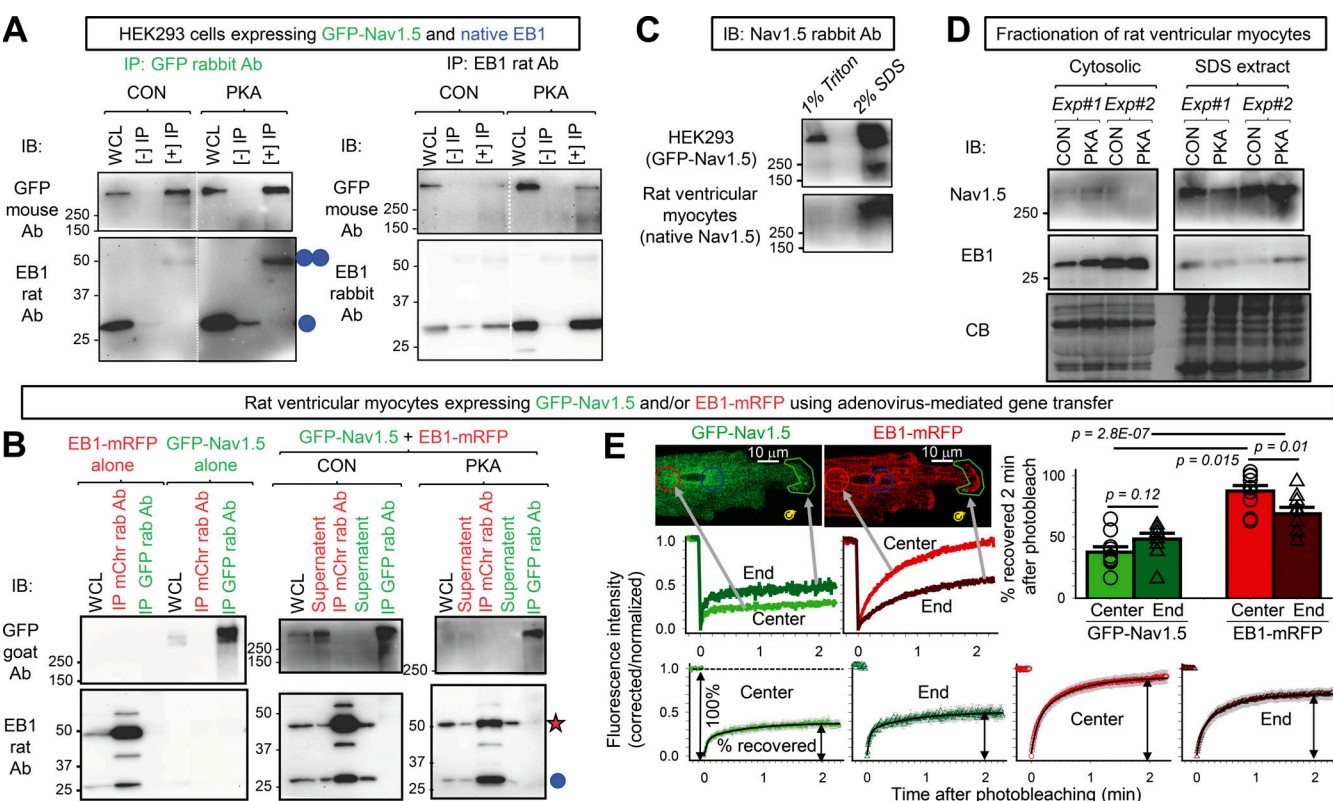

**Figure 9. Na$_V$1.5 and EB1 could be coimmunoprecipitated reciprocally when expressed in HEK293 cells but not in ventricular myocytes, while FRAP experiments suggested Na$_V$1.5/EB1 interactions at the ICD region of myocytes. (A)** GFP-Na$_V$1.5 expressed in HEK293 cells was coimmunoprecipitated with native EB1 reciprocally. WCL was prepared from HEK293 cells incubated with 8CPT-cAMP/okadaic acid for 4 h or time control with 1% Triton lysis buffer. (−) IP lanes were loaded with eluates from protein A/G beads incubated with WCL without antibody. (+) IP lanes were loaded with eluates from protein A/G beads incubated with WCL and immunoprecipitating (IP) antibody: GFP rabbit Ab (left) or EB1 rat Ab (right). The immunoblot (IB) Abs are listed on the left. Double and single blue circles denote EB1 dimer and monomer bands. **(B)** GFP-Na$_V$1.5 and EB1-mRFP coexpressed in cardiac myocytes did not coimmunoprecipitate. Experiments validating GFP-Na$_V$1.5 and EB1-mRFP as surrogates of native Na$_V$1.5 and EB1 in myocytes are presented in Fig. S4. Shown are protein(s) expressed in myocytes and conditions (CON or PKA; top), proteins loaded in lanes: WCL (prepared with 1% Triton lysis buffer), IP with mCherry "mChr" or GFP rabbit "rab" Ab, and supernatant (WCL after immunoprecipitation; middle), and immunoblot images probed with GFP goat Ab (upper row) or EB1 rat Ab (lower row). Left: Specificity of immunoprecipitation. EB1-mRFP expressed alone could be immunoprecipitated with mCherry Ab but not by GFP Ab, and GFP-Na$_V$1.5 expressed alone could be immunoprecipitated with GFP Ab but not by mCherry Ab. Right: In WCLs prepared from myocytes coexpressing GFP-Na$_V$1.5 and EB1-mRFP cultured under CON or PKA conditions for 15 h, mCherry rabbit Ab immunoprecipitated EB1-mRFP but not GFP-Na$_V$1.5, and GFP rabbit Ab immunoprecipitated GFP-Na$_V$1.5 but not EB1-mRFP or native EB1. Red star and blue circle denote the band positions of EB1-mRFP and native EB1, respectively. **(C)** GFP-Na$_V$1.5 in HEK293 cells was present in Triton-soluble fraction (detected in 1% Triton WCL), while Na$_V$1.5 in ventricular myocytes was not present in Triton-soluble fraction (undetectable in 1% Triton lane) but could be extracted with 2% SDS RIPA buffer (detected in 2% SDS lane). **(D)** Contrasting the subcellular environment of Na$_V$1.5 and EB1 in ventricular myocytes. Shown are immunoblot images of cytosolic and SDS extracted fractions of CON and PKA myocytes (incubation 15 h) probed for Na$_V$1.5 and EB1. The CB stain shows loading levels. The CB stain of the cytosolic fraction is modified from the one shown in Fig. 4 A. **(E)** Using FRAP to monitor mobilities of GFP-Na$_V$1.5 and EB1-mRFP expressed in ventricular myocytes. Top left: Representative images of a live myocyte with four ROIs marked: red—cell center, green—cell end, blue—reference in cell area not photobleached, yellow—background in cell-free area. The corresponding time courses of FRAP are plotted below. Background bleach was corrected based on fluorescence decline in ROI 3, and the fluorescence intensity was normalized to between 1 (right before photobleaching) and 0 (the first scan after photobleaching). Bottom: Average time courses of FRAP of GFP-Na$_V$1.5 and EB1-mRFP. Shown are the mean (colored bright and dark green for GFP-Na$_V$1.5 or bright and dark red for EB1-mRFP) and standard error (gray) values superimposed on double-exponential fit (black curve). Left most panel illustrates the calculation of "% of fluorescence recovered 2 min after photobleaching." Top right: Bar graphs (mean and SE) and individual data points of percentage of fluorescence recovered 2 min after photobleaching for GFP-Na$_V$1.5 and EB1-mRFP measured from cell center and cell end. Listed P values are from t tests between specified groups. Source data are available for this figure: SourceData F9.

compartment in response to changes in the cellular milieu. Fig. 12 C presents our working hypothesis: persistent PKA activation increases functional Na$_V$1.5 channels on the myocyte surface and at ICD by promoting EB1 expression and increasing dynamic (looped) microtubules in the myocyte periphery.

There are two possibilities for why Na$_V$1.5 stays in jSR. First, Na$_V$1.5 translated in the rough ER of nuclear envelope traffics through the jSR (He et al., 2020). This is supported by the time-

dependent distribution of GFP-Na$_V$1.5 in ventricular myocytes (Fig. S4 A). Second, native Na$_V$1.5 may be translated in jSR. A recent report using the so-called "mRNA-ribosomal RNA proximity-ligation in situ hybridization" approach showed distributed translation of native Na$_V$1.5 in mouse ventricular myocytes (Bogdanov et al., 2021). Furthermore, chaperones required for the folding of newly translated membrane proteins in rough ER, calnexin and binding immunoglobulin protein (BiP,

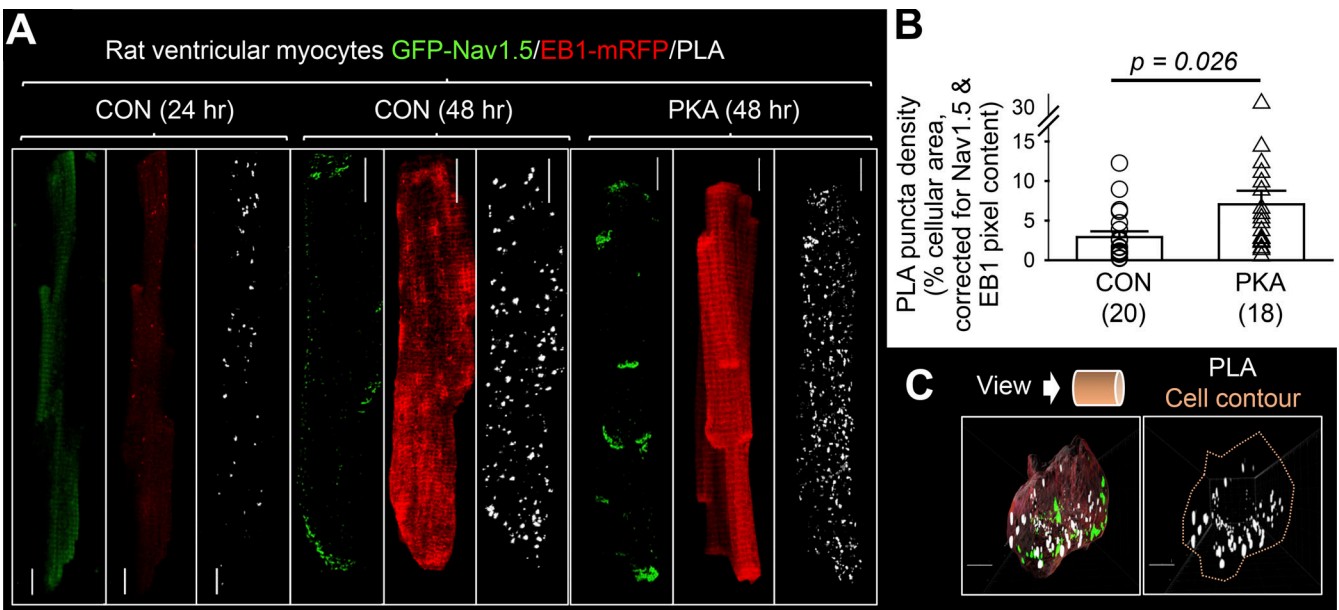

**Figure 10. Persistent PKA activation promoted Na$_V$1.5/EB1 interactions in myocytes. (A)** Detecting GFP-Na$_V$1.5 and EB1-mRFP interaction in ventricular myocytes by in situ proximity ligation assay (PLA). Shown are PLA signals and immunofluorescence of GFP-Na$_V$1.5 and EB1-mRFP in myocytes transduced with Adv-GFP-Na$_V$1.5 and Adv-EB1-mRFP and cultured for 24 h (low expression) and 48 h (strong expression), the latter under the control conditions or with PKA activation for 15 h. These images are z-stack projections of maxima. **(B)** Persistent PKA activation increased PLA puncta density. The images of z-projection of maxima were segmented in a systemic manner across CON and PKA myocytes to define PLA puncta, followed by calculating the percentage of cellular area occupied by PLA puncta. To correct for the differences in GFP-Na$_V$1.5 and EB1-mRFP expression level, the PLA puncta density of each of the myocytes was divided by the product of normalized GFP-Na$_V$1.5 and EB1-mRFP pixel content of the same myocyte from the same myocyte. Numbers in parentheses are myocytes studied from two independent experiments, with P value from t test between the two groups. **(C)** Top: Cartoon depicting plane of view. Bottom: 3-D views of PLA puncta in a CON myocyte, as a merge of PLA puncta (white), GFP-Na$_V$1.5 (green), and EB1-mRFP (salmon) or PLA puncta within cell contour (salmon line). Scale bars in all image panels are 10 μm.

also known as GRP-78), are concentrated along the z-lines in adult ventricular myocytes, suggesting the protein translation capacity of jSR (Jiang et al., 2017). Na$_V$1.5 may be stored in jSR by the RXR ER retention signals in its domain I–II loop (Zhou et al., 2002). Indeed, a previous study using the *Xenopus* oocyte model showed that mutating these RXR motifs enhances Na$_V$1.5 export to the cell surface and prevents further increase in cell surface Na$_V$1.5 by PKA activation (Zhou et al., 2002).

### EB1 distribution and its relationship with microtubules in CON and PKA myocytes

An in vitro investigation of EB1/microtubule interactions (Vitre et al., 2008) proposed that EB1 binding to the microtubule lattice can stabilize microtubules, while EB1 binding to the microtubule plus-ends can promote their depolymerization (catastrophe)/repolymerization (growth) dynamics. Translating these in vitro findings to EB1 and microtubules in cells, EB1 binding to microtubule lattice stabilizes these cytoplasmic cytoskeletons that are important for mechanical strength and protein trafficking. EB1 binding to microtubule plus ends in the cell periphery promotes their dynamics, serving the purpose of rapid reorganization of cell cortex cytoskeletons to meet the cells' needs, such as migration and membrane protein redistribution. In CON myocytes, small EB1 speckles overlapped with the microtubule lattice (yellow dots along the β-Tub strands in the merged EB1/β-Tub view, Fig. 8 A), indicating EB1 binding to microtubule lattice. This is similar to a previous report from mouse

ventricular myocytes (Drum et al., 2016), but different from the "EB1 comet" morphology seen in some other cell types (Shaw et al., 2007). The latter reflects preferential EB1 binding to the growing plus-ends of microtubules. Our data presented in Fig. 4 and Fig. 8 A collectively show that in CON myocytes, EB1 mainly binds along the microtubule lattice and stabilizes interfibrillar microtubules.

In PKA myocytes, although the total α-/β-tubulin protein levels were not changed, the portion of stable interfibrillar microtubules, reported by detyrosinated α-tubulin, was significantly reduced (Fig. 4). Both EB1 and β-Tub were enriched at the cell periphery and ICD (Figs. 6 and 7), where EB1 overlapped with looped β-Tub structures (Fig. 8 A). These observations indicate that in PKA myocytes, enhanced EB1 interactions with cell peripheral microtubules promote their catastrophe/growth dynamics, and the looped microtubule structures reflect fast-growing microtubules close to the myocyte surface (Tortosa et al., 2013).

Persistent PKA activation increased Na$_V$1.5/EB1 encounters (Fig. 10) and apparently strengthened Na$_V$1.5/β-Tub association (Fig. 11). These data support a scenario where PKA activation facilitates Na$_V$1.5-containing vesicles trafficking on microtubules guided by EB1 (and likely other EB1-associated proteins [Shaw et al., 2007; Marchal et al., 2021]) toward the cell periphery. This scenario is supported by the increase in correlation in Na$_V$1.5/β-Tub and Na$_V$1.5/EB1 on myocyte surface and at ICDs (Fig. 8).

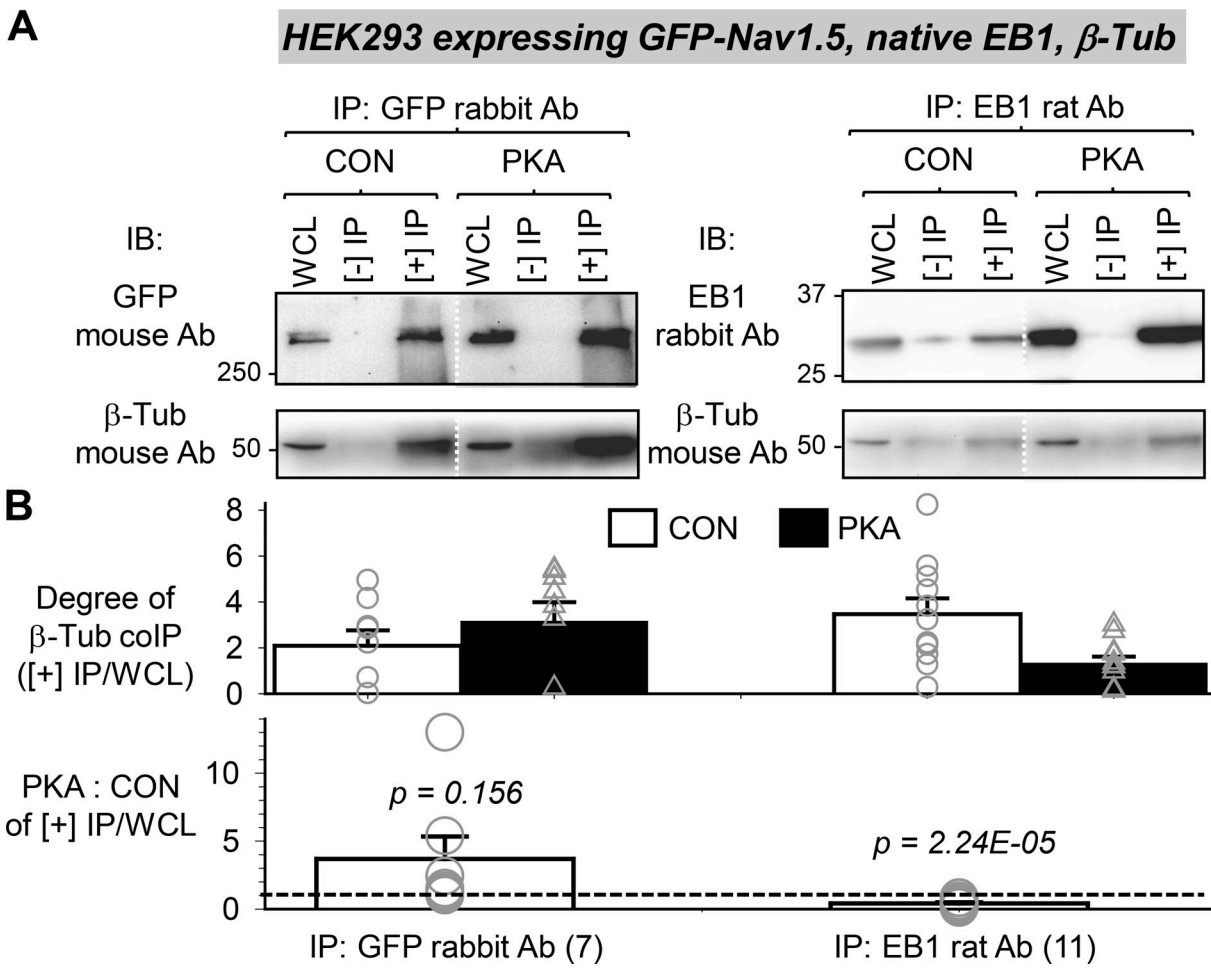

**Figure 11. EB1 and β-Tub coimmunoprecipitation was reduced, while Na$_V$1.5 and β-Tub coimmunoprecipitation (co-IP) trended higher with persistent PKA activation.** Experiments were done in HEK293 cells expressing GFP-Na$_V$1.5 and native EB1 and β-Tub, without or with PKA activation (4 h). Cells were lysed in 1% Triton lysis buffer, and WCLs were subject to immunoprecipitation with GFP rabbit or EB1 rat Ab and protein A/G beads ([+] IP). "(−) IPs" were negative control (WCL incubated with protein A/G beads without immunoprecipitating antibodies). **(A)** Representative immunoblot images probed with Abs listed on the left. The EB1 rabbit Ab IB image of EB1 rat Ab IP is modified from the same experiment image shown in Fig. 9 A. **(B)** Top: Degrees of β-Tub co-IP with GFP-Na$_V$1.5 or EB1 from CON and PKA cells quantified by β-Tub band intensity in (+) IP lane divided by that in WCL ([+] IP/WCL). Bottom: PKA:CON ratio of degree of β-Tub co-IP with GFP-Na$_V$1.5 or with EB1. The dotted line denotes PKA:CON of 1. Numbers in parentheses are those of independent experiments. Listed P values are from t tests against null hypothesis. Source data are available for this figure: SourceData F11.

### Role of microtubules in Na$_V$1.5 distribution and function

Fig. 13 A shows that disrupting microtubules by nocodazole prevents GFP-Na$_V$1.5 trafficking from the perinuclear zone to the lateral surface and ICDs despite the maintained ICD structure based on the immunofluorescence of nCadherin and EB1. Without microtubules, GFP-Na$_V$1.5 was trapped in cytoplasmic vesicles. These observations indicate that newly translated Na$_V$1.5 requires intact microtubules to traffic to their destinations, either as Na$_V$1.5-containing vesicles or as Na$_V$1.5-containing jSR trafficking on microtubules (He et al., 2020). On the other hand, nocodazole does not alter the distribution pattern of native Na$_V$1.5 despite clear disruption of interfibrillar microtubules (Fig. 13 B). This can be explained by the fact that native Na$_V$1.5 has already reached its destination, where it is anchored by macromolecular complexes (Chen-Izu et al., 2015; Rook et al., 2012).

Although disrupting interfibrillar microtubules does not alter the overall pattern of Na$_V$1.5 distribution, proper Na$_V$1.5 function on the myocyte surface does require dynamic microtubules at the cell periphery. It was shown that stabilizing microtubules by taxol reduces cell surface Na$_V$1.5 and suppresses I$_{Na}$ amplitude in neonatal rat cardiomyocytes (Casini et al., 2010). Furthermore, reducing stable microtubules by parthenolide (Kerr et al., 2015) doubles the I$_{Na}$ density in ventricular myocytes from a mouse model of arrhythmogenic cardiomyopathy (homozygous knockout of plakophilin-2) that exhibits microtubule densification and I$_{Na}$ reduction (Nasilli et al., 2022). It is possible that a dynamic microtubule network can surveil the sub-sarcolemmal space, capturing Na$_V$1.5-containing vesicles and delivering them to the myocyte surface. This scenario is consistent with our observations that in PKA myocytes subsarcolemmal microtubules became more dynamic while the cell surface of Na$_V$1.5 was increased.

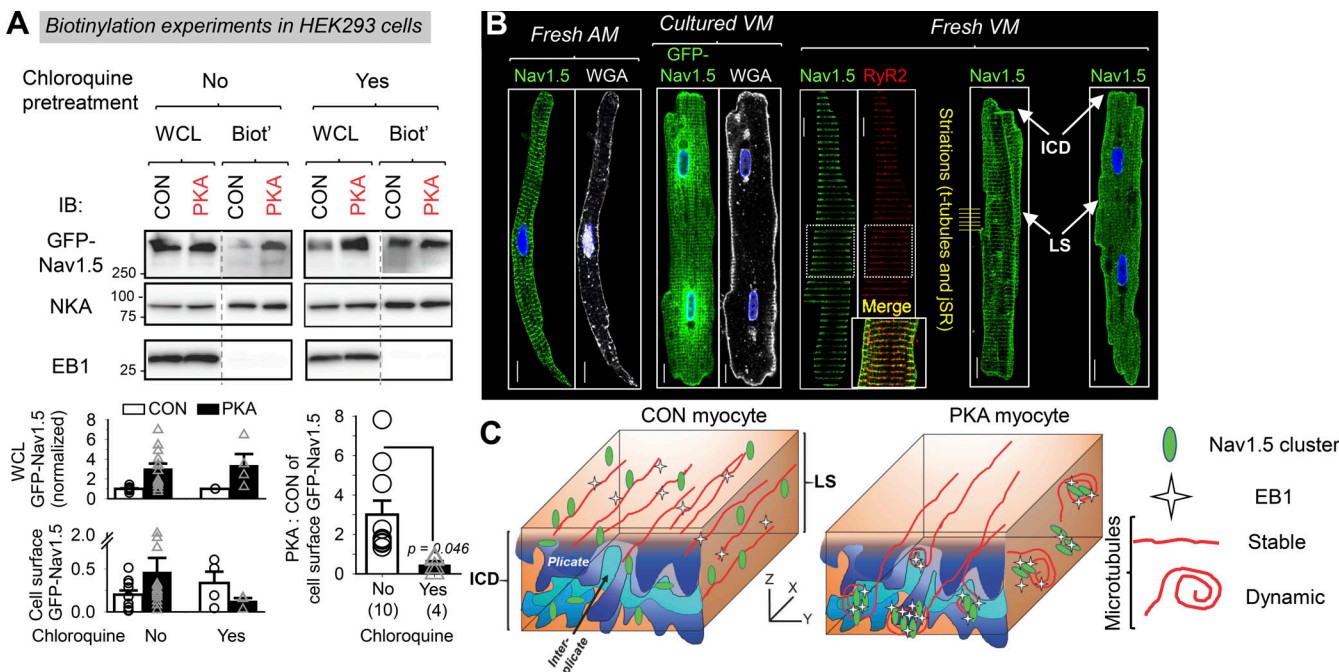

**Figure 12.** **Persistent PKA activation promoted trafficking of Na$_V$1.5 from a cytosolic reservoir to the plasma membrane. (A)** Biotinylation experiments in HEK293 cells expressing GFP-Na$_V$1.5. Top: Representative immunoblot images of WCL and biotinylated fraction (Biot') from HEK293 cells expressing GFP-Na$_V$1.5 and native EB1, exposed to PKA for 4–6 h or without PKA for the same duration, without or with pretreatment with chloroquine 100 µM (chloroquine was present during incubation with PKA or CON). Abs used in immunoblotting are listed on the left. NKA = Na/K pump α-subunit as the loading control. The absence of EB1 in Biot' lanes confirms the lack of contamination from cytosolic proteins. Bottom left: Summary of WCL GFP-Na$_V$1.5 (normalized to CON) and cell surface GFP-Na$_V$1.5 (Biot'/WCL), shown as mean ± SE with individual data points. Right: PKA:CON ratio of cell surface GFP-Na$_V$1.5 in "No chloroquine" and "With chloroquine" groups ($n$ =10 and 4, respectively; $t$ test between the two groups: P = 0.046). **(B)** Immunofluorescence images of native Na$_V$1.5 or GFP-Na$_V$1.5, RyR2, and fluorescence images of wheat germ agglutinin (WGA, marker of plasma membrane and t-tubules) from the types of myocytes listed above. A detailed description is in the Discussion section. LS: lateral surface. The myocyte image second from right is a duplicate from Fig. 3 A, Na$_V$1.5 immunofluorescence image from a control myocyte after culture for 1 h. **(C)** Cartoon of working hypothesis. Scale bars in B are 10 mm. Source data are available for this figure: SourceData F12.

## Pathophysiological implications

Persistent PKA activation occurs in chronically stressed heart, e.g., heart failure and aging (de Lucia et al., 2018). Our finding of increased Na$_V$1.5 clustering at ICDs in PKA-activated myocytes suggests protection of impulse propagation through gap junctions at ICDs. An additional benefit is the reduction of detyrosinated microtubules that can improve myocyte contractile function (Chen et al., 2018).

Transcriptomic experiments show that persistent β-adrenergic stimulation (isoproterenol administration for 14 d) causes a greater than twofold increase in EB1 transcript in mouse heart (Ochsner et al., 2019), likely related to the activation of the transcription factor, CREB1 (Zhang et al., 2005). Indeed, our RNA-seq and RT-qPCR experiments confirmed marked EB1 upregulation in PKA versus CON myocytes (Fig. S2). Upregulation of EB1 can have far-reaching implications. In addition to Na$_V$1.5, EB1 is involved in targeting connexin-43 to ICDs (Shaw et al., 2007) and targeting KCNQ1 to the peripheral membrane (Wilson et al., 2021). Future experiments will examine EB1 interactomes in CON and PKA myocytes to understand proteins that help EB1, or need help from EB1, for proper targeting.

## Technical consideration

Fig. 14 shows that forskolin (elevating native cAMP by activating adenylate cyclases) mimicked the effects of PKA activation, while a membrane-permeable PKA peptide inhibitor (PKI-14-22 amide, myristoylated) reduced the effects of PKA without affecting CON myocytes, in terms of Na$_V$1.5 clustering to ICDs. These data strengthen the notion that experimental findings reported here reflect PKA activation instead of EPAC activation or other unidentified mechanisms.

The cytomegalovirus promoter in the Adv-GFP-Na$_V$1.5 construct has three cAMP response elements. Incubation with 8CPT-cAMP will increase the GFP-Na$_V$1.5 protein level, interfering with experiments designed to study Na$_V$1.5 distribution and function. Therefore in all myocyte experiments, we focused on native Na$_V$1.5, except those presented in the following figures where an increase in GFP-Na$_V$1.5 protein level was not expected to affect the conclusions (Fig. 9, B and E; Fig. 12 B; and Fig. 13 A), or when data could be corrected for the increase in GFP-Na$_V$1.5 expression (Fig. 10).

Fig. 2 A shows that with a difference in Z plane position of only 0.3 µm, the Na$_V$1.5 pattern switched from striations and lateral surface (myocyte interior) to random cluster (myocyte surface), supporting the ability of high-resolution Airyscan to resolve myocyte surface Na$_V$1.5 signals from those in myocyte interior. Our observations with Na$_V$1.5 are similar to the change in RyR2 distribution from striations alone z-lines (when viewed inside a myocyte) to random clusters on the myocyte surface (Hiess et al., 2018).

To quantify Na$_V$1.5 clusters in the ICD areas, ideally, we would obtain en face views of ICDs. This can be achieved by

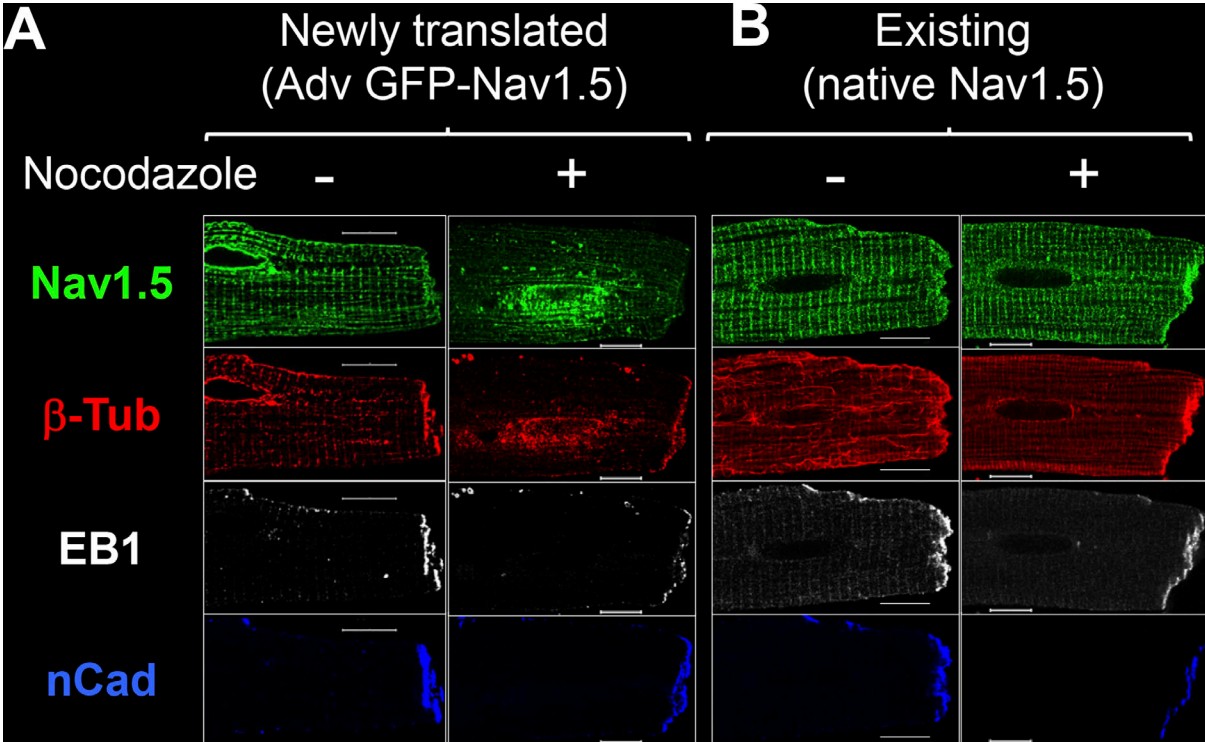

Figure 13. **Differential effects of nocodazole on newly translated versuss existing Na$_V$1.5. (A and B)** Disrupting microtubules by nocodazole (10 µM, 12 h) prevented newly translated GFP-Na$_V$1.5 from reaching the lateral surface and intercalated discs (A), but did not alter the distribution pattern of native Na$_V$1.5 in ventricular myocytes (B). Scale bars are 10 µm.

immunostaining thin sections of the myocardium and looking for en face ICD views under a microscope (Veeraraghavan et al., 2018). This approach is not suitable because we are interested in the specific effects of PKA activation, instead of systemic PKA activation that involves multiple signaling cascades, whose interactions and effects could be intractable. Another approach is to embed isolated and immunostained myocytes in agar, cut the agar into thin slices, and look for en face ICD views in the slices under a microscope (Chen-Izu et al., 2006). This approach was not successful in our hands. Therefore, we used 3-D Airyscan and created en face ICD views by reconstructing YZ-plane images. Although we could not rule out the possibility of overestimation of cluster sizes due to the point spread function in the Z-axis, our comparison was between CON and PKA myocytes that were imaged and analyzed under the same conditions. The marked differences between the two sets of data supported genuine PKA effects.

It is known that the *creb1* gene produces multiple CREB1 isoforms through alternative splicing. A search of the NCBI Protein database with the term "rat CREB1" retrieved more than five CREB1 isoforms, ranging from 26 to 49 kD. The 37 kD one is the major CREB1 isoform ubiquitously expressed. A previous study of the relationship between CREB1 isoforms and long-term memory formation in the *Alysia* model showed that PKA phosphorylated only one of the CREB1 isoforms, which then entered the nuclei to modulate gene transcription (Bartsch et al., 1998). The immunoblot in Fig. 1 A shows that the CREB1 antibody detected a single band at ~37 kD in both CON and PKA myocytes, while the CREB1-S133$^P$ antibody detected an extra band of ~50 kD in PKA but not CON myocytes. It is possible

that PKA phosphorylated the 49 kD CREB1 isoform detected by CREB1-S133$^P$ antibody, and it was this 49 kD isoform that entered nuclei to modulate gene transcription in ventricular myocytes.

### Data availability
Data are available in the article itself and its supplementary materials.

### Acknowledgments
Jeanne M. Nerbonne served as editor.

The authors would like to thank Dr. Seth Weinberg (Department of Biomedical Engineering, the Ohio State University) for his critical reading and feedback on this manuscript.

This study was supported by HL163101 (to G.-N. Tseng) and HL94459 (to I. Deschenes) from the National Heart, Lung, and Blood Institute of the National Institutes of Health (NIH). Confocal imaging and bioinformatics services were provided by the VCU Massey Cancer Center Shared Resource, supported in part, with funding from NIH-National Cancer Institute Cancer Center Support Grant P30 CA016059.

Author contributions: T. Bernas: methodology, formal analysis, and investigation; J. Seo: immunoblot data acquisition and investigation; Z.T. Wilson: image data acquisition and investigation; B.-h. Tan: RT-qPCR experiment; I. Deschenes: funding acquisition and RT-qPCR experiment; C. Carter and J. Liu: RNA-seq data analysis; G.-N. Tseng: funding acquisition,

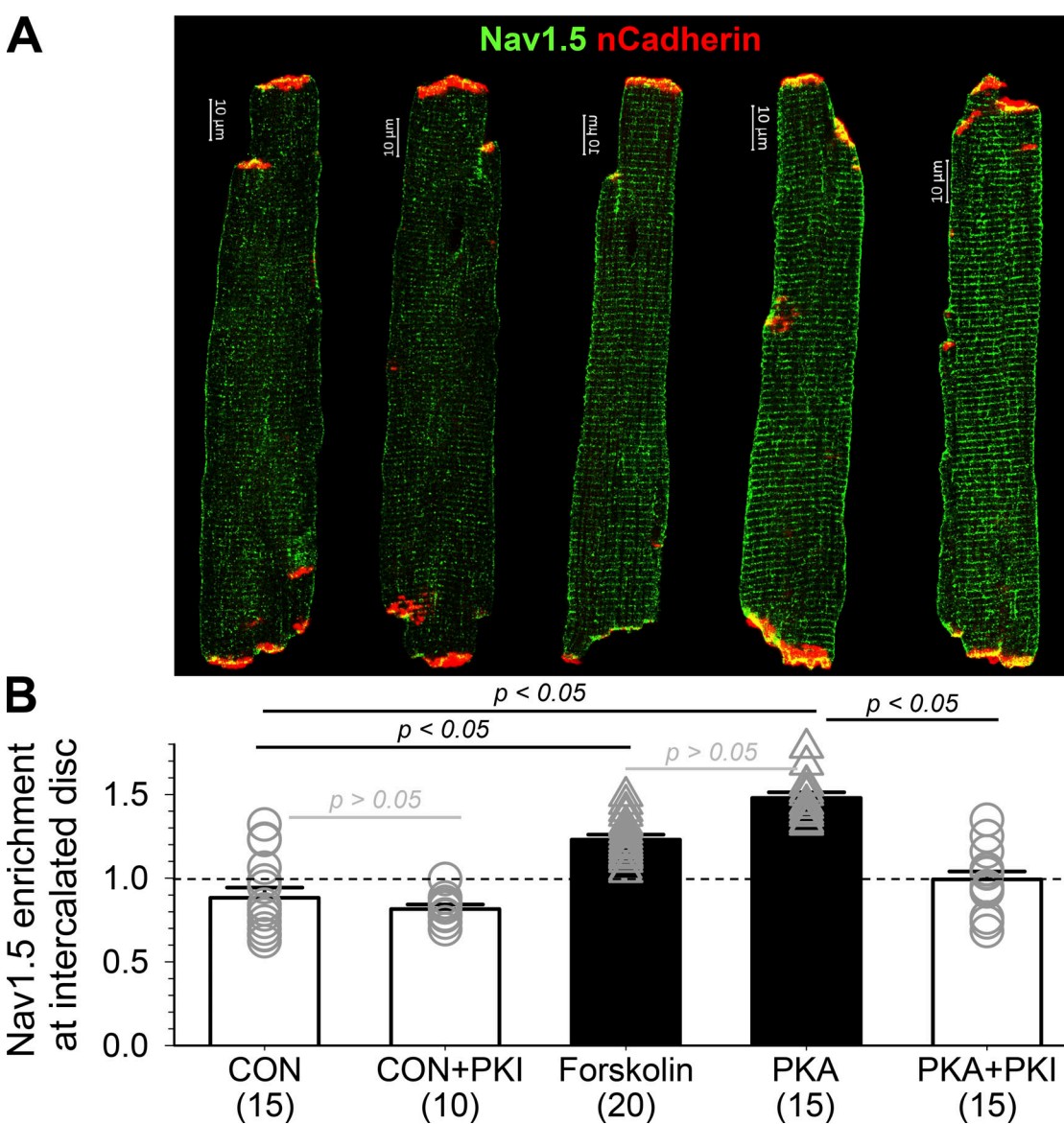

**Figure 14.** **Forskolin (adenylate cyclase activator) mimics the effects of PKA activation, while a membrane-permeable PKA peptide inhibitor (PKI-14-22 amide, myristoylated) reduces the effects of PKA without affecting CON myocytes.** Five groups of myocytes isolated from the same heart were incubated for 4 h under the control conditions (CON), with 10 μM PKI (CON+PKI), with 10 μM forskolin, with 8CPT-cAMP (100 μM)/okadaic acid (100 nM) (PKA), and further 10 μM PKI (PKK+PKI). Myocytes were immunostained for Na$_V$1.5 and nCadherin, and subject to Airyscan imaging followed by quantification. **(A)** Representative merged Na$_V$1.5 and nCadherin views of myocytes corresponding to the groups in B. **(B)** Summary of Na$_V$1.5 enrichment at ICDs (nCad-positive areas) from the five groups listed along the abscissa. The dotted line at the y-axis 1 denotes no enrichment. Numbers in parentheses are those of myocytes quantified. Statistical analysis was one-way ANOVA on ranks, P < 0.05, followed by all-pairwise comparisons using Dunn's method. Shown are P values <0.05 or >0.05 between specified groups.

project administration, conceptualization, data acquisition, data analysis, writing—original draft, and writing-review & editing.

Disclosures: The authors declare no competing interests exist. J Seo and Z.T. Wilson were premed students working in their labs as technicians while waiting for med school admission. They are now in med school and are unreachable.

Submitted: 14 June 2023

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

## Supplemental material

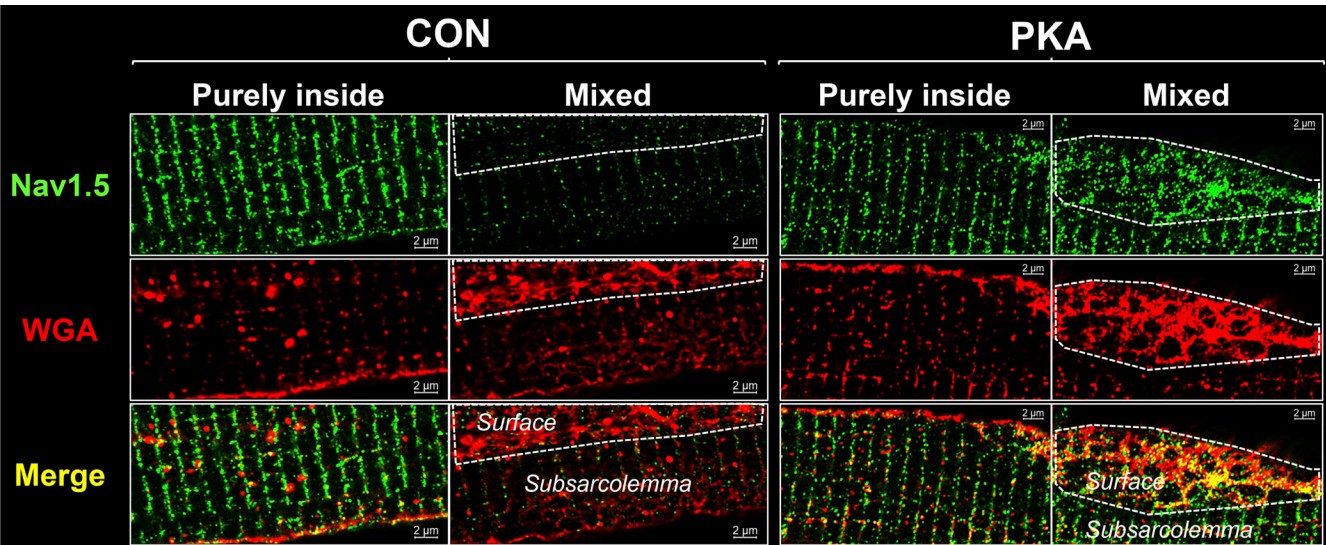

Figure S1.  **Using wheat germ agglutinin (WGA) to verify differential distribution patterns of Na$_V$1.5 inside myocytes and on myocyte surface.** CON and PKA myocytes (incubation time 15 h) were stained for native Na$_V$1.5 (green) and WGA (red). Z-stack images at 50-nm z steps were collected. Shown are images of Na$_V$1.5, WGA, and their merge at z planes deep inside myocytes (purely inside), and at the interface between surface and subsarcolemmal space (mixed). In the latter, the surface areas are demarcated by dashed white lines. In the "purely inside" views, both Na$_V$1.5 and WGA are in striations of ~2 μm spacing, and along the lateral cell edge. There are WGA-positive vesicles. In the "mixed" views, Na$_V$1.5 and WGA are in faint striations in the subsarcolemmal space. However, WGA manifests a vague hexagonal pattern while Na$_V$1.5 is in random clusters on the surface. Na$_V$1.5 clusters are much denser on the surface of PKA than CON myocytes.

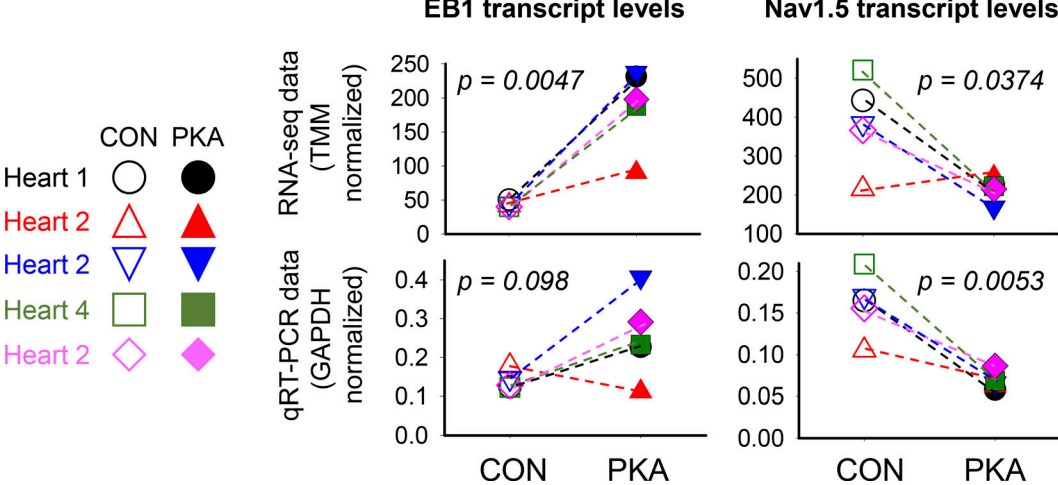

**Figure S2.** **The EB1 transcript was upregulated, while Na$_V$1.5 transcript was downregulated in PKA versus CON myocytes.** Total RNAs were extracted from CON and PKA myocytes (incubation time 15 h, five independent experiments) with TRIZOL reagents. After confirming high sample quality (RNA integrity number $\geq$7), the RNA concentrations were adjusted to 100 ng/µl and used for two groups of experiments. Group 1 was RNA-seq experiment: mRNA library construction followed by next-generation sequencing (VCU Genomics Core). Raw FASTQ sequences of individual samples were inspected for quality using the tool FastQC and then merged with multiQC. PE 100 bp reads were aligned to the rat primary genome assembly (mRatBN7.2, version 110). Raw counts were normalized between and within samples, using EdgeR's *calcNormFactors* scaling factor of trimmed mean of M-values (TMM), that took into account of variations in library size, sequencing depth, and gene lengths. Group 2 was quantitative RT-PCR experiment: mRNA was reverse transcribed followed by PCR reactions using the following primer pairs that cross exon-intron boundaries: (a) EB1: forward 5′-TGTCGCTCCAGCTTTGAGTA-3′, reverse 5′-AGCAGCTTCGTC ATCTCCAT-3′, (b) Na$_V$1.5: forward 5′-TCAATGACCCAGCCAATTACCT-3′, reverse 5′-CCCGGCATCAGAGCTGTT-3′. House-keep transcript GAPDH was included in the qPCR reactions. Data of TMM (RNA-seq) or GAPDH (qPCR) normalized transcript levels are presented as individual data points of CON-PKA pairs. Each pair is connected by a dashed line for visual effect clarification. The P values are from paired *t* tests between CON and PKA data points.

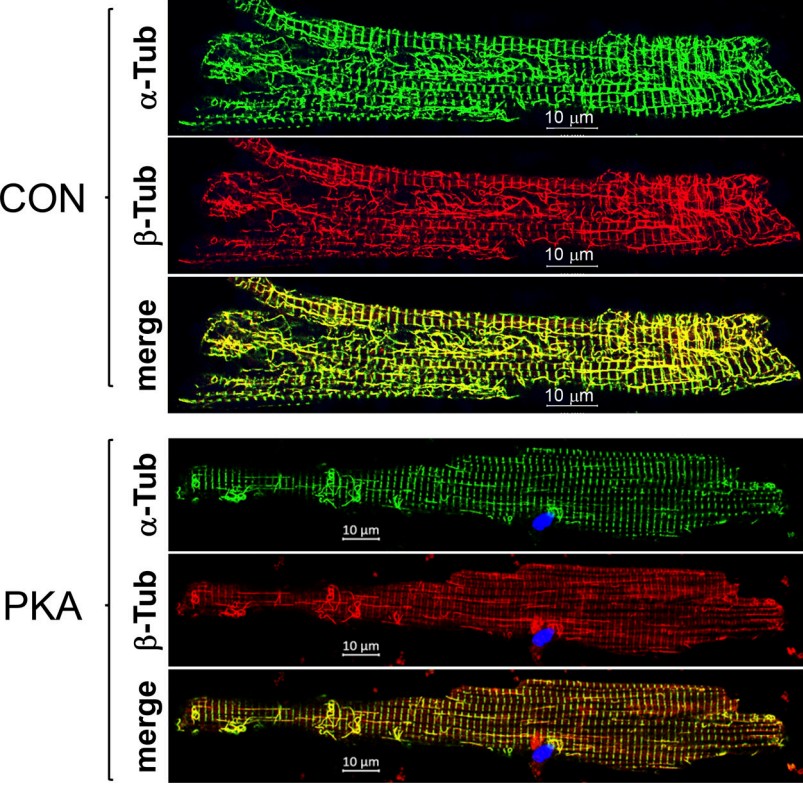

**Figure S3.** **Colocalization of α-tubulin and β-tubulin (α- and β-Tub, respectively) immunofluorescence in rat ventricular myocytes.** Shown are Airyscan images α-Tub and β-Tub immunofluorescence (detected by α-Tub rat Ab and β-Tub mouse Ab) and their merge in control and PKA myocytes (PKA activation 12 h). Images were obtained in a z-plane close to the myocyte's surface.

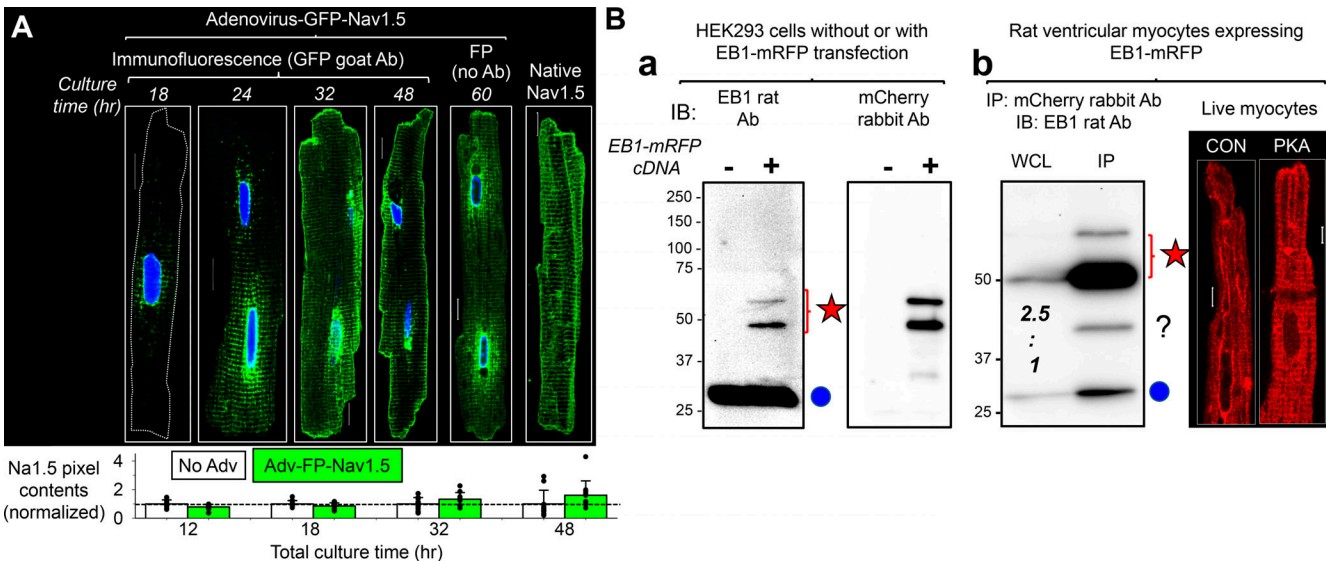

Figure S4.    **Validating GFP-Na$_V$1.5 and EB1-mRFP as surrogates for native Na$_V$1.5 and EB1 in ventricular myocytes.** Validating GFP-Na$_V$1.5 and EB1-mRFP as surrogates for native Na$_V$1.5 and EB1 in ventricular myocytes. **(A)** Myocytes were incubated without (no Adv) or with Adv-GFP-Na$_V$1.5 for 12 h. The medium was exchanged for virus-free fresh medium and culture continued for 0, 6, 20, 36, or 48 h (total culture times: 12, 18, 32, 48, and 60 h). At specified times, myocytes were fixed for experiments. Top: Airyscan images of GFP-Na$_V$1.5 immunofluorescence detected with GFP goat Ab (culture time: 18–48 h), and GFP fluorescence (FP) of a myocyte cultured for 60 h. Right: Immunofluorescence of native Na$_V$1.5 in a freshly isolated myocyte. At 18 h, GFP-Na$_V$1.5 clustered around nuclear envelope representing protein translation in the NE/rER. At 24 h, GFP-Na$_V$1.5 spread out in striations along z-lines, similar to the nuclear envelope to SR along t-tubules or NEST pathway (He et al., 2020). By 32 h, GFP-Na$_V$1.5 reached the lateral surface and ICD. The distribution pattern was stable at 48 and 60 h culture times. These data showed that the steady-state distribution pattern of GFP-Na$_V$1.5 was similar to that of native Na$_V$1.5, except active GFP-Na$_V$1.5 translation at NE/rER. Bottom: Degree of GFP-Na$_V$1.5 expression versus native Na$_V$1.5 based on pixel contents. Myocytes without or with Adv-GFP-Na$_V$1.5 for the total culture times shown along the abscissa were immunostained with Alomone asc005 rabbit Ab, which detected both GFP-Na$_V$1.5 and native Na$_V$1.5. The total pixel contents were quantified and normalized to the mean pixel contents of "No Adv" myocytes of the same culture time. Shown is a bar graph of mean $\pm$ SE superimposed with individual data points. "Adv-GFP-Na$_V$1.5" myocytes did not have more immunofluorescence than "no Adv" myocytes until the 32 h time point. By the 48 h time point, "Adv-GFP-Na$_V$1.5" myocytes had 30% more immunofluorescence than "no Adv" myocytes. These data showed a modest expression of GFP-Na$_V$1.5 over native Na$_V$1.5. The native Na$_V$1.5 immunofluorescence image is a duplicate from Fig. 3 A, Na$_V$1.5 immunofluorescence image from a control myocyte after culture for 1 h. **(B) (a)** Confirming immunoblot banding pattern of EB1-mRFP and native EB1 in HEK293 cells. EB1 rat Ab detected only the native EB1 band (30 kD, blue dot) in untransfected cells, while it detected both native EB1 and EB1-mRFP (expected 55.7 kD, two bands at and above 50 kD, red star) in transfected cells. The identity of the two EB1-mRFP bands was confirmed by reprobing the membrane with mCherry rabbit Ab. **(b)** EB1-mRFP expressed in ventricular myocytes. Left: IP lane shows EB1-mRFP coimmunoprecipitation with native EB1 (red star and blue circle), indicating dimerization between the two to form active EB1 (Chen et al., 2014). The WCL lane shows that Adv-mediated EB1-mRFP expression was ~150% over native EB1 (band intensity ratio 2.5:1). This immunoblot image is modified from the same experiment shown in Fig. 9 B, bottom right. Right: In live myocytes (images obtained during FRAP experiments, Fig. 9 E), EB1-mRFP had a wavy strand morphology along the long axis of CON myocyte, suggesting binding along the microtubule lattice. In PKA myocyte, EB1-mRFP clustered to ICD and manifested striations at the z-plane adjacent to myocyte surface. The difference in EB1-mRFP patterns between CON and PKA myocytes is similar that of native EB1 between CON and PKA myocytes (Figs. 4 and 8). Scale bars are 10 mm in A and 5 mm in B. Source data are available for this figure: SourceData FS4.

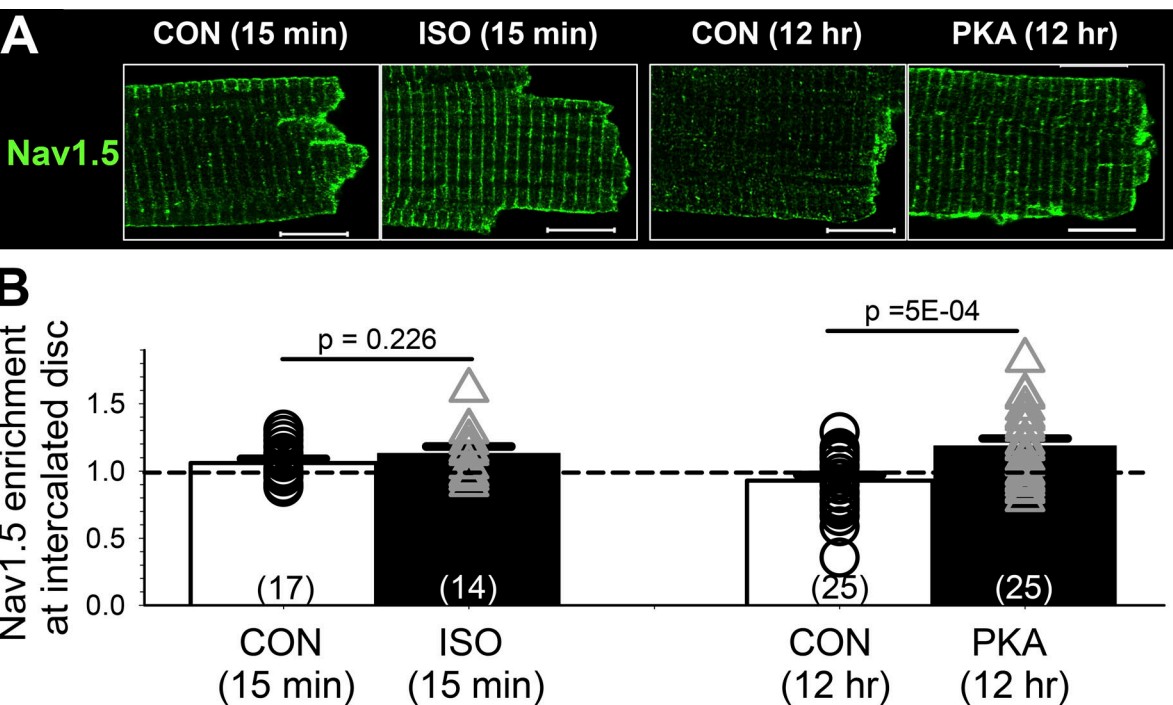

**Figure S5. Acute β-adrenergic stimulation does not induce Na$_V$1.5 clustering to intercalated disc region.** Adult rat ventricular myocytes were incubated under the control conditions or with isoproterenol (ISO, 100 nM) for 15 min before the experiment. Myocytes from the same heart were cultured under the control conditions or with 8CPT-cAMP (100 μM)/okadaic acid (100 nM) for 12 h before the experiment. **(A)** Representative Airyscan images of Na$_V$1.5 immunofluorescence from the four groups of myocytes. **(B)** Bar graph (mean ± SE) and individual data points of Na$_V$1.5 enrichment at intercalated discs. The numbers of myocytes analyzed are in parentheses. Listed P values are from *t* test between PKA and CON myocytes. Scale bars in A are 10 μm.

**Video 1. A CON myocyte cultured for 12 h under the control conditions and immunostained for nCadherin (blue, sheep anti-nCad/Alexa405 donkey anti-goat), Na$_V$1.5 (green, rabbit anti-Na$_V$1.5/Alexa488 donkey anti-rabbit), β-Tub (red, mouse anti-β-tubulin/Alexa568 donkey anti-mouse), and EB1 (white, rat anti-EB1/Alexa647 goat anti-rat).** The video shows rotating 3-D views of the myocyte end in the following sequence: (a) merged four colors, (b) nCad alone, (c) nCad plus the ICD volume as a magenta semitransparent surface, (d) Na$_V$1.5 with the ICD volume, (e) EB1 with the ICD volume, and (f) β-Tub with the ICD volume. The 3-D views and animation were created with IMARIS (v. 10).

**Video 2. A PKA myocyte cultured for 12 h with 8CPT-cAMP (100 μM) and okadaic acid (100 nM) and immunostained for nCadherin (blue, sheep anti-nCad/Alexa405 donkey anti-goat), Na$_V$1.5 (green, rabbit anti-Na$_V$1.5/Alexa488 donkey anti-rabbit), β-Tub (red, mouse anti-β-tubulin/Alexa568 donkey anti-mouse), and EB1 (white, rat anti-EB1/Alexa647 goat anti-rat).** The video shows rotating 3-D views of the myocyte end in the following sequence: (a) merged four colors, (b) nCad alone, (c) nCad plus the ICD volume as a magenta semitransparent surface, (d) Na$_V$1.5 with the ICD volume, (e) EB1 with the ICD volume, and (f) β-Tub with the ICD volume. The 3-D views and animation were created with IMARIS (v. 10).

