## [Peer Review File · The Journal of General Physiology]

Persistent PKA activation redistributes Nav1.5 to the cell surface of adult rat ventricular myocytes

Tytus Bernas, John Seo, Zachary Wilson, Bi-Hua Tan, Isabelle Deschenes, Christiane Carter, Jinze Liu, and Gea-Ny Tseng

Corresponding Author(s): Gea-Ny Tseng, Virginia Commonwealth University

Review Timeline:

Submission Date:	June 14, 2023
Editorial Decision:	July 13, 2023
Revision Received:	September 15, 2023
Editorial Decision:	October 10, 2023
Revision Received:	November 8, 2023
Editorial Decision:	November 28, 2023
Revision Received:	December 5, 2023

Editor: Jeanne Nerbonne

Transaction Report:

DOI: <https://doi.org/10.1085/jgp.202313436>

July 13, 2023

Prof. Gea-Ny Tseng
Virginia Commonwealth University
Physiology & Biophysics
1101 E MARSHALL ST
Sanger 3-042a
Richmond, VA 23298

Re: 202313436

Dear Gea-Ny,

Thank you for submitting your manuscript, titled "Persistent PKA activation promotes Nav1.5 trafficking to cardiomyocyte surface and intercalated discs by upregulating microtubule plus-end binding protein 1 (EB1) and reorganizing microtubules", to JGP. Your manuscript has now been seen by three reviewers, whose comments are appended below. As you will see that the reviewers have raised several substantive concerns about the manuscript that the editors agree should be addressed prior to further consideration of the manuscript at JGP.

We would be happy to receive an extensively revised manuscript that addresses the reviewers' concerns. Please note that we will have the revised manuscript re-reviewed, preferably by the original referees, pending their availability. Based on the scope of the requested changes, we would anticipate that the revision process will take no longer than 6 months. If additional time is required, please do let us know. In addition, please do not hesitate to contact me (via the editorial office) if you feel that a discussion of the reviewers' and editors' comments would be helpful.

Please submit your revised manuscript via the link below along with a point-by-point letter that details your responses to the editors' and reviewers' comments, as well as a copy of the text with alterations highlighted (boldfaced or underlined). If the article is eventually accepted, it would include a 'revised date' as well as submitted and accepted dates. If we do not receive the revised manuscript within one year, we will regard the article as having been withdrawn. We would be willing to receive a revision of the manuscript at a later time, but the manuscript will then be treated as a new submission, with a new manuscript number.

Please pay particular attention to recent changes to our instructions to authors in sections: Data presentation, Blinding and randomization and Statistical analysis, under Materials and Methods, as shown here: <https://rupress.org/jgp/pages/submission-guidelines#prepare>. Re-review will be contingent on inclusion of the required information (including for data added during revision) and demonstration of the experimental reproducibility of the results (i.e., all experimental data verified in at least 2 independent experiments).

Please note, JGP now requires authors to submit Source Data used to generate figures containing gels and Western blots with all revised manuscripts (when applicable). This Source Data consists of fully uncropped and unprocessed images for each gel/blot displayed in the main and supplemental figures. If your paper includes cropped gel and/or blot images, please be sure to provide one Source Data file for each figure that contains gels and/or blots along with your revised manuscript files. File names for Source Data figures should be alphanumeric without any spaces or special characters (i.e., SourceDataF#, where F# refers to the associated main figure number or SourceDataFS# for those associated with Supplementary figures). The lanes of the gels/blots should be labeled as they are in the associated figure, the place where cropping was applied should be marked (with a box), and molecular weight/size standards should be labeled wherever possible.

Source Data files will be made available to reviewers during evaluation of revised manuscripts and, if your paper is eventually published in JGP, the files will be directly linked to specific figures in the published article.

Source Data Figures should be provided as individual PDF files (one file per figure). Authors should endeavor to retain a minimum resolution of 300 dpi or pixels per inch. Please review our instructions for export from Photoshop, Illustrator, and PowerPoint here: <https://rupress.org/jgp/pages/submission-guidelines#revised>

When revising your manuscript, please be sure it is a double-spaced MS Word file and that it includes editable tables, if appropriate.

Please submit your revised manuscript via this link:
Link Not Available

Thank you for the opportunity to consider your manuscript.

Sincerely,

Jeanne

Jeanne M. Nerbonne, Ph.D.

On behalf of Journal of General Physiology

Journal of General Physiology's mission is to publish mechanistic and quantitative molecular and cellular physiology of the highest quality; to provide a best-in-class author experience; and to nurture future generations of independent researchers.

Reviewer #1 (Comments to the Authors):

In Bernas et al, the authors study the effects of PKA activation on cardiac voltage gated sodium channel NaV1.5 using (primarily) rat ventricular myocytes and the cell permeable cAMP analogue 8-CPT-cAMP (in addition to phosphatase inhibitor okadaic acid). The authors use a combination of biochemistry, microscopy, and electrophysiology to assess the impact of PKA on NaV1.5. Initial microscopy results showed increased NaV1.5 expression at the intercalated disc and lateral surface, along with increased currents as assessed by cell-attached patch clamp. The multi-hour time course of the effect of 8-CPT-cAMP indicated gene expression may be changing, and strong evidence is presented that EB1 (microtubule end binding protein 1) is upregulated with persistent 8-CPT-cAMP application (presumed PKA activation). Western blotting indicated that total NaV1.5 protein is unchanged with 8-CPT-cAMP, however biotinylation in HEK293 cells showed that cell surface NaV1.5 can be increased with 8-CPT-cAMP application. Co-IP experiments in HEK293 cells encouragingly showed EB1 and NaV1.5 may interact. The authors further show using microscopy that microtubules appear immature with 8-CPT-cAMP treatment, evidenced by a decrease in detyrosinated alpha tubulin, for reasons largely unknown and consequences not tested. The authors speculate in the discussion that this could be a path for NaV1.5 trafficking, but single particle tracking is not utilized to test this hypothesis.

The manuscript reads well with few grammatical errors and has thorough explanations in text/legends. The authors have included numerous controls and clearly have an excellent biochemical/microscopy background. The data regarding EB1 and tubulins show large, clear effects. It is less clear the physiological significance of this effect on NaV1.5, as we do not learn how NaV1.5 trafficking is increased in myocytes, if EB1 and NaV1.5 physically interact in myocytes, or how this increased cell surface NaV1.5 leads to the physiological consequences such as heart failure. Moreover, the unsuccessful biotinylation in myocytes and the unsuccessful co-IP in myocytes are significant limitations. The electrophysiological evidence for increased NaV1.5 at the surface is minimal and should be expanded to increase support for the claims of PKA increasing NaV1.5 myocyte surface expression. This is further explained in the following major and minor comments below:

Major:

- 1) The authors exclusively use 100 μ M 8-CPT-cAMP as their PKA activator. There are other published functions of 8-CPT-cAMP, including inhibition of PDE. With dosage near 100 μ M, it's likely other processes within the cell are being affected. The authors are encouraged to use forskolin to increase endogenous cAMP and test if NaV1.5 currents are increased. Demonstrating endogenous cAMP increasing NaV1.5 macroscopic currents would greatly enhance their conclusion.
- 2) In addition, using a PKA inhibitor to block this response would also be supportive as it would focus the pathway through PKA, which at present is presumed given that 8-CPT-cAMP is correlated with CREB1-S133P. While this shows PKA is activated, it doesn't show PKA as the cause of increased NaV1.5 surface expression. Increasing proof of PKA to NaV1.5 would help the authors claims.
- 3) Time courses of 8-CPT-cAMP/okadaic acid incubation: the authors convincingly demonstrate in Figure 3 that EB1 expression is maximally increased around 14 hours. However, later figures have different incubation timeframes, such as Figures 5 (12 hours) and Figure 10 (4 hours). The authors frequently use the phrase "Persistent PKA activation" in figure legends and text, which covers a wide time frame. The authors should carefully edit the manuscript such that time frames are defined every time "persistent PKA expression" is stated.
- 4) The authors should discuss why cell-attached patch clamp was used to compare current magnitudes between cell populations instead of whole-cell. While the authors that show tip resistances were similar, the uncontrollable variable is channel density under the patch. This may explain the variability in maximal current, which is large given the sample sizes. Whole-cell, normalized to call capacitance (size), may have negated this issue and should have generated currents with smoother decay functions. Whole-cell would've also offered the opportunity to more thoroughly investigate channel kinetics, such as inactivation and recovery, which could be implicated if NaV1.5 is phosphorylated. Whole-cell would've increased confidence that cell surface NaV1.5 expression was increasing if macroscopic currents were larger and there were no changes to macroscopic kinetics. These experiments would have provided significant support to increased cell surface expression in myocytes as biotinylation experiments were unsuccessful.
 - a. It's also recommended that the authors choose representative traces for their electrophysiology. Mean peak current is roughly 100 pA in PKA treated but traces appear to be closer to 200 pA.

5) Figure 2A/B: it is difficult to determine if these images correspond to the cell surface without a surface marker. Where possible, the authors should include images with cell surface markers (WGA) overlaid to help show reader what is the surface.

6) In figure 8B, the authors successfully IP GFP with a molecular weight above 250, suggestive of Nav1.5-GFP. Although it didn't IP with EB1-mRFP, this would suggest the protein was successfully extracted with 1% Triton, correct? If so, the parsimonious conclusion would be these two proteins (GFP-Nav1.5 and EB1-mRFP) do not co-IP in myocytes. Unfortunately, proximity ligation assays and FRAP are useful in showing similar cellular locations, but precise interactions are not described in these assays. Thus, at present, there isn't evidence that EB1 and Nav1.5 physically interact. If the authors wish to test this further they could try different detergents, including those which successfully extracted Navs for cryo-EM, or use site directed mutagenesis of Nav1.5 in expression systems to better validate the interaction. Otherwise, the authors should temper language in the study discussing a physical interaction.

7) The authors should explain leaky myocytes further in the context of Figure 11A. This result would be stronger in cardiomyocytes, as it was demonstrated in Figure 8 that EB1/Nav1.5 may not interact in cardiomyocytes.

Minor:

1) Define PKA in abstract. Consider using full name in title.

2) Abstract: Na should be Na⁺

3) Unneeded space between 'aging' and '1'.

4) Ca current should be "Ca²⁺". All ions should be labelled with their charges.

5) 8cpt-cAMP should be 8-CPT-cAMP

6) Methods table for antibodies: add another column called "Knockout verified" and state yes or no for each antibody used.

7) Grammar: "Fig. 2C summaries surface Nav1.5 cluster data". Summary?

8) Figure 3A, con, 14 hrs: this myocyte lacks vivid striations and is losing its boxy shape. In my experience, I would consider this an unhealthy myocyte. Do the authors have a better image they could replace it with? Also, the authors should add images for 40 hours, as this is shown in 3B.

9) Figure 5A: authors encouraged to move Fig. S2 into Fig 5. It's necessary for interpretation of lateral surface and intercalated disk.

10) 'We sought 2 alternative strategies'.

Reviewer #2 (Comments to the Authors):

Summary: The manuscript by Bernas et al. is a well written account of the effects of persistent PKA activation on Nav1.5's trafficking. The group shows that chronic PKA activation causes a significant increase in peak Nav1.5 amplitude. However, the number of Nav1.5 channels was unchanged while increased trafficking and clustering of the channel at the myocyte surface was observed. The mechanism of altered trafficking is shown to be connected to upregulated EB1 which leads to microtubule reorganization in ventricular myocytes. Specifically, the PKA activation strengthens the interactions between Nav1.5/EB1 and Nav1.5/ β -Tub at the intercalated discs of myocytes.

The manuscript provides clear evidence to support the conclusion that persistent PKA activation upregulates EB1, which in turn increases Nav1.5 trafficking to the myocyte surface. There are some missing details regarding the pathophysiology of persistent PKA activation in both the introduction and discussion. More information regarding the physiological levels and duration of PKA activation are warranted.

Specific Critiques:

1. There are too many figures crowding the main body of the manuscript. Figures 6,7,10, and 12 are not critical for understanding and could be moved to the supplementary information.

2. How long and at what levels does PKA activation actually last in a physiological setting? Is persistent activation for 6 - 15 hours within the physiological range for humans? Are the levels probed consistent with increased beta-adrenergic tone?

3. The introduction is extremely brief and does not do an adequate job of explaining the background and rationale for this study. In particular, the significance for human health is not well established and really only justified with two sentences: "Persistent β -adrenergic receptor stimulation occurs in chronically stressed hearts, such as heart failure and aging. Activation of β -adrenergic downstream effector networks including cAMP-dependent protein kinase (PKA), in concert with other neurohumoral responses, can induce electrical, structural, and metabolic remodeling in the heart, leading to protective but eventually maladaptive consequences." After these, the manuscript lists off a few roles of acute PKA activation, but further elaboration is essentially

absent, particularly regarding the potential maladaptive consequences. Are there any specific conditions that can cause the chronic stress that leads to persistent activation? What specific maladaptive consequences can the initially protective β -adrenergic responses lead to? How could those consequences affect long-term patient health and outcomes?

4. Aside from the significance for human health, the introduction is also missing background information, some of which is included later in the paper but could have been mentioned earlier. In particular, more background on EB1 and Nav1.5's interactions with microtubules would be helpful. The following description (given on page 10) could probably be moved to the introduction to partially accomplish this: "Nav1.5 channels traffic on microtubules, targeted to and anchored at their destinations by protein-protein interactions. Microtubule end binding protein 1 (EB1) is a major microtubule regulator in cardiac myocytes. Furthermore, it has been suggested that EB1 in conjunction with CLIP-associating protein 2 (CLASP2) is involved in targeting Nav1.5 to intercalated discs (ICDs). Importantly, the promoter regions of EB1 genes in human, mouse and rat contain cAMP response elements."

5. The first sentence in Abstract has unclear/confusing grammar: "Persistent PKA activation occurs in chronically stressed hearts, that can produce protective or maladaptive effects." It sounds like the chronically stressed hearts are producing the effects independent of the PKA activation.

6. On page seven, at the beginning of the second paragraph of the results section, the sentence refers to "PAK activation" when it should say "PKA activation."

7. In the section where CREB1 is first brought up, a brief explanation of the role of phosphorylated CREB1 should be included. The only explanation currently is "CREB1-S133phos enters nuclei and, together with coactivators, modulates gene expression." At the very least, include a sentence justifying why the presence of CREB1-S133phos "confirms PKA activation."

Reviewer #3 (Comments to the Authors):

In this manuscript, the authors examined the effects of persistent PKA activation on the expression, distribution and function of Nav1.5 channels in adult rat ventricular myocytes. The authors report that PKA activation increases Na⁺ current amplitude and Nav1.5 cell surface expression without affecting Nav1.5 total protein expression. Additionally, they show in HEK293 cells that this increased Nav1.5 cell surface expression is prevented by inhibition of trafficking. They also show an increased expression of EB1 protein, an increased co-immunoprecipitation of EB1 and β -tubulin with Nav1.5, and microtubule reorganization upon PKA activation. This study addressed an interesting question regarding ion channel regulation, and the authors used various methods to serve their arguments. Nevertheless, the results present several weaknesses as follow. Also, the data suggest that the regulation of Nav1.5 channels by PKA activation is not associated with a direct post-translational effect on the channel, but that it is rather due to transcriptional regulation of EB1. This mechanism is therefore of interest and should be strengthened by additional experimental demonstrations. If revealed true, it should then be made explicit in the main conclusions so no misinterpretations could be made. Finally, and of major importance, the overall organization of the manuscript and figures is too dense and difficult to follow. The number of figures is too large. Too many technical details are included outside of the materials and methods section. The presentation of data should therefore be revised thoroughly, which will help the reading and increase the impact of the work by better highlighting the various experimental approaches developed.

Major comments:

1- In the introduction, the authors discuss the previously published effects of PKA activation on the cardiac Na⁺ current, mentioning an increased peak Na⁺ current by PKA activation. It seems, however, that the effects of PKA activation on the cardiac Nav1.5 channels are not that consistent from one experiment/laboratory to another, and that no consensual conclusions can currently be made. This shall therefore be discussed accordingly and with more caution. Additionally, a lot of literature is available on the potential molecular mechanisms mediating PKA regulation and phosphorylation of Nav1.5, which shall also be discussed in the context of this study.

2- Although the effect of PKA activation on CREB1 phosphorylation is clear (Figure 1A), the number of repeated experiments should be given, and data from all repeated experiments should be quantified and illustrated as bar/individual data points graphs.

3- The authors mention about previously reported subcellular (cell center compared to cell end) differences in amplitude and gating properties of Na⁺ currents in ventricular myocytes. Although differences in peak amplitudes were not observed here, do the authors observe differences in voltage-dependence of steady-state inactivation as previously reported in several publications (PMID: 21767519 and 29222390)? Whatever the findings, it would be interesting to show these data as well as the other Na⁺ current gating (activation, inactivation) properties in a table. Additionally, in Figure 1C, it would be preferable to show, on the top of bars, individual data points in order to show data point distribution in each condition.

4- The duration of PKA activation (from 6 to 15 hours) is large and variable from experiments to experiments, and also sometimes within each single set of experiments, which is somehow worrisome as a lot of different (transcriptional and/or post-translational) effects on channel expression and/or gating could occur during short-, mid- or long-time frames. It would therefore be interesting to distinguish the effects obtained after different durations, at least after short and long duration. This will give the reader a better idea on the time frame of the different mechanisms involved in the regulation of interest.

5- The use of adenoviruses expressing GFP-Nav1.5 under the control of the CMV promoter is worrisome. It is indeed well-recognized that the CMV promoter contains cAMP response elements (CRE, PMID: 25612069) that are responsive to CREB, and therefore to PKA activation. Therefore, one could always wonder whether the observed regulation of Na⁺ channel is the

consequence of CMV promoter activation? Information concerning whether this construct has been used in the experiments or whether the experiments were performed on endogenous cardiac channels (which is preferable in this context) is not always clearly defined and should therefore be made clearer in each figure. Alternatively, or additionally, this could also be defined in the methods section. Additionally, the use of this adenovirus refers to reference #8. Annotation of this reference is not correct. This paper was published in Nature Communications, not in eLife. Also, although this adenovirus has been used in this later study, details concerning its construction and content cannot be found in the publication, which requires that the authors report it here in the present manuscript.

6- In Figures 1D and 4A, the authors should show only one representative figure (images of repeated experiments will be found in the Source Data file available from JGP). In addition, the bar graph should show bars and individual data points for each condition, which allow evaluation of variability in each condition, rather than a ratio between the two conditions.

7- The authors demonstrate that the PKA regulation of Nav1.5 expression depends on an increased EB1 protein expression. They also suggest that this increased expression may be associated with an increased binding of CREB1 to the EB1 gene promoter region. To further the argument that the mechanism involved is indeed a transcriptional regulation, and not a post-translational regulation of the channel per se, the authors should strengthen this argument with additional experimental demonstrations. Notably, the authors should at least analyze the expression levels of EB1 (and Nav1.5) transcripts in CON and PKA myocytes. If revealed to be true, this finding should be emphasized as it is the mechanism of interest involved in the regulation of Nav1.5. This will avoid confounding the reader into misinterpretation.

8- In Figure 4B, data should be quantified and illustrated. Also, images of b-Tub and deY a-Tub should be merged to show changes in microtubule organization. Also, no images of CON myocytes are shown in Figure 4D. Are there any microtubule loops observed in CON myocytes? These data should also be quantified and illustrated.

9- In the calculation of degrees of enrichment on lateral surface and at ICD (Figures 6 and S2), why do the authors divide % pixel contents by % area, rather than simply divide pixel contents by area?

10- In paragraph 3 of the Results section, the authors say that Nav1.5 and EB1, but no b-Tub, were more enriched along the lateral surface, and that is consistent with image data presented in Figures 1E, 2 and 4B. However, illustration and especially quantification of these data are not provided in these related Figures.

11- In the experiments presented in Figure 8A, the authors conclude an increased interaction of EB1 with Nav1.5 upon PKA activation. However, Nav1.5 expression and therefore immunoprecipitation are increased upon PKA activation, which will simply allow greater co-immunoprecipitation of EB1. Also, importantly, this quantification shall be made from the same blot and blot exposure, which is unclear in the presented figure because of the line drowned in between the CON and PKA conditions.

12- In Figure 10, images of Nav1.5 immunoprecipitation, as well as of EB1 immunoprecipitation are required to make sure the IPs worked. Also, it would be more convincing to show a representative co-immunoprecipitation in which the expression level of b-tubulin is similar in CON and PKA conditions. Again, in Figure 10B, the authors should show the plots in both CON and PKA conditions, rather than showing a PKA/CON ratio, so the variability in each condition is illustrated.

13- In Figure 11A, because of the fact that PKA activation can increase total expression of GFP-Nav1.5 through transcriptional regulation (CMV promoter activation), expression of both total and cell surface expression of GFP-Nav1.5 should be shown in both representative blot images and in the graph. This is in fact the case in each condition (with and without chloroquine). Again, values and data points in each CON and PKA condition should be shown in the graph. Also, in order to demonstrate an effect of PKA activation on Nav1.5 trafficking, the authors should use other tools (such as brefeldin A for example...) in addition to chloroquine which not only impacts trafficking but most certainly many other cellular mechanisms, such lysosomal degradation, protein recycling, etc... The authors shall also pay attention to the vocabulary used as both trafficking and recycling are used in the text. Each mechanism is specific, involves distinct cellular and molecular pathways, and should therefore not be mixed up.

14- Too much speculation on mechanisms are provided from experiments performed in Figures 11B and 12. I would recommend removing these data from the manuscript.

Minor comments:

1- In the abstract, the authors use the term "recycling" to talk about the trafficking of proteins from intracellular compartments to plasma membrane. This is incorrect. They should use one or the other depending on findings (see above).

2- In the abstract, the authors cannot say that Nav1.5 "directly" interacts with EB1 and b-tubulin because they cannot exclude the possibility, using co-immunoprecipitation experiments, that the observed interactions are mediated through intermediary proteins.

3- The description of the cell-attached patch-clamp recording method is confusing. Indeed, the voltage-clamp values should be expressed using the usual, intracellular versus extracellular potentials convention, rather than the cell-attached invert convention. This will help comprehension by non-electrophysiologist readers.

4- At the end of paragraph 2 in the Results section, the authors used "(see below)" to discuss findings described underneath the text. This should be changed to the associated Figure.

5- The increased % ICD area occupied by clusters and cluster size shall be better illustrated in Figure 6A. In other words, images in Figure 6A are not representative of the quantified data in Figure 6B.

6- Figures 8C and 8D are not essential in the main Figures and shall be included as Supplemental Figures.

7- In Figure 8E, is it really useful to double the Nav1.5 and EB1 graphs showing fluorescence intensity (end combined with center; and center and end separated)?

8- Figure 11C should stand by itself.

Authors' response to reviewers of JGP submission 202313436

PKA activation upregulates EB1, reorganizes microtubules and promotes Nav1.5 to cardiomyocyte surface

Tytus Bernas, John Seo, Zachary T. Wilson, Isabelle Deschenes,
Christiane Morecock, Jinze Liu, and G.-N. Tseng

We thank all 3 Reviewers and the Editor for their careful consideration of our manuscript. We have conducted new experiments and added the following data:

1. Forskolin (activating adenylate cyclases and producing native cAMP) mimicked the effect of PKA activation, and a membrane-permeable PKA peptide inhibitor (PKI-14-22 amide, myristoylated) reduced the effect of PKA activation without affecting CON myocytes (new Fig. 14).
2. Wheat germ agglutinin 'WGA' was used as a cell membrane marker to validate cell surface Nav1.5 cluster identification (Fig. S1).
3. The effect of persistent PKA activation on CREB1-S133^P band intensity was quantified with statistical analysis (Fig. 1A).
4. The effects of persistent PKA activation on the transcript levels of EB1 and Nav1.5 (as part of an RNA-seq experiment) were quantified (Fig. S2).
5. Co-immunoprecipitation experiments in HEK293 cells presented in Fig. 9A and Fig. 11 are strengthened with new data.

We have also made the following changes:

1. Adding more data or analysis requested by the reviewers (details in our specific response to the reviewers).
2. Uploading 2 videos of 3D views of nCadherin, Nav1.5, β -Tubulin and EB1 in CON and PKA myocytes. This is the most efficient way to illustrate how persistent PKA activation affects the distribution and interrelationship of these native proteins in myocytes' ICDs.
3. In response to the critique by both reviewers 2 and 3 (*the manuscript was too crowded with figures and technical details*), we have removed technical details from the main text and kept them either in the Methods section (if generally applicable to all experiments) or in figure legends (specific for the presented experiments). This change streamlines the main text and makes it much more readable.
4. The manuscript title was shortened, in accordance with JGP guidelines.
5. Two new authors were added (responsible for RNA-seq analysis).
6. Per JGP policy, we have uploaded all Source data (full, uncropped immunoblot images) for those presented in the figures.

Below is our point-by-point response to the *Reviewers' comments*.

Reviewer #1

Major:

1. *The authors are encouraged to use forskolin to increase endogenous cAMP... In addition, using a PKA inhibitor to block this response would also be supportive as it would focus the pathway through PKA.*

Thank you for the suggestions. We have followed them and the new are presented in Fig. 14.

2. *Time course of 8CPT-cAMP/okadaic acid incubation: The authors convincingly demonstrate in Fig. 3 that EB1 expression is maximally increased around 14 hours. However, later figures have different incubation timeframes, such as Figures 5 (12 hours [on myocytes]) and Figure 10 (4 hours [on HEK cells]). The authors frequently use the phrase "Persistent PKA activation" in figure legends*

and text, which covers a wide time frame. The authors should carefully edit the manuscript such that time frames are defined every time "persistent PKA expression" is stated.

We have specified the 8CPT-cAMP/okadaic acid incubation times in all figure legends or related text. The incubation time differed between myocyte and HEK experiments for the following reasons.

- a. In myocytes, the EB1 peaking time of 14 hr shown in Fig. 3 was based on the time points we chose in this experiment. The EB1 upregulation, as well as changes in Nav1.5 and microtubule distribution, is expected to occur over a time window of 4 to >15 hr. Therefore, the incubation time was 12-15 hr in most myocyte experiments. The only exception is Fig. 14 (incubation time 4 hr), because we observed myocyte deterioration in forskolin medium after 4 hr.
- b. A pilot experiment in HEK cells showed that cell surface GFP-Nav1.5 was increased when the 8CPT-cAMP/Okadaic acid incubation time was prolonged from 1 to 4 hr, but was not further increased after 6 hr of incubation. Therefore, the incubation time was 4-6 hr in HEK experiments.

3. *The authors should discuss why cell-attached patch clamp was used to compare current magnitudes between cell populations instead of whole-cell.*

I used cell-attached patch clamp to quantify myocyte surface Nav channels for 2 reasons. First, this recording conditions provided much better temporal and spatial control of membrane voltage, thus much higher fidelity of Nav channel recording, than whole myocyte recording. Adult rat ventricular myocytes are large (120-180 pF), and the Nav current amplitudes are high (with $[Na]_o$ 10 mM, the maximal I_{Na} peak amplitude could be 10 nA), making it extremely challenging to achieve appropriate spatio-temporal control of membrane voltage. Indeed, under optimal conditions in my hand (1 M Ω pipette tip, 98% R_s compensation, $[Na]_o$ 10 mM, room temperature), I still saw signs of losing spatial control of whole myocyte membrane voltage (abrupt increase in I_{Na} peak amplitude upon small depolarization above the I_{Na} threshold).

Second, a previous study reported that in adult rat ventricular myocytes, cell attached patch clamp recordings showed higher I_{Na} amplitudes at cell end than at cell center (Lin et al., 2011). Therefore, I was interested in knowing whether PKA's effect on myocyte surface Nav1.5 was region-dependent.

While the authors show that the tip resistances were similar, the uncontrollable variable is channel density under the patch. This may explain the variability in maximal current, which is large given the sample sizes.

This is an important critique. Patch clamp recording took time: on average, I could do ~ 5 patches within a one-hour time limit after myocytes were removed from the incubator and placed in a cell chamber on the microscope stage superfused with room temperature Tyrode's solution. This one-hour time limit was set to minimize the confounding factor of dissipating PKA effects on Nav channel distribution. I began patch clamping CON myocytes roughly 4 hr after myocyte isolation, and alternated recordings between CON and PKA myocytes over 10 - 12 hr total patch clamp time. This long patch clamp procedure and the inevitable variations in 8CPT-cAMP/okadaic acid incubation time are expected to contribute to the variations in reported maximal I_{Na} peak amplitude, especially in the PKA group.

Whole-cell would've also offered the opportunity to more thoroughly investigate channel kinetics, such as inactivation and recovery, which could be implicated if Nav1.5 is phosphorylated..

I agree. In adult rabbit ventricular myocytes, β -adrenergic stimulation with 10 μ M isoproterenol increased I_{Na} by 2 mechanisms. One was PKA-mediated and involved a modest acceleration of I_{Na} inactivation (τ of inactivation shortened from 3.87 to 2.57 ms) (Matsuda, Lee and Shibata, 1992). The fact that I did not observe any change in τ of inactivation of maximal I_{Na} between PKA and CON myocytes (Fig. 1C, lower) is consistent with the notion that the increase in I_{Na} amplitude was not related to PKA phosphorylation of the Nav channel.

It's also recommended that the authors choose representative traces for their electrophysiology. Mean peak current is roughly 100 pA in PKA treated but traces appear to be closer to 200 pA.

The current traces shown in Fig. 1B were chosen because they were obtained on the same day (November 1, 2022), at 5 - 6 pm (CON myocytes) and 11 pm - midnight (PKA myocytes).

4. *Figure 2A/B: it is difficult to determine if these images correspond to the cell surface without a surface marker. Where possible, the authors should include images with cell surface markers (WGA) overlaid to help show reader what is the surface.*

The new experiment presented in Fig. S1 addresses this point. Please note that although wheat germ agglutinin (WGA) staining may appear as a smooth line along cell edges when imaging z-planes inside myocytes, WGA manifests a distinct distribution pattern when viewed on the myocyte surface. This pattern reflects the uneven distribution of cell surface glycosylated proteins.

5. *In figure 8B, the authors successfully IP GFP with a molecular weight above 250, suggestive of Nav1.5-GFP. Although it didn't IP with EB1-mRFP, this would suggest the protein was successfully extracted with 1% Triton, correct? If so, the parsimonious conclusion would be these two proteins (GFP-Nav1.5 and EB1-mRFP) do not co-IP in myocytes.*

You are correct that the extremely faint band at > 250 kDa in WCL lanes means we could extract minute amount of GFP-Nav1.5 using 1% Triton. This band was prominent in the IP lanes because we were loading IP samples equivalent to 20 fold WCL protein in these lanes. We believe the low level of GFP-Nav1.5 extraction with 1% Triton prevented its detection in EB1-mRFP immunoprecipitate. This motivated us to use other means to detect GFP-Nav1.5/EB1-mRFP interactions.

Unfortunately, proximity ligation assays and FRAP are useful in showing similar cellular locations, but precise interactions are not described in these assays. .. the authors should temper language in the study discussing a physical interaction.

We agree: proximity ligation assay (PLA) detects proteins 40-100 nm apart (Soderberg et al., 2008). Therefore, we described our results as GFP-Nav1.5/EB1-mRFP interactions, instead of direct physical binding.

6. *The authors should explain leaky myocytes further in the context of Figure 11A.*

Leaky myocytes refer to those that are dying or dead in the culture dishes, which inevitably occurs during myocyte isolation and culture. Shown on the right are immunoblot images from a pilot experiment on adult rat ventricular myocytes. The purpose was to see whether we could detect increase in cell surface native Nav1.5 by biotinylation after PKA activation for 15 hr. The α -actin image (middle row) clearly showed contamination of cytosolic proteins in the biotinylated fraction. This motivated us to switch to HEK293 cells for the biotinylation experiments.

Minor:

1. *Define PKA in abstract.* This is done. *Consider using full name in title.* This cannot be done because the title is limited to 100 characters plus spaces.
2. *Abstract: Na should be Na⁺* Done throughout.
3. *Unneeded space between 'aging' and '1'.* Corrected.
4. *Ca current should be "Ca²⁺". All ions should be labelled with their charges.* Done throughout.
5. *8cpt-cAMP should be 8CPT-cAMP* Done throughout.
6. *Methods table for antibodies: add another column called "Knockout verified" and state yes or no for each antibody used.* Done.
7. *Grammar: "Fig. 2C summaries surface Nav1.5 cluster data".* Corrected.

8. *Figure 3A, con, 14 hrs: this myocyte lacks vivid striations and is losing its boxy shape. In my experience, I would consider this an unhealthy myocyte. Do the authors have a better image they could replace it with? Also, the authors should add images for 40 hours, as this is shown in 3B.* Both are done.
9. *Figure 5A: authors encouraged to move Fig. S2 into Fig 5. It's necessary for interpretation of lateral surface and intercalated disk.* Done.
10. *'We sought 2 alternative strategies'.* Corrected.

Reviewer 2

Specific Critiques:

1. *There are too many figures crowding the main body of the manuscript. Figures 6,7,10, and 12 are not critical for understanding and could be moved to the supplementary information.*
We have removed technical details from the main text and kept them either in the Methods section (if generally applicable to all experiments) or in figure legends (specific for the presented experiments). This streamlines the main text and makes it much more readable.
2. *How long and at what levels does PKA activation actually last in a physiological setting? Is persistent activation for 6 - 15 hours within the physiological range for humans? Are the levels probed consistent with increased beta-adrenergic tone?*
These are important questions. cAMP production and PKA activation following β -adrenergic stimulation in myocytes are known to be compartmentalized (reviewed in (Agarwal et al., 2022)). We are taking a reductionist approach to understand how PKA activation for hours can affect Nav1.5 distribution and function in adult ventricular myocytes and to explore potential mechanisms.
3. *The introduction is extremely brief and does not do an adequate job of explaining the background and rationale for this study. In particular, the significance for human health is not well established.*
Thank you for these important comments. At the time of writing, we had little information beyond what our experiments had told us. Now we have finished comprehensive transcriptomic and proteomic experiments on myocytes exposed to PKA activation for 15 hr and time control (N=5 each). These data will allow us to speculate protective and potentially maladaptive consequences of PKA activation in myocytes.
4. *Aside from the significance for human health, the introduction is also missing background information, some of which is included later in the paper but could have been mentioned earlier.*
Thank you for the suggestion. The description of relationship among Nav1.5, microtubules and EB1 is now in the *Introduction*.
5. *The first sentence in Abstract has unclear/confusing grammar: "Persistent PKA activation occurs in chronically stressed hearts, that can produce protective or maladaptive effects." It sounds like the chronically stressed hearts are producing the effects independent of the PKA activation.*
This sentence has been modified to clarify the point: During chronic stress persistent activation of cAMP-dependent protein kinase (PKA) occurs, that can contribute to protective or maladaptive changes in the heart.
6. *On page seven, at the beginning of the second paragraph of the results section, the sentence refers to "PAK activation" when it should say "PKA activation."*
This is corrected. Thank you.
7. *In the section where CREB1 is first brought up, a brief explanation of the role of phosphorylated CREB1 should be included.*

We have added this information to the beginning of the *Results* section: PKA can phosphorylate serine 133 of cAMP response element binding protein 1 (CREB1). Phosphorylated CREB1 (CREB1-S133^P) then enters nuclei and regulates gene expression.

Reviewer #3

General comments

1. *The data suggest that the regulation of Nav1.5 channels by PKA activation is not associated with a direct post-translational effect on the channel, but that it is rather due to transcriptional regulation of EB1. This mechanism is therefore of interest and should be strengthened by additional experimental demonstrations. If revealed true, it should then be made explicit in the main conclusions so no misinterpretations could be made.*

We have followed this reviewer's suggestion in making our findings explicit in the *Summary* statement. This is based on the following data:

- a. Newly finished RNA-seq experiment showed that EB1 transcript was markedly elevated in PKA vs CON myocytes (fold change = 4.52, FDR = 4.82E-07, Fig. S2).
 - b. Immunoblot data showed EB1 protein was increased in PKA vs CON myocytes (cytosolic fraction, Fig. 4A).
 - c. There was no change in the time constant (τ) of inactivation of maximal I_{Na} between PKA and CON myocytes (Fig. 1C). This is different from the shortening of τ of inactivation in acute PKA mediated increase in I_{Na} in adult rabbit ventricular myocytes (Matsuda, Lee and Shibata, 1992).
2. *Of major importance, the overall organization of the manuscript and figures is too dense and difficult to follow. The number of figures is too large. Too many technical details are included outside of the materials and methods section. The presentation of data should therefore be revised thoroughly, which will help the reading and increase the impact of the work by better highlighting the various experimental approaches developed.*

Thank you for this important comment. We have removed technical details from the main text and kept them either in the Methods section (if generally applicable to all experiments) or in figure legends (specific for the presented experiments). This streamlines the main text and makes it much more readable.

Major:

1. *In the introduction, the authors discuss the previously published effects of PKA activation on the cardiac Na⁺ current, mentioning an increased peak Na⁺ current by PKA activation. It seems, however, that the effects of PKA activation on the cardiac Nav1.5 channels are not that consistent from one experiment/laboratory to another, and that no consensual conclusions can currently be made. This shall therefore be discussed accordingly and with more caution.*

We have added a whole new section in the beginning of 'Discussion' to address this issue. Its subtitle is '1. Effects of persistent PKA activation are distinctly different from those of acute PKA activation on Nav1.5 in cardiac myocytes'. This new section is highlighted by gray shading for easy detection.

Additionally, a lot of literature is available on the potential molecular mechanisms mediating PKA regulation and phosphorylation of Nav1.5, which shall also be discussed in the context of this study.

We have cited an important and comprehensive proteomics study of phosphorylation sites in the Nav1.5 channel (Lorenzini.. Marionneau, J Gen Physiol 2021;153:1-23).

2. *Although the effect of PKA activation on CREB1 phosphorylation is clear (Figure 1A), the number of repeated experiments should be given, and data from all repeated experiments should be quantified and illustrated as bar/individual data points graphs.*

These new data are summarized in Fig. 1A.

3. *The authors mention about previously reported subcellular (cell center compared to cell end) differences in amplitude and gating properties of Na⁺ currents in ventricular myocytes. Although differences in peak amplitudes were not observed here, do the authors observe differences in voltage-dependence of steady-state inactivation as previously reported in several publications (PMID: 21767519 (Lin et al., 2011) and 29222390 (Rivaud et al., 2017))? Whatever the findings, it would be interesting to show these data as well as the other Na⁺ current gating (activation, inactivation) properties in a table.*

I do not have any kinetic data on Nav channels in PKA or CON myocytes, other than the lack of change in τ of inactivation of maximal I_{Na} (Fig. 1C, lower panel). The issues of patch clamp experiments are discussed in details in response to Reviewer 1, point 3, above.

I would humbly state that this study is about the effects of persistent PKA activation on Nav1.5 distribution in adult ventricular myocytes and possible underlying mechanism.

Additionally, in Figure 1C, it would be preferable to show, on the top of bars, individual data points in order to show data point distribution in each condition.

The box plot in Fig. 1C has been switched to a bar graph with individual data points as requested.

4. *The duration of PKA activation (from 6 to 15 hours) is large and variable from experiments to experiments, and also sometimes within each single set of experiments, which is somehow worrisome as a lot of different (transcriptional and/or post-translational) effects on channel expression and/or gating could occur during short-, mid- or long-time frames.*

The issue of different PKA activation times was also raised by Reviewer 1. Please see our detailed response above (to major concern 2).

5. *The use of adenoviruses expressing GFP-Nav1.5 under the control of the CMV promoter is worrisome. It is indeed well-recognized that the CMV promoter contains cAMP response elements (CRE, PMID: 25612069) that are responsive to CREB, and therefore to PKA activation. Therefore, one could always wonder whether the observed regulation of Na⁺ channel is the consequence of CMV promoter activation? Information concerning whether this construct has been used in the experiments or whether the experiments were performed on endogenous cardiac channels (which is preferable in this context) is not always clearly defined and should therefore be made clearer in each figure. Alternatively, or additionally, this could also be defined in the methods section.*

We have added one new paragraph in the 'Technical consideration' of Discussion: The cytomegalovirus (CMV) promoter in the Adv-GFP-Nav1.5 construct has three cAMP response elements. Incubation with 8CPT-cAMP will increase GFP-Nav1.5 protein level, interfering with experiments designed to study Nav1.5 distribution and function. Therefore in all myocyte experiments we focused on native Nav1.5, except those presented in the following figures where an increase in GFP-Nav1.5 protein level was not expected to affect the conclusions: Fig. 9B, Fig. 9E, Fig. 10, Fig. 12B, and Fig. 13A.

Additionally, the use of this adenovirus refers to reference #8. Annotation of this reference is not correct. This paper was published in Nature Communications, not in eLife. Also, although this adenovirus has been used in this later study, details concerning its construction and content cannot be found in the publication, which requires that the authors report it here in the present manuscript.

The GFP-Nav1.5 construct originated from Dr. Abriel's lab, and this has been properly noted with permission from Dr. Abriel. It has GFP directly fused to the N-terminus of the Nav1.5 isoform, hH1a (Uniprot ID Q14524). There was no other modification made to hH1a. This information has been added to the Materials and Methods section. Reference #8 was removed.

6. *In Figures 1D and 4A, the authors should show only one representative figure (images of repeated experiments will be found in the Source Data file available from JGP). In addition, the bar graph should show bars and individual data points for each condition, which allow evaluation of variability in each condition, rather than a ratio between the two conditions.*

The immunoblot experiments summarized in Figs. 1A, 1D and 4A were run on different days because they were independent experiments. In each of the experiments, we controlled the amount of protein loading to be equal between CON and PKA samples, all the lanes in the membrane were exposed to the same 1st and 2nd Abs solutions, and subject to ECL simultaneously. However, densitometry readouts would vary among these independent experiments and cannot be directly pooled. To pool these data for statistical analysis, we normalized band intensity in the PKA lane vs that in the CON lane of the same experiment. Therefore, the meaningful data are the PKA : CON ratio. Plotting data as separate CON and PKA conditions will produce CON = 1, and PKA = PKA:CON ratio.

7. *The authors demonstrate that the PKA regulation of Nav1.5 expression depends on an increased EB1 protein expression. They also suggest that this increased expression may be associated with an increased binding of CREB1 to the EB1 gene promoter region. To further the argument that the mechanism involved is indeed a transcriptional regulation, and not a post-translational regulation of the channel per se, the authors should strengthen this argument with additional experimental demonstrations. Notably, the authors should at least analyze the expression levels of EB1 (and Nav1.5) transcripts in CON and PKA myocytes. If revealed to be true, this finding should be emphasized as it is the mechanism of interest involved in the regulation of Nav1.5. This will avoid confounding the reader into misinterpretation.*

We have finished an RNA-seq experiment on CON and PKA myocytes (n=5 per group) by the VCU Genomics core, and the data have been analyzed by Christiane Morecock and Jinze Liu (Bioinformatics). The transcript of EB1 was markedly elevated while that of Nav1.5 was modestly reduced in PKA myocytes. These new data are presented in Fig. S2.

The observation that the Nav1.5 protein level was not altered in PKA myocytes despite the modest decrease in the mRNA could be explained by the stability of Nav1.5 protein in myocytes: in canine ventricular myocytes the Nav1.5 protein has a 'functional half-life' of ~ 35 hr (Maltsev et al., 2008).

8. *In Figure 4B, data should be quantified and illustrated. Also, images of β -Tub and deY α -Tub should be merged to show changes in microtubule organization. Also, no images of CON myocytes are shown in Figure 4D. Are there any microtubule loops observed in CON myocytes? These data should also be quantified and illustrated.*

Fig. 4B now includes merged β -tubulin and deY α -tubulin signals as requested.

Microtubule looping is rarely seen in CON myocytes but is a frequent observation in PKA myocytes. Fig. 4D now includes both CON and PKA myocytes to illustrate this point.

I am not aware of any program suitable for quantifying microtubule looping.

9. *In the calculation of degrees of enrichment on lateral surface and at ICD (Figures 6 and S2), why do the authors divide % pixel contents by % area, rather than simply divide pixel contents by area?*

If we simply calculate % of pixel contents at ICD and along the lateral surface, the results will be sensitive to the ICD volume and the cell width, respectively. Normalizing % pixel contents by % of cellular area at ICD or cell boundary avoids this issue.

10. *In paragraph 3 of the Results section, the authors say that Nav1.5 and EB1, but no β -Tub, were more enriched along the lateral surface, and that is consistent with image data presented in Figures*

1E, 2 and 4B. However, illustration and especially quantification of these data are not provided in these related Figures.

Image data presented in Figures 1E, 2 and 4B illustrate the point, and summary quantification is presented in Fig. 6.

11. In the experiments presented in Figure 8A, the authors conclude an increased interaction of EB1 with Nav1.5 upon PKA activation. However, Nav1.5 expression and therefore immunoprecipitation are increased upon PKA activation, which will simply allow greater coimmunoprecipitation of EB1. Also, importantly, this quantification shall be made from the same blot and blot exposure, which is unclear in the presented figure because of the line drowned in between the CON and PKA conditions.

The data presented in Fig. 8A (now Fig. 9A) were from the same blot and blot exposure. Full uncropped blot images are uploaded as shown below:

12. In Figure 10, images of Nav1.5 immunoprecipitation, as well as of EB1 immunoprecipitation are required to make sure the IPs worked. Also, it would be more convincing to show a representative co-immunoprecipitation in which the expression level of β -tubulin is similar in CON and PKA conditions. Again, in Figure 10B, the authors should show the plots in both CON and PKA conditions, rather than showing a PKA/CON ratio, so the variability in each condition is illustrated.

New data are added to Fig 11 (original Fig. 10) to show that Nav1.5 and EB1 immunoprecipitation were both specific and efficient, and the expression of β -tubulin was even between CON and PKA myocytes. New graphs of [+]IP/WCL of both CON and PKA myocytes are included as requested.

13. In Figure 11A, because of the fact that PKA activation can increase total expression of GFP Nav1.5 through transcriptional regulation (CMV promoter activation), expression of both total and cell surface expression of GFP-Nav1.5 should be shown in both representative blot images and in the graph. This is in fact the case in each condition (with and without chloroquine). Again, values and data points in each CON and PKA condition should be shown in the graph.

These data are now included as requested by this reviewer in Fig. 13A (original Fig. 11A).

Also, in order to demonstrate an effect of PKA activation on Nav1.5 trafficking, the authors should use other tools (such as brefeldin A for example...) in addition to chloroquine which not only impacts trafficking but most certainly many other cellular mechanisms, such lysosomal degradation, protein recycling, etc...

Thank you for the suggestion. We do plan to pursue the pathways of Nav1.5 trafficking in myocytes under control conditions and after PKA activation in future experiments.

The authors shall also pay attention to the vocabulary used as both trafficking and recycling are used in the text. Each mechanism is specific, involves distinct cellular and molecular pathways, and should therefore not be mixed up.

Thank you for the comment. We have revised all 'recycling' to 'trafficking'.

14. *Too much speculation on mechanisms are provided from experiments performed in Figures 11B and 12. I would recommend removing these data from the manuscript.*

We respectfully disagree, and have retained them in the manuscript (now Fig. 12B and Fig. 13).

Minor:

1. *In the abstract, the authors use the term "recycling" to talk about the trafficking of proteins from intracellular compartments to plasma membrane. This is incorrect. They should use one or the other depending on findings (see above).*

Done and tone down.

2. *In the abstract, the authors cannot say that Nav1.5 "directly" interacts with EB1 and b-tubulin because they cannot exclude the possibility, using co-immunoprecipitation experiments, that the observed interactions are mediated through intermediary proteins.*

We agree that co-immunoprecipitation may occur through intermediate proteins, and thus have toned down the language.

3. *The description of the cell-attached patch-clamp recording method is confusing. Indeed, the voltage-clamp values should be expressed using the usual, intracellular versus extracellular potentials convention, rather than the cell-attached invert convention. This will help comprehension by non-electrophysiologist readers.*

Since during cell-attached patch recordings the myocytes were superfused with Tyrode's solution that contained 4 mM [K⁺], I did not have data on the true membrane potential.

4. *At the end of paragraph 2 in the Results section, the authors used "(see below)" to discuss findings described underneath the text. This should be changed to the associated Figure.*

We apologize for not understanding that this means.

5. *The increased % ICD area occupied by clusters and cluster size shall be better illustrated in Figure 6A. In other words, images in Figure 6A are not representative of the quantified data in Figure 6B.*

We respectfully disagree. The cluster density and size for all proteins (Nav1.5, EB1 and β -Tub) in the ICD areas were clearly higher in PKA than CON myocytes, shown in the representative cases of Fig. 7A (original Fig. 6A).

6. *Figures 8C and 8D are not essential in the main Figures and shall be included as Supplemental Figures.*

We kept these two panels because they are important for the points we want to make.

7. *In Figure 8E, is it really useful to double the Nav1.5 and EB1 graphs showing fluorescence intensity (end combined with center; and center and end separated)?*

In Fig. 9E (original Fig. 8E), the upper row shows superimposed cell end and cell center time courses from the specific experiment depicted above. The lower row shows averaged time courses from multiple experiments, with SE bars and superimposed curve fitting. If I combined the cell end and cell center average time courses, it would be difficult to highlight the time point of 2 min after photobleaching (as a measure of FRAP rates used in the upper right panel).

8. *Figure 11C should stand by itself.*

We kept this panel in Fig. 12 (original Fig. 11) to reduce the number of figures.

Literature cited

Agarwal, S.R., R.T. Sherpa, K.S. Moshal, and R.D. Harvey. 2022. Compartmentalized cAMP signaling in cardiac ventricular myocytes *Cellular Signalling*. 89:110172.

Lin, X., N. Liu, J. Lu, J. Zhang, J.M.B. Anumonwo, L.L. Isom, G.I. Fishman, and M. Delmar. 2011. Subcellular heterogeneity of sodium current properties in adult cardiac ventricular myocytes. *Heart Rhythm*. 8:1923-1930.

Maltsev, V.A., J.W. Kyle, S. Mishra, and A.I. Undrovinas. 2008. Molecular identity of the late sodium current in adult dog cardiomyocytes identified by Nav1.5 antisense inhibition. *Am J Physiol*. 295:H667-H676.

Matsuda, J.J., H. Lee, and E.F. Shibata. 1992. Enhancement of rabbit cardiac sodium channels by β -adrenergic stimulation. *Circ Res*. 70:199-207.

Rivaud, M.R., E. Agullo-Pascual, X. Lin, A. Leo-Macias, M. Zhang, E. Rothenberg, C.R. Bezzina, M. Delmar, and C.A. Remme. 2017. Sodium channel remodeling in subcellular microdomains of murine failing cardiomyocytes. *J Am Heart Assoc*. 6:e007622.

Soderberg, O., K.-J. Leuchowius, M. Gullberg, M. Jarvius, I. Weibrecht, L.-G. Larsson, and U. Landegren. 2008. Characterizing proteins and their interactions in cells and tissues using the in situ proximity ligation. *Methods*. 45:227-232.

October 10, 2023

Dr. Gea-Ny Tseng
Virginia Commonwealth University
Physiology & Biophysics
1101 East Marshall Street
Sanger 3-042a
Richmond, Virginia 23298

Re: 202313436R1

Dear Gea-Ny,

We have now received the reviews of your manuscript, titled "PKA activation upregulates EB1, reorganizes microtubules and promotes Nav1.5 to cardiomyocyte surface" to JGP. The comments of the three reviewers are appended below. As you will see, the reviewers recognized the improvements made in the manuscript in response to the previous critiques. They have, however, also identified a number of issues that require additional attention. The editors agree with the comments and suggestions of the reviewers and, in addition, request that you include the term "persistent" before PKA activation and change 'promotes Nav1.5 to cardiomyocyte surface' to "promotes increased cell surface expression of Nav1.5" in the title. The final title would then be: "Persistent PKA activation upregulates EB1, reorganizes microtubules and promotes increased cell surface expression of Nav1.5".

We hope that you will be able to submit a revised manuscript that addresses the specific concerns identified. Based on the scope of the requested changes, we would expect that the revision process will take no longer than 2 months. If you find you need additional time, however, please let us know. In addition, please do not hesitate to contact me (via the editorial office) if you feel that a discussion of the reviewers' and editors' comments would be helpful.

Please submit your revised manuscript via the link below, along with a point-by-point letter that details your response to the reviewers' and editors' comments, as well as a copy of the text with alterations highlighted (boldfaced or underlined). If the article is eventually accepted, it would include a 'revised date' as well as submitted and accepted dates. If we do not receive the revised manuscript within one year, we will regard the article as having been withdrawn. We would be willing to receive a revision of the manuscript at a later time, but the manuscript will then be treated as a new submission, with a new manuscript number.

Please pay particular attention to recent changes to our instructions to authors in the following sections: Data presentation, Blinding and randomization and Statistical analysis, under Materials and Methods, as shown here: <https://rupress.org/jgp/pages/submission-guidelines#prepare>. Re-review will be contingent on inclusion of the required information (including for data added during revision) and demonstration of the experimental reproducibility of the results. Also, To improve the reproducibility of published content, we have partnered with SciScore. Authors are prompted in eJP to copy and paste the Materials and Methods section of their manuscript for a SciScore assessment when submitting their revised manuscript. Authors are encouraged (not required) to further revise their Materials and Methods if the SciScore is below 4. More information can be found here: <https://rupress.org/jgp/pages/submission-guidelines#sciscore>.

Please note, JGP now requires authors to submit Source Data used to generate figures containing gels and Western blots with all revised manuscripts (when applicable). This Source Data consists of fully uncropped and unprocessed images for each gel/blot displayed in the main and supplemental figures. If your paper includes cropped gel and/or blot images, please be sure to provide one Source Data file for each figure that contains gels and/or blots along with your revised manuscript files. File names for Source Data figures should be alphanumeric without any spaces or special characters (i.e., SourceDataF#, where F# refers to the associated main figure number or SourceDataFS# for those associated with Supplementary figures). The lanes of the gels/blots should be labeled as they are in the associated figure, the place where cropping was applied should be marked (with a box), and molecular weight/size standards should be labeled wherever possible. Source Data files will be made available to reviewers during evaluation of revised manuscripts and, if your paper is eventually published in JGP, the files will be directly linked to specific figures in the published article.

Source Data Figures should be provided as individual PDF files (one file per figure). Authors should endeavor to retain a minimum resolution of 300 dpi or pixels per inch. Please review our instructions for export from Photoshop, Illustrator, and PowerPoint here: <https://rupress.org/jgp/pages/submission-guidelines#revised>

Whilst you are revising your manuscript, we ask that you consider whether you have any artwork that might be suitable for the cover of JGP. Microscopy images are particularly good for cover artwork, but other types of image can be very effective, so we encourage you to be creative. Please don't restrict yourself to images from the paper; an image that is relevant to the work

described would be just as suitable. Images should be a minimum resolution of 300 dpi. To see recent examples, visit the following page and click on 'Show covers? Yes': <https://jgp.rupress.org/content/by/year>)

Thank you for submitting your interesting research to JGP.

Please submit your revised manuscript, and any associated files, via this link:
Link Not Available

Sincerely,

Jeanne

Jeanne Nerbonne, Ph.D.
On behalf of Journal of General Physiology

Journal of General Physiology's mission is to publish mechanistic and quantitative molecular and cellular physiology of the highest quality; to provide a best-in-class author experience; and to nurture future generations of independent researchers.

Reviewer #1 (Comments to the Authors):

The authors have accepted many of the recommendations and the revised manuscript is improved. This reviewer was particularly appreciative of the extra experiments for forskolin and the tempering of language regarding a physical binding. As someone who has patch-clamped rat ventricular myocyte sodium currents, it is possible to record these with high fidelity in whole-cell mode. Low access resistance after break-in, stable access, and high compensation are required. It can be laborious/slow, but something for the authors to be aware of in future studies, as this configuration can offer new insights into their fascinating PKA pathway.

The following are considered very minor adjustments for further clarity.

Spelling of 'validated' in table 1.

"Patch clamp recording took time: on average, I could do ~ 5 patches within a one-hour time limit after myocytes were removed from the incubator and placed in a cell chamber on the microscope stage superfused with room temperature Tyrode's solution. This one-hour time limit was set to minimize the confounding factor of dissipating PKA effects on Nav channel distribution. I began patch clamping CON myocytes roughly 4 hr after myocyte isolation, and alternated recordings between CON and PKA myocytes over 10 - 12 hr total patch clamp time."

Include temporal details of patch-clamp in methods for future reproducibility.

"The current traces shown in Fig. 1B were chosen because they were obtained on the same day (November 1, 2022), at 5 - 6 pm (CON myocytes) and 11 pm - midnight (PKA myocytes)."

Representative traces should be representative of the data. These are exceptional traces (or exceptional paired traces). Small change of language for precision.

"Leaky myocytes refer to those that are dying or dead in the culture dishes, which inevitably occurs during myocyte isolation and culture."

The authors should add dead or dying in parenthesis beside the word leaky for precision.

Reviewer #2 (Comments to the Authors):

The authors have addressed many of the critiques from the previous review. However, some still remain.

Is 6 - 15 hours persistent PKA activation within the physiological range for humans? It is unclear how compartmentalization relates to this question. Also, if a reductionist approach is taken, how are these findings to be interpreted? The rationale for these experiments is still not clear in the introduction.

Also in the introduction, there is still not enough information regarding EB1 (see previous review). This absence will make it difficult for a reader outside of this particular field to follow the manuscript and will reduce its impact.

Reviewer #3 (Comments to the Authors):

Although the authors have responded to some of my comments, the manuscript remains too long and difficult to read. There are too many figures, and it seems that moving some Figures to supplemental information and reducing the text would help the reading. Additionally, few improvements would strengthen the study as follows.

Major comments:

- 1- QPCR results of EB1 transcript expression presented in Figure S2 should be included in the main Figure 3 as it is major finding concerning the underlying regulatory mechanisms. Also, total expression of EB1 proteins, presented in Figure 4A, should be included in Figure 3. As it is a major finding, this should also be highlighted in the discussion section (line 522).
- 2- I do not understand the results presented in Figure 4C. Denser signals of b-Tub are observed on images, but the graph says that b-Tub is decreased. Also, you say in the text, lines 402-403, that this decrease is consistent with the decreased deY a-Tub. Explanations concerning this correlation is not provided, which makes it confusing, at least for someone like me who does not know much about these mechanisms. More detailed explanations are therefore required.
- 3- In Figure 9A, total expression of Nav1.5 and EB1 in WCL is increased in PKA, versus CON, conditions, which, as a consequence, increases the IP yields of Nav1.5 and of EB1. It is therefore logical that co-immunoprecipitated EB1 dimers and Nav1.5, respectively, also increase in the IPs. The conclusion made by the authors that PKA activation increases the Nav1.5-EB1 interaction is therefore not correct and should be removed.
- 4- Similar to my comment above, a graph showing GFP-Nav1.5 and EB1-mRFP protein expression has to be shown to verify that the increased PLA puncta density is not due to an increased expression of the two proteins, which would be expected based on the fact that the CMV promoter is regulated by PKA/CREB (which is indeed observed in HEK-293 cells).
- 5- In Figure 11, the b-Tub co-IP signals should be normalized by the Nav1.5 IP yields in CON and PKA conditions, as this later difference will be determinant in the ability to co-immunoprecipitate b-Tub. Also, how the b-Tub co-IP signals in CON (~2) and PKA (~3) (top graph) could result in a PKA:CON ratio of 4 (lower graph)? Also, the b-Tub co-IP signal in the EB1 IP is also present in the [-] IP, thus questioning the co-IP signal and resulting quantification. Could the authors provide with the other data source immunoblots to make sure the signal is specific?
- 6- Parts 2, 3 and 4 of discussion section are too long and speculative, and should therefore be reduced to few/shorter suggestions.
- 7- Does nocodazole treatment inhibit the effect of PKA activation on Nav1.5 cell surface expression? As previously requested, the authors should use other tools than chloroquine (such as nocodazole) to show involvement of specific mechanisms (trafficking or recycling) or not show this latter figure which necessarily brings the question.

Minor comments:

- 1- Results from Figure 1E is presented in the text after Figure 3. I would recommend presenting them with the results presented in Figure 1.
- 2- Data quantifications of Figures 1D and 4B, including data points in bar graphs, are missing.
- 3- In Figure 6B, the minus sign has been forgotten in the p value generated by data analysis of EB1 at ICD.
- 4- In Figure 8B, p value indicates no statistical differences ($p=0.27$) for Nav1.5/EB1 correlation coefficient, which is a surprising result. Could you please comment?

Authors' response to reviewers of JGP submission 202313436R1

Persistent PKA activation upregulates EB1, reorganizes microtubules and promotes increased cell surface expression of Nav1.5

Tytus Bernas, John Seo, Zachary T. Wilson,
Bihua Tan, Isabelle Deschenes, Christiane Carter, Jinze Liu and G.-N. Tseng

We thank all 3 Reviewers and the Editor for their careful consideration of our revised manuscript. In response to the Reviewers' comments, we have conducted new experiments (qPCR quantification of EB1 and Nav1.5 transcripts in CON and PKA myocytes), added new data analysis (Fig. 10B), and revised the manuscript accordingly. Revised/added text is highlighted by gray shading in the uploaded 'alteration-highlighted' copy. Below is our point-by-point response to the Reviewers' comments.

Reviewer #1

1. Correct typo in Table 1. Done. Thank you.
2. Include temporal details in patch-clamp in Methods for future reproducibility. The following text (slightly modified from the text specified by the Reviewer) has been added to the Methods section: *To minimize the confounding factor of dissipating PKA effects on Nav channel distribution, we limited the patch clamp recording time to 1 hour after myocytes were removed from the incubator and placed in a cell chamber on the microscope stage superfused with room temperature Tyrode's solution. On average, recordings from ~ 5 patches could be completed within this one-hour time limit. Patch clamp recording began on CON myocytes roughly 4 hr after myocyte isolation, and recording alternated between CON and PKA myocytes over 10 - 12 hr of total patch clamp time.*
3. Current traces shown in Fig. 1B should be described as 'exceptional', instead of 'representative'. The word 'representative' has been removed. The following sentence was added to Fig. 1B legend: *These exceptional traces were obtained on the same day (CON myocytes 5-6 pm, PKA myocytes 11 pm-midnight).*
4. The authors should add dead or dying in parenthesis beside the word leaky for precision. Done.

Reviewer #2

1. Is 6 - 15 hours persistent PKA activation within the physiological range for humans? ...The rationale for these experiments is still not clear in the introduction. We apologize for disappointing this reviewer. We have finished analyzing RNA-seq and proteomic experiments on CON and PKA myocytes (a *concise description* is presented below). These data reveal the protective as well as maladaptive consequences of persistent PKA activation in cardiac myocytes. The manuscript will be submitted for consideration of publication shortly, and thus we will not discuss its contents in the current manuscript.

Predicted protective effects of persistent (15 hr) PKA activation- (a) Increasing transcripts encoding mitochondrial proteins involved in fatty acid metabolism, electron transport chain, and ATP synthesis, and increasing the protein level of ACADV (acyl-CoA dehydrogenase very long chain, a mitochondrial enzyme catalyzing the first step of FA β -oxidation). (b) Increasing transcripts encoding K channel subunits, Na/K pump components, and SERCA2, but decreasing Na/Ca exchanger transcript. (c) Increasing transcripts encoding proteins involved unfolded protein response. (d) Prominent increase in scaffold proteins (spectrins, filamin-C, SORBS2), that may enhance trafficking of proteins and organelles from reservoirs to destinations.

Predicted maladaptive effects of persistent PKA activation - (a) A general suppression of integrin signaling, actin signaling, and paxillin signaling. (b) A general suppression of cellular response to stress and inflammation. (c) A marked increase in CaMKII δ protein level.

- In the introduction, there is still not enough information regarding EB1. In addition to the information on EB1 provided in the *Introduction*, we have provided more information on the relevance of EB1 to Nav1.5 distribution in *Results* (first paragraph of section '2': Persistent PKA activation upregulated EB1 and induced microtubule reorganization in ventricular myocytes'), and in *Discussion* (section '3': EB1 distribution and its relationship to microtubules in CON and PKA myocytes; section '4': Role of microtubules in Nav1.5 distribution and function; section '5' Patho-physiological implications).

Reviewer #3

Major comments

- QPCR results of EB1 transcript expression presented in Figure S2 should be included in the main Figure 3 as it is major finding concerning the underlying regulatory mechanisms. Also, total expression of EB1 proteins, presented in Figure 4A, should be included in Figure 3.

To meet this reviewer's request for qPCR data, we have conducted RT-qPCR experiments on RNA extracted from 10 groups of myocytes, CON and PKA each from 5 hearts. Although this reviewer did not ask for qPCR of Nav1.5, for the sake of completeness we have also conducted RT-qPCR on Nav1.5. These new data are presented in Fig. S2, along with the RNA-seq data.

We did not incorporate these new data in Fig. 3, to avoid breaking the logic flow.

- More detailed explanations concerning the correlation between Fig. 4C (decrease in myocyte center β -tub) and Fig. 4D (decrease in deY α -Tub) are required. To better explain the correlation between data presented in the two panels, the following text was added (which should be read within the context of the paragraph): 'This scenario is supported by the lack of deY α -Tub looping in PKA myocytes (Fig. 4D)'.
- Fig. 9A data do not support the claim that PKA activation increases the Nav1.5-EB1 interaction. The purpose of Fig. 9A was to show that Nav1.5 and EB1 coIP reciprocally, in contrast to the lack of co-IP in cardiac myocytes.
- Increased PLA puncta density could be due to increased expression of Nav1.5 and EB1-mRFP. We have reanalyzed the PLA data, normalizing the puncta density by the product of GFP-Nav1.5 and EB1-mRFP pixel contents (surrogates of protein expression, normalized by the mean of CON data). Our conclusion remains the same (updated Fig. 10B).
- In Fig. 11, how did the bTub coIP in CON and PKA result in a PKA/CON ratio of 4? The exact excel spread sheet for Fig. 11B data is shown on the right for the reviewer's inspection.

The bTub coIP signals should be normalized by the Nav1.5 IP efficiency and the bTub coIP is present in the [-] IP. Here we address this reviewer's two comments together, because they are related to the technical issues of quantifying co-IP. To properly quantify protein band intensities (i.e. avoiding oversaturation of ECL signals in WCL lanes), we adjusted the loading so that the WCL and 'WCL equivalent used in IP' was kept at 1:20. Because of

REDO 230825						
[+] IP/WCL of CON and PKA, and then PKA/CON						
IP: GFP rabbit			IP: EB1 rat			
	CON	PKA	PKA/CON	CON	PKA	PKA/CON
t-Test against '1': Two-Sample Assuming Unequal Variances						
	p=		0.156	p=		2.2395E-05
MEAN	2.55	3.95	3.68	3.47	1.32	0.42
SE	0.66	0.69	1.66	0.69	0.29	0.08
N	7	7	7	11	11	11
Exp 1	0.71	3.82	5.38	5.12	1.19	0.23
Exp 2	NA	NA	NA	5.57	2.73	0.49
Exp 3	0.02	0.26	13.00	3.23	1.75	0.54
Exp 4	NA	NA	NA	4.53	1.36	0.30
Exp 5	4.16	5.43	1.31	NA	NA	NA
Exp 6	NA	NA	NA	0.28	0.14	0.50
Exp 7	NA	NA	NA	8.23	0.97	0.12
Exp 8	NA	NA	NA	1.27	0.13	0.10
Exp 9	2.22	5.35	2.41	2.12	1.21	0.57
Exp 10	2.93	4.45	1.52	2.23	1.83	0.82
Exp 11	2.85	3.27	1.15	3.82	3.02	0.79
Exp 12	4.95	5.05	1.02	1.72	0.21	0.12

the high abundance of β -tubulin in HEK293 cells, we did see non-specific binding in [-] IP lanes. To compensate for the efficiency of Nav1.5 IP and then the non-specific β -tubulin signal in [-] IP will introduce undue variations to the data set. Therefore we presented our data as the degree of β -tubulin co-IP ([+] IP/WCL) and then PKA:CON of [+] IP/WCL, as was specifically requested by this reviewer during the last review.

6. Parts 2-4 of Discussion are too long. We respectfully disagree.
7. Does nocodazole inhibit the effect of PKA activation on Nav1.5 cell surface expression? No.

The authors should use other tools than chloroquine to show involvement of specific mechanisms.

We respectfully disagree. There are many experimental strategies to query the pathways of protein trafficking and their regulation. Given the large amount of data presented in the manuscript, especially given this reviewer's criticism about the length of the text and the number of figures, doing more explorative experiments is beyond the scope of this study.

Minor comments:

1. Results from Fig. 1E should be presented in Fig. 1, not after Fig 3. We cannot do this because it will break the flow of our logic.
2. Data in Fig. 1D and Fig. 4B should be quantified. Data quantification is presented in Fig. 2, Fig. 6 and Fig. 7.
3. Fig. 6B minus sign of p value exponent is missing. This is corrected. Thank you.
4. Fig. 8B, p=0.27 for Nav1.5/EB1 correlation coefficient? This is due to the large spread in this particular data set. We have quantified the spatial relationship between Nav1.5 and EB1 at intercalated disc using a different approach and reached the same conclusion (presented below). This latter analysis will be included in a separate technical report, and thus will not be used in the current one.

Fractions of fluorescence signals associated between channels were computed using Ripley (K) correlation. Briefly, autocorrelation functions (ACF) were calculated in 64 – 1400 nm range at voxels positioned in a cardiomyocyte in regular intervals (forming 320 x 320 x 960 nm mesh), for each of the color channels. Subpopulations of voxels where n-cadherin ACF (64 – 110 nm distance) was positive (>1) were selected to represent ICDs. Voxels with positive EB1 and Nav1.5 ACFs were isolated in the ICD subpopulations in the same fashion. Fractions of EB1 voxels positive with Nav1.5 ACF were calculated to represent association of EB1 signals with their Nav1.5 counterparts. Association between Nav1.5 and EB1 was calculated in a similar manner.

Shown in the panel are association fraction of EB1 with Nav1.5 (fraction of Nav1.5-positive voxels that are also EB1-positive), and the reciprocal. The p values are from U-test (Mann-Whitney or rank sum, because the distribution of data points was not normal) on 37-40 myocytes each of CON and PKA groups.

November 29, 2023

Dr. Gea-Ny Tseng
Virginia Commonwealth University
Physiology & Biophysics
1101 East Marshall Street
Sanger 3-042a
Richmond, Virginia 23298

Re: 202313436R2

Dear Gea-Ny,

I am pleased to let you know that your manuscript, titled "Persistent PKA activation upregulates EB1, reorganizes microtubules and promotes increased cell surface expression of Nav1.5" is scientifically acceptable for publication in the Journal of General Physiology. Formal acceptance will follow when it is modified in accordance with our editorial policies.

Please note items that need attention are listed at the bottom of this email (under 'manuscript formatting checklist') and on the attached marked-up pdf file. Please also be sure to include a letter addressing the reviewers' comments point-by-point and a copy of the text with any alterations highlighted (boldfaced or underlined). Your manuscript should be a double-spaced MS Word file and include editable tables, if appropriate.

JGP now requires a data availability statement for all research article submissions. These statements will be published in the article directly above the Acknowledgments. The statement should address all data underlying the research presented in the manuscript. Please visit the JGP instructions for authors for guidelines and examples of statements at <https://rupress.org/jgp/pages/editorial-policies#data-availability-statement>.

Also, please be sure your Source Data Figures are provided as individual PDF files and only *one* file per figure. (Authors should endeavor to retain a minimum resolution of 300 dpi or pixels per inch).

Lastly, JGP adds short captions to articles listed on our weekly newest article emails. If you haven't, please provide a short, ~40-word summary statement for the online JGP table of contents and alerts. This summary should describe the context and significance of the findings for a general readership and be placed on/near the title page.

Please submit your final files via this link:

Link Not Available

Thank you for choosing to publish your research in JGP and please feel free to contact me with any questions.

Sincerely,

Jeanne

Jeanne Nerbonne, Ph.D.
On behalf of Journal of General Physiology

Journal of General Physiology's mission is to publish mechanistic and quantitative molecular and cellular physiology of the highest quality; to provide a best in class author experience; and to nurture future generations of independent researchers.

Manuscript formatting checklist:

- MS Word document of text needed (including editable tables)
- MS Word document of supplemental text needed, if applicable (including figure legends and editable tables)
- Brief Statement describing supplementary information needed, if applicable (in subsection at end of Materials & Methods)
- Please include a data availability statement preceding the Acknowledgments section. Please see <https://rupress.org/jgp/pages/editorial-policies#data-availability-statement>
- Figures created at sufficient resolution and in acceptable format (including supplemental if applicable). If working in Illustrator, we prefer .ai or .eps file format. If working in Photoshop please use 600dpi/1000dpi .tiff or .psd file format. Minimum resolution at estimated print size: Minimum resolution for all figures is 600 dpi. For figures that contain both photographs and line art or text, 600 dpi is highly recommended. Figures containing only black and white elements (line art, no color, and no gray) should be

1,000 dpi. Maximum figure size is 7 in wide x 9 in high (17.5 x 22.8 cm) at the correct resolution. <https://jgp.rupress.org/fig-vid-guidelines>

- Supplemental figures, if any, conforming to same guidelines as manuscript figures (noted above)
 - If images resemble one from a prior publications, the author must seek permissions (to reproduce or adapt) from the original publisher. [You can resubmit your paper while waiting to hear back from the original publisher but please keep us updated]
 - All authors must complete a disclosure form prior to acceptance. A link to complete the form has been sent to all coauthors. Please provide the editorial office with updated email addresses if necessary
-

Reviewer #1 (Comments to the Authors):

The authors have addressed all concerns and no further edits are warranted.

Reviewer #2 (Comments to the Authors):

The revision was not responsive to comments. The relevance of the experimental model to human physiology is not clearly explained and key background information is spread throughout the manuscript, making it difficult to follow. Overall, the introduction falls short of what would be expected.

Reviewer #3 (Comments to the Authors):

Although the authors have responded to most of my comments, two more improvements concerning (1) interpretation of co-immunoprecipitation experiments and (2) organization of figures could be provided as follows.

(2) I continue thinking that transcript analysis of EB1 and Nav1.5 are main findings that shall be included in a main figure, not in a supplemental figure.

(2) Figure 5 is more technical and could be a supplemental figure.

(2) I would recommend pooling Figures 7 and 8.

(1) I disagree with the conclusion made concerning the quantification of EB1 in Nav1.5 immunoprecipitates (left panel of Figure 9A, and lines 459-460). In my opinion, the increased abundance of EB1 in Nav1.5 IP is most likely the direct consequence of increased Nav1.5 (and EB1) expression in PKA vs CON, but not the consequence of an increased interaction. The same conclusion can be made from experiments illustrated in Figure 11 (left panel). In addition, the statistical analysis shows non-significant differences so I would not mention about a "trend" (lines 498-500). I also disagree with the conclusion made from the experiments presented in the right panel of Figure 11 because the increased expression of EB1, and of b-Tub, makes quantification of interaction complicated and most likely unreliable. Additionally, the interaction of b-Tub with EB1 does not seem to be specific as b-Tub is also detected in the control IP. I am therefore not convinced by the authors' arguments, including those presented in their rebuttal, and I would therefore recommend the authors to revise/tone down all of their conclusions concerning co-IP experiments.

(2) Figures 9B and 9C can be included as a supplemental figure.

Authors' response to reviewers of JGP submission 202313436R2
Persistent PKA activation redistributes Nav1.5
to the cell surface of adult rat ventricular myocytes

Tytus Bernas, John Seo, Zachary T. Wilson,
Bi-hua Tan, Isabelle Deschenes, Christiane Carter, Jinze Liu and G.-N. Tseng

We thank all 3 Reviewers and the Editors for their careful consideration of our manuscript. Below is our point-by-point response to the Reviewers' comments.

Reviewer #1

The authors have addressed all concerns and no further edits are warranted. Thank you.

Reviewer #2

The revision was not responsive to comments. The relevance of the experimental model to human physiology is not clearly explained and key background information is spread throughout the manuscript, making it difficult to follow. Overall, the introduction falls short of what would be expected. We apologize for not being able to meet the demands of this Reviewer. Our project is still work in progress, and this Reviewer's critiques are helpful in our current effort of exploring the protective as well as maladaptive consequences of persistent PKA activation in adult cardiac myocytes.

Reviewer #3

(2) I continue thinking that transcript analysis of EB1 and Nav1.5 are main findings that shall be included in a main figure, not in a supplemental figure. We respectfully disagree. The Nav1.5 transcript was reduced yet the protein level stayed the same. Moving the Nav1.5 transcript data to the main figure will create confusion. On the other hand, increase in the EB1 transcript level after β -adrenergic stimulation (thus PKA activation) has been documented in the literature.

(2) Figure 5 is more technical and could be a supplemental figure. This figure was switched from a supplemental figure to a main figure in response to a specific request by Reviewer 1 on the original version.

(2) I would recommend pooling Figures 7 and 8. This will create a very crowded figure difficult for the readers.

(1) I am therefore not convinced by the authors' arguments.... and I would therefore recommend the authors to **revise/tone down all of their conclusions concerning co-IP experiments.**

We have modified the text to satisfy this reviewer's request. These are listed below:

1. Result, section 3, related to Fig. 9A: PKA activation increased EB1 dimer co-immunoprecipitated with GFP-Nav1.5, **suggesting** stronger interaction between the two.
2. Results, section 3, related to Fig. 11: **apparently** stronger Nav1.5/ β -Tub co-immunoprecipitation in PKA-treated HEK293 cells (Fig. 11).
3. Discussion, section 3: Persistent PKA activation ... **apparently** strengthened Nav1.5/ β -Tub association (Fig. 11).

Note that the EB1 protein expression was increased in PKA-activated HEK293 cells, yet β -Tub co-IP with EB1 was reduced. There is no need to tone down this part of the Results.

(2) Figures 9B and 9C can be included as a supplemental figure. We respectfully disagree. Fig. 9B and 9C are integral parts of Fig. 9. Together, these data illustrate a critical point: the success of co-immunoprecipitation, or a lack thereof, can depend on the cellular environment used for protein expression.